# Structural basis of apoptosis induction by the mitochondrial voltage-dependent anion channel

Melina Daniilidis [1,6], Umut Günsel [1,2,6], Georgios Broutzakis [3], Kira D. Leitl[1], Robert Janowski [2], Kai Fredriksson[1,5], Dierk Niessing [2,4], Christos Gatsogiannis [3] & Franz Hagn [1,2] ✉

The voltage-dependent anion channel (VDAC) is the main gateway for metabolites across the mitochondrial outer membrane. VDAC oligomers are connected to apoptosis induced by various stimuli. However, the mechanistic and structural basis of apoptosis induction by VDAC remains poorly understood. Here, using cryo-EM and NMR we show that VDAC1 oligomerization or confinement in small lipid nanodiscs triggers the exposure of its N-terminal α-helix (VDAC1-N) which becomes available for partner protein binding. NMR and X-ray crystallography data show that VDAC1-N forms a complex with the BH3 binding groove of the anti-apoptotic Bcl2 protein BclxL. Biochemical assays demonstrate that VDAC1-N exhibits a pro-apoptotic function by promoting pore formation of the executor Bcl2 protein Bak via neutralization of BclxL. This mechanism is reminiscent of BH3-only sensitizer Bcl2 proteins that are efficient inducers of Bax/Bak-mediated mitochondrial outer membrane permeabilization and ultimately apoptosis. The VDAC pathway most likely responds to mitochondrial stress or damage.

The mitochondrial voltage-dependent anion channel (VDAC) is the major transit pore of (energy-) metabolites and metal ions across the outer mitochondrial membrane[1]. VDAC plays a major role in the molecular pathology of e.g., Alzheimer's[2] and Parkinson's diseases or cancer[3], and cardiac ischemia–reperfusion injury[4,5]. VDAC1 is a porin composed of a 19-strand β-barrel and an N-terminal α-helix (VDAC1-N) attached to the inside of the pore[6–8]. The inner surface of the β-barrel is decorated with positive charges that provide a preference of the channel for negatively charged substrates, such as glutamate[9] and ATP[10], while also transporting positively charged metabolites like acetylcholine or dopamine[9]. Electrophysiology experiments showed that VDAC1 can adopt an open state at low to zero and a closed state at high membrane potential[11], shifting the channel's selectivity to small cations[12–14].

In addition to its role in mitochondrial metabolite transport, VDAC1 has also been implicated as a key player in programmed cell death. VDAC1 has been reported to act as an inducer of apoptosis[15] via mitochondrial outer membrane permeabilization (MOMP)[16], a process that enables the release of pro-apoptotic proteins from the inter-membrane space (IMS) to activate caspases[17]. However, the inner pore of the VDAC1 monomer has only a diameter of 1.5 to ~3.0 nm[6–8,18,19], which renders it too small for the release of pro-apoptotic proteins. Mitochondrial damage caused by various inducers of apoptosis, including inhibitors of the respiratory chain or hydrogen peroxide, has been reported to cause VDAC1 oligomerization and eventually apoptosis[15]. Thus, it has been suggested that such oligomers might form large-enough pores in the membrane to execute MOMP[20] and

[1]Structural Membrane Biochemistry and Bavarian NMR Center (BNMRZ), Department of Bioscience, School of Natural Sciences, Technical University of Munich, Garching, Germany. [2]Molecular Targets and Therapeutics Center (MTTC), Institute of Structural Biology, Helmholtz Munich, Neuherberg, Germany. [3]Institute for Medical Physics and Biophysics and Center for Soft Nanoscience, Westfälische Wilhelms Universität Münster, Münster, Germany. [4]Institute of Pharmaceutical Biotechnology, Ulm University, Ulm, Germany. [5]Present address: AGC Biologics, Copenhagen, Denmark. [6]These authors contributed equally: Melina Daniilidis, Umut Günsel. ✉e-mail: franz.hagn@tum.de

even release mitochondrial DNA fragments[21]. VDAC1 has a tendency to form dimers and oligomers in detergent solution[22] and native membranes[19], but stable oligomerization in a cellular environment requires additional stimuli such as altered mitochondrial lipid composition[23,24], increased $Ca^{2+}$ levels[25,26], low pH[27], or oxidative stress[21,28].

The interaction between VDAC1 and the anti-apoptotic Bcl2 protein BclxL has been reported to be considerably enhanced by apoptotic stimuli[29,30], which was confirmed by in vitro[22] and cellular studies[31,32]. This interaction was reported to be essential for the execution of apoptosis under mitochondrial stress[33]. The Bcl2 protein family is the canonical system to induce MOMP[34,35], where the activation of the pore-forming Bcl2 proteins Bak and Bax leads to large membrane pores of 40 nm to 1 μm in diameter[36,37] that allow for the exit of folded proteins or even mitochondrial DNA[38]. This process is constantly inhibited by the anti-apoptotic Bcl2 protein members, such as BclxL, Bcl2, or Mcl1[35]. Thus, a protein that can neutralize the anti-apoptotic Bcl2 members inevitably leads to apoptosis via MOMP induced by activated Bax/Bak[39].

Despite this large body of evidence, the exact mechanistic role of VDAC in the intrinsic induction pathways of apoptosis remains highly controversial[16]. To address this important question, structural studies of the complex between VDAC1 and BclxL are very likely to provide mechanistic insights into how VDAC and the Bcl2 protein family are connected in the regulation of apoptosis.

Here, we use structural and biochemical methods to explore the mechanistic basis of MOMP induction by VDAC1 via its interaction with the anti-apoptotic Bcl2 protein BclxL. We show that VDAC1 oligomerization in negatively charged detergents or lipids leads to the exposure of the VDAC1-N-terminal α-helix (VDAC1-N). Using cryo-EM in circularized lipid nanodiscs of different sizes[40,41], we structurally characterize VDAC1 in different conformational states where VDAC1-N is either bound inside the pore or exposed to the pore exterior. Using NMR, we show that binding to BclxL is only possible if VDAC1-N becomes exposed. We confirm and validate our findings by solving a high-resolution crystal structure of the VDAC1-N–BclxL complex and alanine scanning experiments, respectively. In addition, pore-forming assays in liposomes provide evidence that VDAC1-N can dissociate the inhibitory complex between BclxL and the pro-apoptotic Bcl2 protein Bak, restoring its pore-forming activity. We conclude that apoptotic stimuli that induce VDAC1 oligomerization lead to the exposure of VDAC1-N, which in turn can act as a sensitizer for BH3 protein to inhibit anti-apoptotic Bcl2 proteins, leading to VDAC1-dependent MOMP.

## Results

### VDAC1 oligomerization leads to exposure of its N-terminal α-helix

First, we wondered whether the N-terminal α-helix of VDAC1, the putative interaction site of Bcl2 proteins[32], can become exposed under apoptotic conditions, i.e., in the oligomeric state of VDAC1, which has been shown to be a marker for apoptosis[15] (Fig. 1a). In the published structures of VDAC1, the N-terminal α-helix is stably attached to the pore interior[6–8]. Thus, a conformational change must take place to allow for helix dissociation from the β-barrel wall. To address this question, we used chemical crosslinking experiments to identify conditions that favor VDAC1 oligomerization in detergent micelles and liposomes. We screened several detergent conditions that promote VDAC1 oligomerization, which was probed by chemical crosslinking experiments with the amino-selective crosslinker bis(sulfosuccinimidyl)suberate (BS³) (Fig. 1b). Not only the zwitterionic detergent LDAO, which has been widely used for structural studies of VDAC1, but also the milder detergent TritonX-100 induced only moderate VDAC1 oligomerization. In contrast, the negatively charged bile acid cholate strongly favors -VDAC1 oligomerization, whereas the structurally very similar zwitterionic bile acid derivative CHAPS does not show this

effect. In a liposome environment, the formation of very large VDAC1 oligomers can be facilitated by negatively charged lipids such as 1-palmitoyl-2-oleyl-glycero-3-phosphoglycerol (POPG) (Supplementary Fig. 1d). To probe whether VDAC1 oligomerization triggers the exposure of its N-terminal α-helix, we used the cysteine-specific maleimide-polyethyleneglycol reagent with a molecular mass of 40 kDa (PM40) to chemically modify a VDAC1 variant with a single cysteine at position 6 (T6C). The native cysteines were conservatively mutated to alanine or serine (C127A and C232S). T6C is located at the N-terminal end of the α-helix that is not exposed to the pore exterior in the helix-inserted canonical state (see VDAC1 cartoon model in Fig. 1c). Due to a calculated hydrodynamic radius of ~6.2 nm[42], the 40 kDa PEG polymer cannot easily access the cysteine thiol moiety inside the pore with an inner diameter of 1.5–3 nm. Thus, the modification reaction takes place more efficiently if the N-terminal part of VDAC1 becomes exposed to the solvent. In very good agreement with the oligomerization assay, the PM40-modification assay of VDAC1 shows a very pronounced reaction in the detergent cholate and to a much lesser extent in the other detergents that do not bear a net negative charge (Fig. 1c). To rule out any adverse effects of the detergents on the VDAC1 structure, we measured CD spectra and thermal melting transitions of the protein at the tested detergent conditions. In all the detergents except SDS, serving as the negative control, VDAC1 showed a characteristic β-sheet secondary structure (Supplementary Fig. 2a) with a cooperative unfolding transition (Supplementary Fig. 2b), indicating a compactly folded protein at ambient temperature, independent of the detergent concentration. The degree of VDAC1 oligomerization and VDAC1-N exposure is not markedly altered at increasing detergent concentrations (Supplementary Fig. 2c, d). These data suggest that VDAC1 oligomerization and the exposure of the N-terminal helix are directly correlated, which would imply that, in turn, VDAC1 stabilization can possibly reduce its degree of oligomerization. To test this hypothesis, we used the well-characterized VDAC1-E73V variant and observed a higher thermal stability than for wt-VDAC1 (Supplementary Fig. 2e). VDAC1-E73V was shown to exhibit reduced dynamics in the β-barrel and a more stable attachment of its N-terminal α-helix[6,43], which facilitated its NMR structure determination[44,45]. In line with these previous data, this mutation caused a reduction in VDAC1 oligomerization in our experiments (Supplementary Fig. 1c) and a lower degree of exposure of its N-terminal helix (Supplementary Fig. 1a). If the bulky hydrophobic side chain of Leu10 within the VDAC1-N-terminal helix that mediates the interaction with the β-barrel was replaced by a cysteine (L10C) in VDAC1-E73V, helix exposure and the degree of oligomerization were again increased (Supplementary Fig. 1b, c), corroborating the assumption of a connection between helix exposure and VDAC1 oligomerization.

In pure lipids, VDAC1 generally showed a lower tendency to form oligomers, partially because higher amounts of VDAC1 could not be inserted into liposomes ($c_{max}$ ~15 μM). Despite this limitation, we could detect the formation of very large oligomeric species in the presence of the negatively charged lipid POPG (Supplementary Fig. 1d), which also led to a higher exposure of the N-terminal helix as monitored by the PM40 assay (Supplementary Fig. 1e). The effect of negatively charged lipids can be rationalized by the positive net charge of VDAC1-N, suggesting that the helix, once exposed, can interact electrostatically with the membrane surface, as reported previously[46–48]. Similarly, the addition of negatively charged cholesteryl hemisuccinate (CHS) into liposomes caused a slight increase in the population of the oligomer (Supplementary Fig. 1f) and VDAC1-N exposure (Supplementary Fig. 1g).

In order to structurally characterize VDAC1 in the helix inserted or exposed states, we used lipid nanodiscs of 10 (with MSP1D1) and 8 nm (with MSP1D1ΔH5) diameter (8 and 6 nm inner diameters, respectively)[49–51] where VDAC1 with an outer diameter of ~5.5 nm can be inserted together with a varying amount of residual lipids between

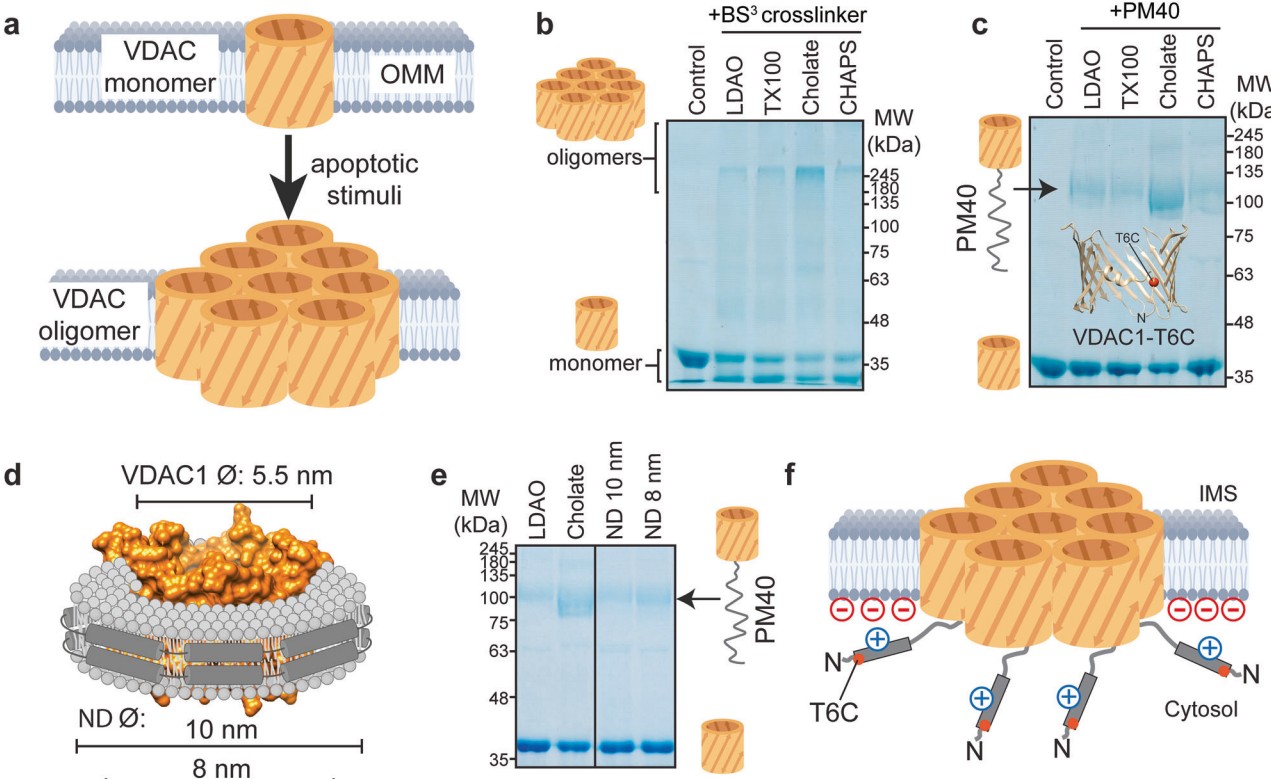

**Fig. 1 | VDAC1 exposes its N-terminal α-helix in the oligomeric state. a** Upon various apoptotic stimuli, VDAC1 was shown to form oligomers in the outer mitochondrial membrane (OMM). **b** VDAC1 oligomerization in detergent micelles, as detected by amino-specific crosslinking with bis(sulfosuccinimidyl)suberate (BS³), is favored by a negative net charge, as present in the detergent cholate. The VDAC1 monomer appears as double bands after crosslinking: unmodified (upper) and modified with internal crosslink (lower). **c** Modification of VDAC1 at the sulfhydryl side chain of cysteine 6 (T6C) in a cysteine-free background by maleimide-polyethyleneglycol of 40 kDa (PM40) occurs more efficiently in the oligomeric form. **d** Lipid nanodiscs (ND) of different sizes mimic the oligomeric state of VDAC1

to various extents. VDAC1 is shown in orange, the membrane scaffold protein (MSP) in dark gray, and lipids in light gray. **e** The smaller nanodisc (8 nm diameter) leads to a more pronounced exposure of cysteine 6 in VDAC1 than the 10 nm nanodisc, as probed by PM40 modification. **f** The PM40 data suggest that in the oligomeric state or if inserted into small lipid nanodiscs, VDAC1 exposes its N-terminal α-helix to the solvent. A net negative charge on the membrane surface facilitates helix exposure by electrostatic interactions with the positively charged α-helix. Data in (**b**, **c**, **e**) are representative of n = 3 technical replicates with similar results. Cartoon models in this figure were created in BioRender: Hagn, F. (2025) https://BioRender.com/9lrsde9.

the membrane protein and the membrane scaffold protein (MSP) (Fig. 1d). In these experiments, we anticipated that a tight nanodisc environment might mimic the oligomeric state to induce a structural transformation as seen in cholate micelles. Helix exposure in VDAC1 in lipid nanodiscs was again probed by the PM40 assay. Our data show that the exposed N-terminal helix conformation of VDAC1 is stabilized in smaller nanodiscs, as seen by the stronger band in the SDS-PAGE at ~100 kDa in 8 nm nanodiscs, representing the VDAC1-T6C-PM40 conjugate (Fig. 1e, Supplementary Fig. 1a, b). Since VDAC1 is present mainly as a monomer in nanodiscs, the effect of the E73V mutation on helix exposure was less pronounced than in detergent micelles. Furthermore, thermal melting experiments with VDAC1 in nanodiscs of different sizes, where the α-helix is either located mainly inside the pore (10 nm) or exposed (8 nm), show that the exposed structural state is less stable (Supplementary Fig. 2f), suggesting that the absence of the N-terminal helix at the barrel wall leads to structural instability of VDAC1, which is in line with a loss in functionality and voltage gating activity reported previously[33,52]. Our data presented here indicate that VDAC1 oligomerization induces the exposure of the N-terminal α-helix (Fig. 1f), whereas in the monomeric state, the helix is stably attached to the interior of the β-barrel.

### Cryo-EM structures of VDAC1 in lipid nanodiscs reveal conformational switching
Motivated by the biochemical experiments where the extent of α-helix exposure in 10 nm and 8 nm nanodiscs differed markedly, we further

set out to characterize the involved structural states of VDAC1 using cryo-EM. For cryo-EM, we used covalently circularized lipid nanodiscs (with cMSP) of enhanced stability and homogeneity but with a slightly larger (~1 nm) diameter[40,41,53,54].

Reconstitution of VDAC1 in cMSP1D1 nanodiscs (11 nm diameter) followed by size-exclusion chromatography (SEC) yielded two major peaks, both of which were analyzed using single-particle cryo-EM (Supplementary Fig. 3a, b). The SEC peak eluting earlier corresponds to the monomeric state of VDAC1 (Fig. 2a–d, Supplementary Fig. 3). For this species, we obtained a map at a resolution of 7.2 Å from 169,625 particles, showing the VDAC1 protomer residing at the lateral edge of the nanodisc (Supplementary Fig. 4, upper left panel, Supplementary Table 1). The N-terminal helix is resolved and located inside the pore at the expected position (Fig. 2c, d). The monodisperse cryo-EM sample of the late-eluting peak corresponds to VDAC1 dimers (Fig. 2e–h, Supplementary Fig. 3e). We were able to obtain a map of the VDAC1 dimer at a resolution of 7 Å from 31,802 particles, with no symmetry imposed (Supplementary Fig. 4, upper right panel, Supplementary Table 1). The N-terminal helix of each protomer is resolved as a clear rod-like density localized within the interior of the barrel (Fig. 2g, h). Thereby, the nanodisc adopts an oval shape to accommodate the VDAC1 dimer, establishing direct interactions between the membrane scaffold protein and the two VDAC1 barrels. The dimerization interface of the two VDAC1 protomers is formed by that side of the β-barrel where the N-terminal helix is attached to, as seen previously in a crystal structure of human VDAC1 dimers[55].

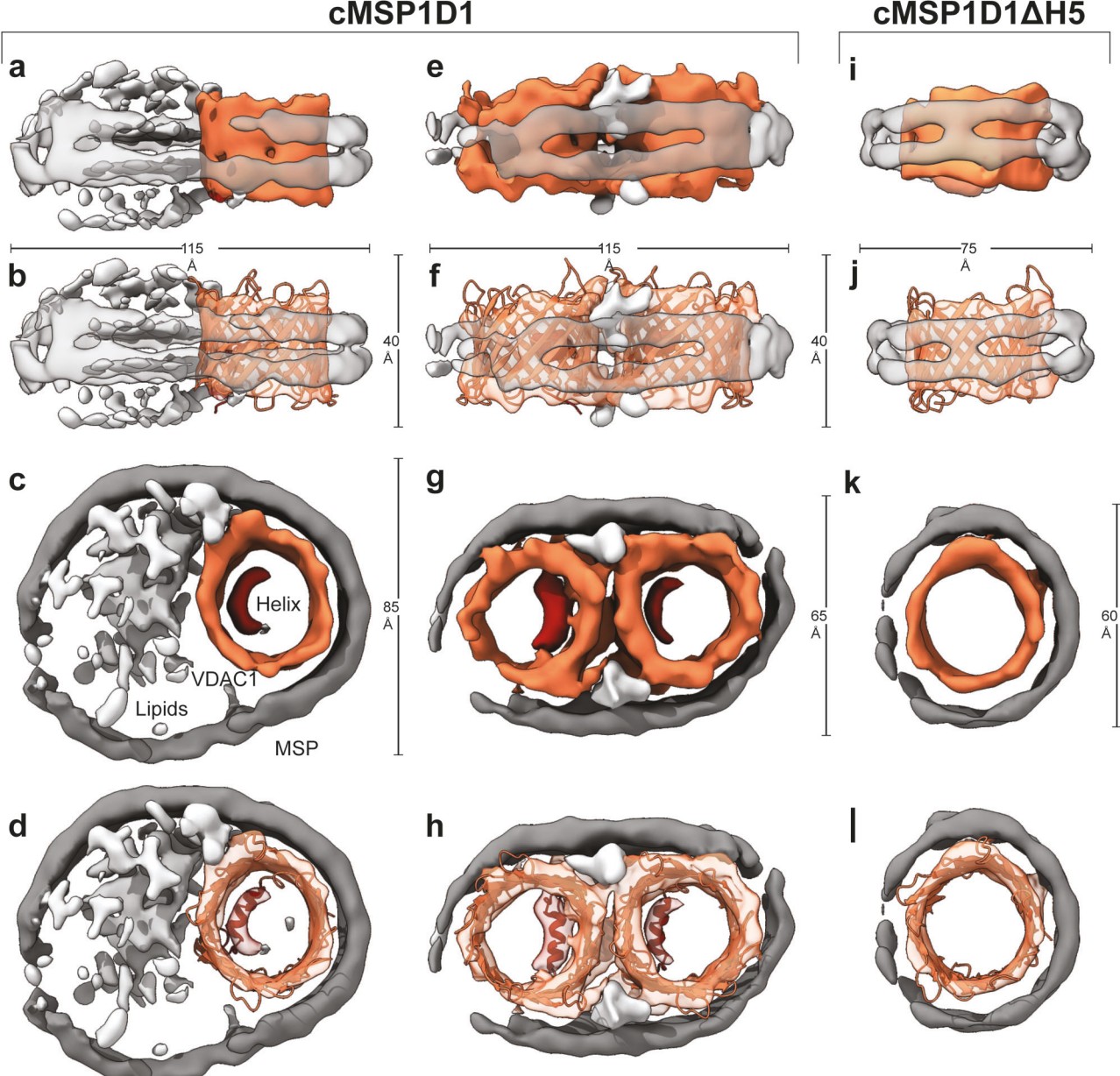

**Fig. 2 | Cryo-EM of VDAC1 reveals different conformational states of its N-terminal helix.** Reconstruction of monomeric (**a**–**d**) and dimeric VDAC1 (**e**–**h**) in cMSP1D1 lipid nanodiscs of 11.5 nm diameter. **i**–**l** Reconstruction of VDAC1 reconstituted in cMSP1D1ΔH5 lipid nanodiscs of 7.5 nm diameter, where VDAC1-N is absent inside the pore (**k**, **l**). MSP is colored dark gray, lipid noise in light gray, VDAC1 in orange, and the internal N-terminal α-helix in red.

We then set out to characterize VDAC1 in cMSP1D1ΔH5 nanodiscs (9 nm diameter), repeating the same strategy as for VDAC1 in cMSP1D1 nanodiscs (Supplementary Fig. 4, lower panel, Supplementary Table 1). Gel filtration revealed a single peak roughly at the same volume as the dimers in cMSP1D1, together with a high molecular weight peak, most likely representing protein-lipid aggregates (Supplementary Fig. 3a).

A subsequent cryo-EM analysis of the lower molecular weight SEC peak revealed several populations. Roughly 20% of the particles are in the canonical, helix-inserted monomeric state with a slightly depleted lipid content (Supplementary Fig. 3f, Supplementary Fig. 4). However, as verified in several independent reconstitutions, most particles (~65%) formed smaller nanodiscs with VDAC1 tightly interacting with the MSP of the nanodisc with no lipid density visible (Fig. 2k), giving rise to a slightly reduced diameter of 7.5 nm. Interactions between the MSP and an incorporated membrane protein have been described previously[56]. While VDAC1 in those "tight" nanodiscs shows an

expected inner diameter of ~3 nm, 15% of the particles contain VDAC1 with larger pore diameters, ranging up to 6 nm (Supplementary Fig. 3f). Due to their rarity and heterogeneity, particles belonging to the latter classes were excluded from further processing.

In contrast to the reconstructions of VDAC1 in cMSP1D1 nanodiscs, where the N-terminal helix was clearly resolved within the pore already at the stage of the initial ab-initio reconstruction, the VDAC1 reconstruction in cMSP1D1ΔH5 nanodiscs showed only residual noise at the expected position of the N-terminal helix within the pore (Fig. 2k). These observations strongly support that in this particle subset the structural state of the internal helix is substantially changed as compared to the canonical VDAC1 structure. The absence of N-terminal helix density suggests that it is not stably attached to the β-barrel but rather flexible, most likely becoming exposed to the channel exterior and thus accessible for interaction partners. This structural data is consistent with the biochemical PM40-modification

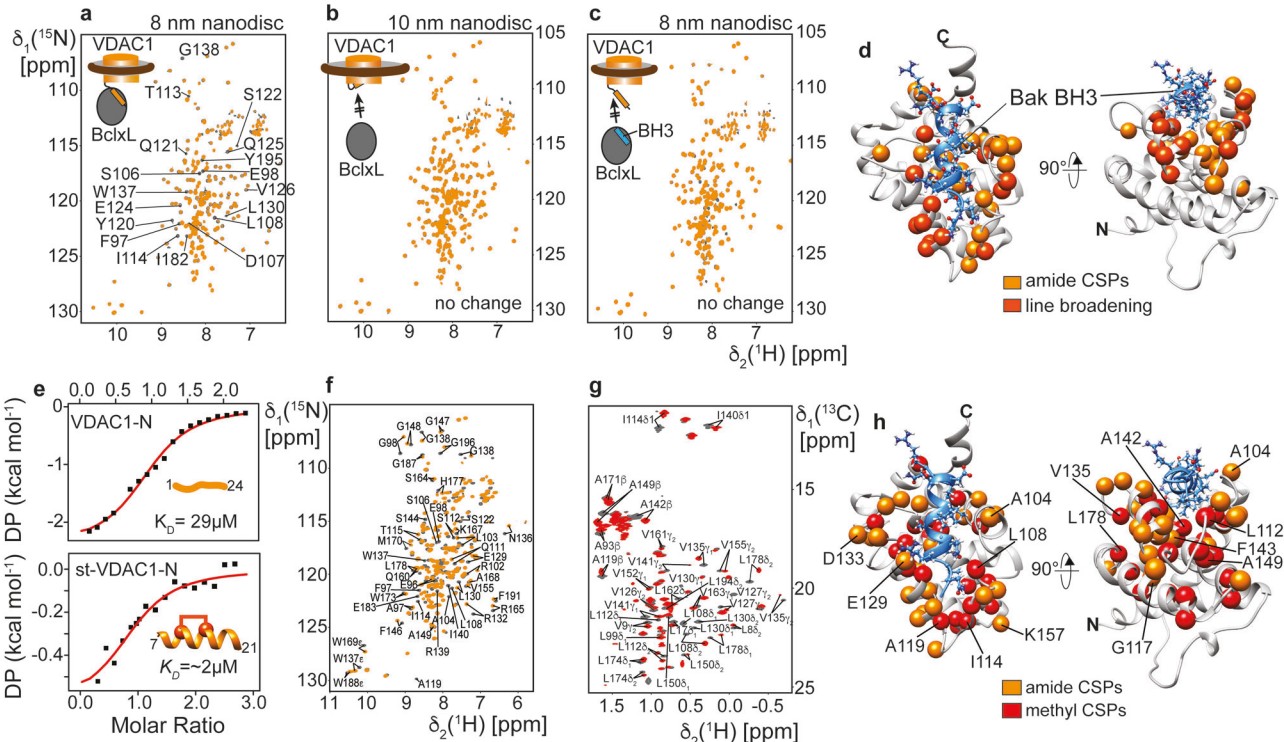

**Fig. 3 | VDAC1-N-terminal α-helix interacts with anti-apoptotic BclxL. a–c** NMR titration experiments with 100 μM of isotope-labeled BclxL (dark gray spectrum) and after the addition of 200 μM of VDAC1 in lipid nanodiscs (orange spectrum). **a** VDAC1 in 8 nm nanodiscs induces chemical shift perturbations (CSPs) and line broadening in BclxL. A selection of affected signals is labeled in the spectrum. **b** VDAC1 in 10 nm nanodiscs does not show binding to BclxL. **c** BclxL saturated with a high-affinity PUMA BH3 peptide does not interact with VDAC1 in 8 nm nanodiscs, indicating a competitive binding scenario. **d** NMR CSPs and NMR line broadening effects from (**a**) mapped onto the structure of BclxL. These effects cluster around the canonical BH3 binding site of BclxL. A bound Bak-BH3 peptide is depicted in blue, highlighting that VDAC1-N binds to the canonical BH3 binding site of BclxL. **e** Isothermal titration calorimetry (ITC) with BclxL and a VDAC1-N-terminal peptide (VDAC1-N, top panel) or a hydrocarbon-stapled VDAC1 peptide (stVDAC1-N, staple between residues 11 and 15, lower panel), indicating a medium to low μM binding affinity. **f** $^2$H,$^{15}$N-labeled BclxL and **g** specifically Ile, Val, Leu, Ala-$^1$H,$^{13}$C-methyl labeled BclxL show strong CSP effects upon the addition of stVDAC1-N. **h** The CSP effects in (**f**, **g**) cluster to the BH3 binding site of BclxL, suggesting that VDAC1-N is the main interaction site with BclxL.

experiments shown in Fig. 1e, suggesting helix exposure in the smaller nanodisc. In the exposed state, not only the N-terminal helix but also the β-barrel becomes more flexible, as indicated by MD simulations using VDAC1 starting structures with the N-terminal helix inside or outside the β−barrel, respectively (Supplementary Fig. 5a–d, Supplementary Table 2), in agreement with previous simulations with VDAC1 lacking its N-terminal segment[57]. To evaluate whether cryo-EM particle misalignment with helix-inserted VDAC1 can also cause the disappearance of the helix density, we rotationally averaged the density of helix-inserted VDAC1 and obtained an additional ring-like density inside the pore (Supplementary Fig. 5e), which is not observed experimentally.

We also used NMR to monitor the structural state of VDAC1 in 8 or 10 nm MSP nanodiscs and to corroborate the cryo-EM data. First, we recorded 2D-TROSY spectra with $^2$H,$^{15}$N-labeled VDAC1 (Supplementary Fig. 6). Both samples showed an NMR signal dispersion indicative of a folded protein structure. However, while the NMR signals are well resolved in the larger nanodisc (Supplementary Fig. 6a, left), line broadening can be seen in the smaller nanodisc (Supplementary Fig. 6a, right), suggesting more pronounced intrinsic motions in the μs to ms time scale, presumably induced by the tight packing of the nanodisc scaffold protein around VDAC1, as seen in the cryo-EM data (Fig. 2). Interestingly, 2D-[$^{13}$C,$^1$H]-HMQC spectra of specifically $^{13}$CH$_3$-ε methionine labeled VDAC1 (Supplementary Fig. 6b) revealed two distinct peaks for Met155 in the larger nanodiscs (Supplementary Fig. 6c) possibly due to a dynamic equilibrium between the higher populated helix-in and the less populated helix-out states (Supplementary Fig. 6d). A conformational change of the VDAC1 β-barrel upon helix

removal was previously suggested by solid-state NMR data[58]. This conformational equilibrium shifted prominently towards the helix-out state in the smaller nanodiscs (Supplementary Fig. 6c, e).

## BclxL binds to the exposed N-terminal α-helix of VDAC1

Next, we wondered whether the VDAC1-N-terminal helix might be the interaction site of partner proteins, such as BclxL, as previously suggested using VDAC1 peptides[32]. To address this question, we produced VDAC1 in MSP lipid nanodiscs of 8 and 10 nm diameter as described above and performed interaction studies with $^2$H,$^{15}$N-labeled BclxL using 2D-[$^{15}$N,$^1$H]-TROSY NMR experiments (Fig. 3). If the N-terminal binding epitope of VDAC1 is exposed, we expect a change in the NMR spectrum of BclxL. In line with the helix exposure and cryo-EM experiments, VDAC1 in nanodiscs of 8 nm diameter induced strong effects on the BclxL NMR spectrum, such as chemical shift perturbations (CSPs) and line broadening (Fig. 3a). In contrast, VDAC1 in larger nanodiscs (10 nm) resulted in only very weak effects at the concentrations used in the NMR experiment (identical NMR spectra in Fig. 3b). The NMR spectrum of BclxL saturated with a high-affinity BH3 peptide derived from the BH3-only protein PUMA[59] was not affected by VDAC1 addition, even if 8 nm nanodiscs were used (Fig. 3c). These data imply that VDAC1-N binds to the BH3 binding groove of BclxL. This assumption was corroborated by the clustering of the NMR effects shown in Fig. 3a at the BH3 binding groove of BclxL (Fig. 3d), where BH3 peptides interact (e.g., BAK BH3[60] shown in blue in Fig. 3d).

Since the main difference between the investigated VDAC1 samples is the degree of N-terminal α-helix exposure, we next wanted to confirm whether this structural element in VDAC1 is the main

interaction site with BclxL. For this, we first used a VDAC1-N-terminal peptide (residues 1–24), representing the entire N-terminal part, which adopts a random coil structure in solution as probed by CD spectroscopy (Supplementary Fig. 7, left). The binding affinity between this peptide and BclxL was determined with isothermal titration calorimetry (ITC), yielding a $K_D$ value of 29 μM (Fig. 3e, top panel). All BH3 peptides that interact with the binding groove in BclxL adopt an α-helical secondary structure, and the VDAC1-N-terminus is also α-helical when bound to the pore interior. To mimic the bound structure, we used a hydrocarbon-stapled VDAC1 peptide encompassing residues 7–21 (stVDAC1-N). This peptide adopts an α-helical secondary structure, as confirmed by CD spectroscopy (Supplementary Fig. 7, right). It binds more tightly to BclxL ($K_D$ of ~2 μM) than the linear peptide (Fig. 3e, lower panel), which could also be confirmed by NMR titrations (Supplementary Fig. 8). Hydrocarbon-stapled peptides have been previously employed to improve the efficiency of pro-apoptotic Bcl2 protein activation[61]. Thus, we here used the higher-affinity stVDAC1-N for NMR titrations monitored by the backbone amide (Fig. 3f) or side chain methyl group signals (Fig. 3g) in BclxL as probes for binding. Both groups of resonances in BclxL undergo strong CSPs upon peptide binding. Mapping of the CSPs on the structure of BclxL clearly shows that the stVDAC1-N peptide specifically interacts with the BH3 binding groove of BclxL (Fig. 3h), identical to the NMR results obtained with full-length VDAC1 in lipid nanodiscs (Fig. 3d). 2D NMR titration experiments with $^{15}$N-labeled VDAC1-N and unlabeled BclxL confirm the interaction (Supplementary Fig. 9a) and show that the central α-helical part of VDAC-N that has a slight sequence homology to BH3 peptides (Supplementary Fig. 9b) interacts with BclxL (Supplementary Fig. 9c). To probe binding of BclxL to isotope-labeled VDAC1 in nanodiscs, we used a specifically $^{13}$CH$_3$-ε methionine labeled VDAC1 where an additional methionine residue was inserted at position 3 (V3M) to monitor the direct interaction of VDAC1-N with BclxL by 2D-[$^{13}$C,$^1$H]-HMQC experiments. In 8 nm nanodiscs, the ε-methyl resonance of Met3 experienced marked line broadening in the presence of BclxL (Supplementary Fig. 9d). In 10 nm nanodiscs, peak doubling was also observed for Met3, representing the helix-in and helix-out conformational equilibrium. In presence of BclxL, only the Met3 resonance representing the helix-out state is affected, confirming a specific interaction between BclxL and the helix-exposed state of VDAC1 in a full-length context.

## High-resolution structure of BclxL in complex with the VDAC1-N-terminal α-helix

To obtain a higher resolution picture of the VDAC1-N · BclxL interaction, we next screened for suitable protein constructs for X-ray crystallography. To ensure that the VDAC1-N peptide remains stably bound to BclxL, we produced a single-chain construct where VDAC1-N (1–26) is fused to the C-terminus of BclxLΔLT (lacking its flexible loop and the transmembrane helix after residue 209) (Fig. 4a). This construct allowed for crystal formation of suitable quality for a high-resolution structure determination by X-ray diffraction. These data led to the structure determination of BclxLΔLT in complex with the VDAC1-N peptide at 1.95 Å resolution (Fig. 4b, Supplementary Table 3, Supplementary Fig. 10). The parallel orientation of BclxL and the VDAC1-N peptide in the complex structure places the C-termini of both proteins at proximal positions next to the membrane surface (Fig. 4c).

In the full-length protein, the C-terminal end of VDAC1-N is attached to its β-barrel and the C-terminus of BclxLΔTM to its transmembrane helix, respectively. To allow for the formation of the herein determined complex structure, VDAC1 and BclxL must have a suitable orientation in the membrane. The soluble domain of BclxL is known to be located at the cytosolic side of the OMM, like all other Bcl2 proteins[62]. For VDAC1, the orientation in the OMM has been probed by a caspase reporter assay in intact cells[63]. It showed that both N- and C-termini are facing the mitochondrial intermembrane space,

suggesting that the attachment point of VDAC1-N at the β-barrel is on the cytosolic side. Once VDAC1-N becomes detached from the pore interior, it can swing out toward the cytosol—which was also observed in our MD simulations with helix-exposed VDAC1 (Supplementary Fig. 5a). The general binding mode between BclxL and VDAC1-N is reminiscent of BH3 peptides, e.g., as seen in the BclxL-Bak-BH3 complex[60] (Fig. 4d). However, the binding site of VDAC1-N is shifted toward the periphery of the BclxL binding groove yet overlapping with the BH3 site. The C-terminal tail of VDAC1-N loosely binds to BclxL in an extended conformation, which is not seen for BH3 peptides. In our X-ray structure, VDAC1-N forms an amphipathic helix with the side chains of Leu10, Ser13, Asp16, Val17, and Phe18 facing the BclxL binding groove (Fig. 4e).

To validate the structural model of the complex, we performed alanine scanning experiments where every individual residue in the binding region of VDAC1-N was mutated to alanine (Supplementary Table 4), followed by NMR-detected affinity measurements with each peptide (Fig. 4f, Supplementary Fig. 11). These experiments showed a strong impact of residues that participate in binding to BclxL, with the L10A mutation having the biggest impact (~20-fold reduction in affinity). Leu10 is the only large hydrophobic residue in VDAC1-N that is establishing a hydrophobic interaction with the BH3 binding groove in BclxL, thus markedly contributing to the overall binding affinity. Leu10 is also one of the main interaction sites between VDAC1-N and the β-barrel. Consequently, its mutation to cysteine weakens the helix attachment and enhances its exposure (Supplementary Fig. 1b–e).

With the structural data at hand, we assembled a structural model of the full-length VDAC1-BclxL complex in silico by attaching the experimental BclxL-VDAC1-N structure to the VDAC1 β-barrel. Structural clashes and unfavorable backbone angles in the assembled structure were resolved in a 200 ns molecular dynamics simulation. In the resulting structural model (Fig. 4g), the dissociated VDAC1-N is located slightly outside the β-barrel, where it can interact with the BH3 binding groove of BclxL. A recent NMR study of full-length BclxL in lipid nanodiscs showed that the BH3 binding groove of the soluble domain of BclxL is facing toward the membrane surface[64], which resembles the orientation of BclxL in the structural model of the complex with VDAC1 shown here. Furthermore, the exposure of the N-terminal helix, as seen in the complex with BclxL that results in a larger VDAC1 inner pore diameter, can explain the observed increase in the VDAC1 open configuration if bound to BclxL as monitored by electrophysiology[65].

## The VDAC1-N-terminus acts as a sensitizer BH3 protein to inhibit anti-apoptotic BclxL

Next, we aimed at addressing the functional relevance of the VDAC1-BclxL interaction in the induction of apoptosis via mitochondrial outer membrane permeabilization (MOMP). In recent literature reports, it has been postulated that VDAC1 oligomers alone can form large pores to enable MOMP[20] and even allow for the exit of mitochondrial DNA[21]. To address this question, we conducted in vitro experiments with VDAC1 in liposomes loaded with cytochrome C or lysozyme that are both of similar size (12 and 14 kDa, respectively) (Supplementary Fig. 12a). VDAC1 proteoliposomes and empty liposomes were injected on a size-exclusion chromatography column, and the protein content inside the liposomes was assayed by their corresponding SDS-PAGE band intensity. For both proteins, no translocation across VDAC1 could be detected (Supplementary Fig. 12b, c). However, the flux of ATP across VDAC1 in proteoliposomes was possible in the same samples (Supplementary Fig. 12d), indicating that VDAC1 was functional but did not form a larger pore in liposomes that would allow for the translocation of proteins.

Thus, we wondered whether VDAC1 could have an indirect pore-forming activity via its interaction with Bcl2 proteins. To probe the effect of VDAC1-N exposure on the homeostasis of the Bcl2 protein

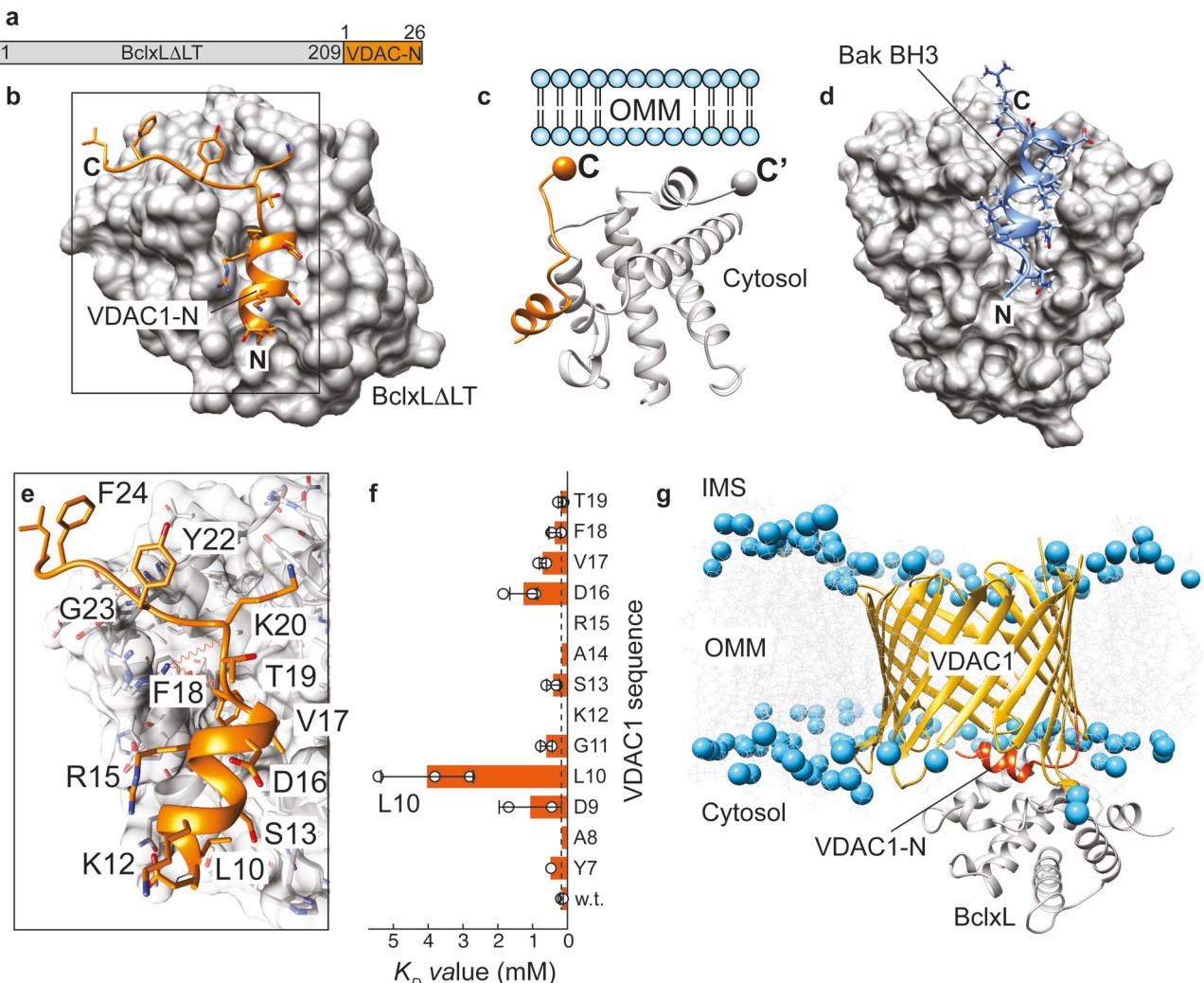

**Fig. 4 | Complex structure of BclxL and VDAC1-N. a** Single-chain construct to enable structure determination of the VDAC1-N-BclxL complex. **b** Structure of BclxLΔLT in complex with VDAC1-N. **c** Schematic view of the C-termini of VDAC1-N and BclxL, both oriented in the same direction. **d** Complex structure of BclxL and the BH3 domain of Bak[60]. **e** Close-up view of the interface between BclxL and VDAC1-N in the complex. Amino acids in VDAC1-N participating in the interaction are labeled. **f** Affinity ($K_D$ values) of VDAC1 peptides for BclxL in an alanine scanning experiment. The substitution of Leu10 to Ala, facing the BclxL binding groove, shows the most pronounced effect. Dashed line: affinity of wild-type VDAC1-N. Missing values for K12A and R15A are due to poor solubility in aqueous buffer. Data are presented as mean values ± SD calculated from three individual NMR resonances as probes (n = 3, circles). **g** Structural model of the VDAC1- BclxL complex based on the structural data reported in this study. VDAC1-N is shown in orange, and lipid phosphate groups are represented as blue spheres.

system, we performed established liposome pore-forming assays[66] with a defined set of pro- and anti-apoptotic Bcl2 proteins: pore-forming Bak, the activator BH3-only protein cBid, and the inhibitor BclxL[67–69]. At low concentrations, Bak alone does not form pores (Supplementary Fig. 13f) but requires activation by cBid (red curve in Fig. 5a), whereas stoichiometric concentrations of BclxL fully inhibit pore formation (black curve in Fig. 5a). To investigate the effect of VDAC1-N, we added increasing concentrations of stVDAC1-N into BclxL-inhibited Bak and observed a concentration-dependent rescue of the pore-forming activity of Bak. At higher protein concentrations, Bak forms pores without activation by BH3-only proteins (autoactivation conditions), which can be partially inhibited by BclxL (Supplementary Fig. 13a). Again, the addition of the stVDAC1-N peptide leads to complex dissociation via competitive binding to BclxL and consequently to a concentration-dependent increase in the pore-forming activity of Bak (Supplementary Fig. 13a, b) to the level of Bak without BclxL (red curve in Supplementary Fig. 13a).

To exclude a direct activating effect of VDAC1-N on Bak, we tested whether VDAC1-N can induce Bak pore formation by itself, like the known Bak activator cBid[69]. The results in Fig. 5a and Supplementary

Fig. 14 clearly show that even a large excess of VDAC1-N (70 µM) does not lead to Bak activation (50 nM concentration), whereas cBid can already fully activate Bak at a sub-stoichiometric concentration of 40 nM. Thus, it is very likely that VDAC1-N rather promotes Bak pore formation via neutralizing the inhibitor BclxL. This behavior is reminiscent of sensitizer BH3 proteins, such as BAD or NOXA, that exhibit pro-apoptotic activity by neutralizing the anti-apoptotic members[35]. As a prototype sensitizer BH3, we used a BAD-BH3 peptide in this assay and could show that it has a similar effect as VDAC1-N, but due to its low nanomolar affinity, only a concentration of 50 nM is required for full recovery of the Bak pore-forming activity (Supplementary Fig. 13c, d). As expected, BAD-BH3 alone was not capable of activating Bak (Supplementary Fig. 14b).

Human VDAC exists in three isoforms (Supplementary Fig. 13e), where VDAC1 and VDAC3 are almost identical and only VDAC2-N has an N-terminal extension of 11 amino acids. To test whether VDAC2 has direct activation properties for Bak, we performed the Bak pore-forming assay with both peptides derived from VDAC1-N and VDAC2-N. However, none of these peptides showed direct activating properties (Supplementary Fig. 13f) at concentrations of up to 10 µM. Finally,

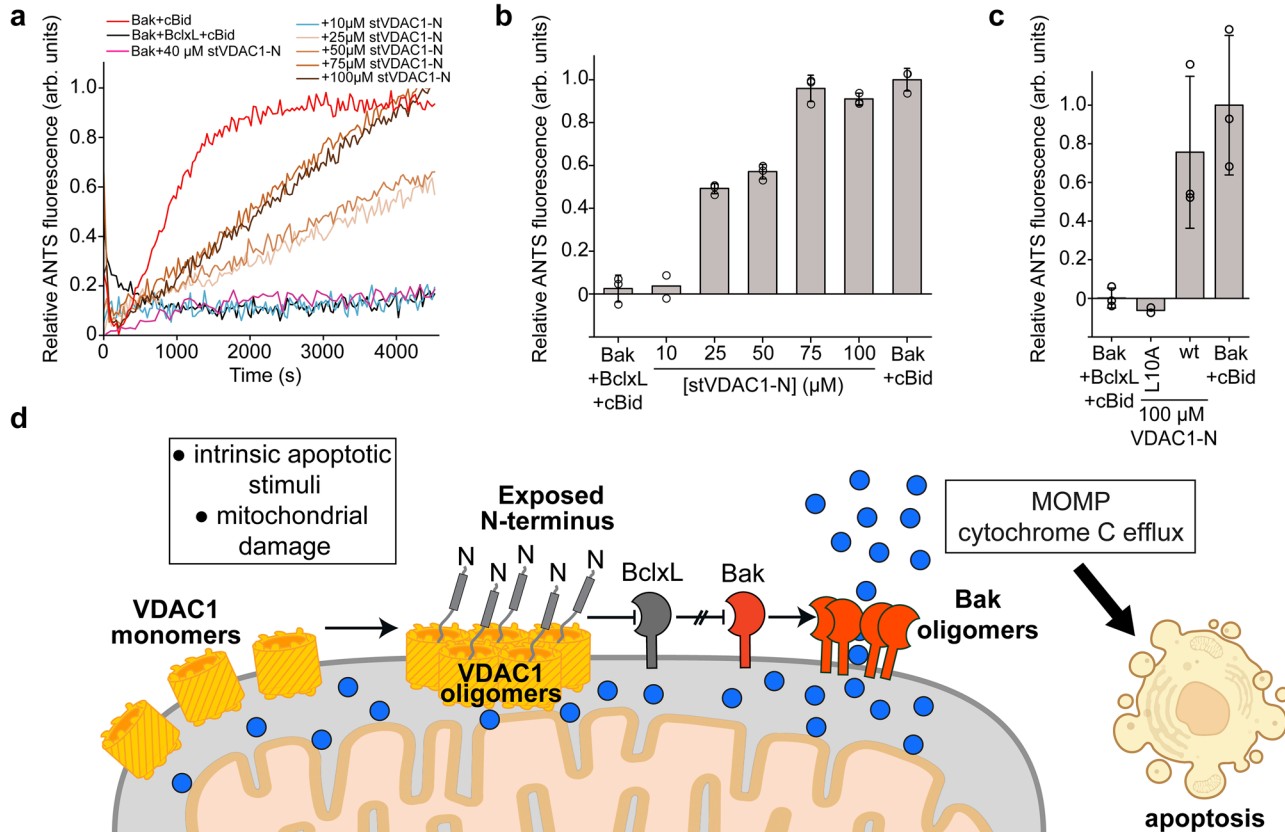

**Fig. 5 | VDAC1-N acts as a sensitizer BH3 protein to induce Bak-mediated membrane permeabilization. a** Pore formation of BakΔTM (50 nM concentration, red) can be activated by cBid (40 nM) can be inhibited by the addition of BclxLΔTM (50 nM, black). The addition of stVDAC1-N (10–100 μM) induces Bak pore formation in the presence of BclxL in a concentration-dependent manner (sand to brown colors). In contrast to cBid, the addition of stVDAC1-N, even at a 40 μM concentration, cannot directly activate pore formation of Bak. **b** Fluorescence intensity taken from (**a**) averaged between 3500 and 4000 s. Data are presented as mean values ± SD calculated from technical replicates (n = 3, circles). **c** The effect of 100 μM of linear VDAC1-N on Bak pore formation can be abolished with a L10A mutation, consistent with its 20-fold lower binding affinity to BclxL (Fig. 4f). Data are presented as mean values ± SD calculated from technical replicates (n = 3, circles). **d** Functional model: apoptotic stimuli or mitochondrial damage lead to VDAC1 oligomerization, inducing the exposure of its N-terminal α-helix, which can neutralize anti-apoptotic Bcl2-like proteins, such as BclxL, to induce the execution of MOMP and cytochrome C release via pore-forming Bcl2 proteins, such as Bak. Blue spheres: cytochrome C.

we validated the specific binding of VDAC1-N to BclxL in the pore-forming assay. Since mutated VDAC1-N-L10A was shown to have a ~20-fold lower binding affinity to BclxL (Fig. 4f), we monitored the activity of this peptide in the pore-forming assay. The wt-VDAC1-N peptide had a strong rescuing effect at 100 μM concentration to approximately 80% of the value obtained without inhibition by BclxL (Fig. 5c, Supplementary Fig. 13g), whereas the L10A variant was completely inactive at the same concentration, providing a clear correlation between binding affinity and activity in the pore-forming assay.

## Discussion

In this study, we show that VDAC1 can expose its N-terminal α-helix (VDAC1-N) to the pore exterior. In a cellular environment, the VDAC1-N-terminal segment is facing the cytosolic side of the mitochondrial outer membrane, which is a prerequisite to interact with anti-apoptotic Bcl2 proteins such as BclxL. We here present a large set of biophysical and structural data that confirms a specific interaction between VDAC1-N and BclxL. Via its interaction with BclxL, VDAC1 acts as a so-called sensitizer BH3 protein, a class of pro-apoptotic Bcl2 proteins that inhibit the anti-apoptotic Bcl2 family members and eventually induce the mitochondrial apoptosis pathway[35] (Fig. 5d). In our liposome pore-forming assays (Fig. 5a–c), we see that the BclxL-mediated inhibition of the pore-forming activity of the Bcl2 protein Bak is abolished in a dose-dependent manner upon the addition of VDAC1-N. In this context, the exposure of the VDAC1-N-terminal helix is required

to execute its pro-apoptotic functionality. In agreement with this model, cells expressing VDAC1, where the N-terminal helix was deleted (Δ26-VDAC1), were resistant to apoptosis induced by various stimuli[33].

Under non-apoptotic conditions, the N-terminal helix is most likely attached to the pore interior, as evident from the available structures of VDAC1[6–8] alone or in complex with the ER membrane protein complex (EMC)[70]. Furthermore, free energy calculations with the canonical VDAC1 structure indicate that the helix-inserted state is strongly favored[48]. Therefore, a trigger is most likely necessary to stimulate a structural change within VDAC1 that enables helix release. Our data suggest that VDAC1 oligomerization and the release of the N-terminal helix are connected (Fig. 1). VDAC1 oligomers[19,22] have been reported to be induced by diverse stimuli, such as VDAC1 up-regulation[71], changes in the mitochondrial lipid composition[23,24], increased Ca[2+] levels[25,26], low pH[27], oxidative stress[21,28], or specific small molecules that inhibit the respiratory chain[72], and cholesterol[73]. VDAC is highly abundant in the mitochondrial outer membrane (MOM)[74] and has been shown to form dynamic supramolecular assemblies of various sizes ranging from monomers up to 20-mers[19,73,75]. Thus, slight changes in the MOM environment and in the VDAC protein levels have the potential to strongly impact the VDAC oligomeric state. Even though the mechanistic details leading to tight VDAC oligomerization remain unclear, there is culminating evidence that VDAC oligomerization is a marker for apoptosis in human cells[15,20,33]. Thus, VDAC might be an internal checkpoint for the functional state of

mitochondria, which contrasts with the action of pro-apoptotic Bcl2 proteins, where apoptosis signals originate from outside the cell, the cytoplasm, or other organelles[76]. Due to its increasing significance in apoptosis induction, VDAC is also considered an emerging drug target in cancer therapy[74,77].

We here observed strong VDAC1 oligomerization in the negatively charged detergent cholate but not in the structurally very similar zwitterionic detergent CHAPS or the zwitterionic detergent LDAO (Fig. 1b). In these experiments, the degree of oligomerization highly correlated with the exposure of the N-terminal helix as probed by chemical modification experiments with PM40 (Fig. 1c). In line with these observations, enhanced helix exposure was also detected in liposomes containing negatively charged lipids and the cholesterol analog CHS (Supplementary Fig. 1d–g). These findings are consistent with previous data showing an interaction between VDAC1-N and negatively charged lipids[46–48] or detergents[78], as well as the impact of CHS on VDAC1 oligomerization[73,79].

In lipid nanodiscs, the exposure of the N-terminal helix was more pronounced in the smaller (8 nm) nanodiscs[49,50] than in the larger (10 nm) nanodiscs (Fig. 1d, e). Structural data obtained by cryo-EM demonstrate that only a small portion of the VDAC1-cMSP1D1ΔH5 particles are seen in the canonical, helix-inserted state, whereas this is the case for almost all particles with VDAC1 in the larger nanodiscs (Fig. 2, Supplementary Fig. 3).

Since the "tight" VDAC1 state in cMSP1D1ΔH5 nanodiscs is stabilized by protein-protein interactions with the MSP instead of a lipid environment, we consider it plausible that this setup mimics the oligomeric state where protein-protein interactions between VDAC1 protomers dominate[19], especially concerning the exposure of VDAC1-N. This assessment is further supported by the very similar outcome of the PM40-modification experiments in detergent micelles, liposomes and small lipid nanodiscs (Fig. 1, Supplementary Fig. 1). Furthermore, our nanodisc setup shows that VDAC1 can adopt a helix-exposed structural state with the β-barrel fully intact (Fig. 2) which is compactly folded (Supplementary Fig. 2f) and capable of binding to the partner protein BclxL (Fig. 3). Thus, the chosen nanodisc strategy most likely represents a functionally relevant setup, which was crucial to obtain further structural information by cryo-EM and NMR. Nanodiscs have been previously used to selectively stabilize functional structural states of membrane proteins[80,81]. By using circularized MSPs[41,54], we obtained stable VDAC1 preparations in nanodiscs of different sizes with optimized homogeneity (Supplementary Fig. 3). This setup facilitated the structural characterization of both states by single-particle cryo-EM. Despite recent progress in cryo-EM, membrane proteins such as VDAC1 with a molecular weight of just 31 kDa without a large protein mass outside the membrane are still very challenging[82]. Nonetheless, we achieved a resolution range of 5.7–7.2 Å, sufficient to detect the location of VDAC1-N in the helix-inserted canonical state in 11 nm nanodiscs and its absence in the exposed state in the smaller nanodiscs. Even though a cryo-EM resolution of 6–7 Å leaves some degree of uncertainty, it is most likely that VDAC1-N in the smaller nanodisc is either extruded from the pore or substantially more mobile than in the canonical structures. In addition, the strong variation of noise surrounding the particle suggests that the exposed N-terminal helix is tumbling outside the pore in a dynamic manner as seen in our MD simulations (Supplementary Fig. 5). Furthermore, we identified a dimeric VDAC1 state in 11 nm nanodiscs (Fig. 2) with the dimerization interface located at that side of the β-barrel wall where the N-terminal helix is attached, resembling a previously published crystal structure[55]. The decent resolution in our cryo-EM structure was likely enabled by the specific interaction between VDAC1 and the MSP forming the rim of the nanodisc particles. Interactions of the MSP with membrane proteins in cryo-EM structures have been previously analyzed[56] and the interaction with the MSP also enabled the design of water-soluble MSP-membrane protein fusions in *E. coli*[83].

Although controversially discussed in the literature[16], previous studies have suggested the direct involvement of VDAC1 in forming a large pore in the outer mitochondrial membrane[14,21,29]. Even though our data does not support a direct role of VDAC1 in forming a large-enough pore for the release of pro-apoptotic proteins, we observed in our single-particle cryo-EM analysis a minor species (15%) with a larger pore diameter of up to 6 nm (Supplementary Fig. 3f), which was excluded from further analysis due to heterogeneity. Even though the relevance and nature of a larger VDAC1 species is unclear, we can conclude that such larger VDAC1 pores can be formed under conditions where the β-barrel is destabilized.

Our data suggest that VDAC1 oligomerization triggers VDAC1-N dissociation from the interior surface of the β-barrel. Once presented at the MOM, VDAC1-N binds to BclxL to inhibit its anti-apoptotic activity. The rather low µM affinity between VDAC1-N and BclxL is compensated by the high abundance of VDAC1 in the MOM[74]. Looking at apoptosis induction mechanisms at the mitochondrial surface, the action of VDAC1-N is reminiscent of sensitizer BH3 proteins with pro-apoptotic activity[35], rendering helix-exposed VDAC1 an indirect inducer of cytochrome C release and finally apoptosis (Fig. 5d). In a more general context, our study suggests a regulation mechanism involving β-barrel membrane proteins where the plasticity of the β-barrel enables the occupation of different functional states. Such a mode-of-action can be modulated by factors like oligomerization or partner protein and metabolite binding and offers an additional layer of regulation, as shown recently for a chloroplast metabolite channel[84]. Nevertheless, a more global investigation on the interaction of VDAC1 with partner proteins is necessary to estimate the broader role of VDAC1 and its isoforms in apoptosis induction.

## Methods

### Molecular cloning and protein production

Human BclxL, where the transmembrane helix (TMH) alone or the TMH and the flexible loop (residues 44–84) were deleted, BclxLΔTM or BclxLΔLT[22,85], respectively, and human VDAC1[22] in pET21a plasmid (Merck) were used for protein production. Site-directed mutagenesis of BclxL or VDAC1 was done with the Quikchange Lightning Kit (Agilent). The sequences of all DNA primers used in this study are shown in Supplementary Table 5.

VDAC1 and its variants were produced, refolded, and purified as described[7]. The expression was performed in *E. coli* BL21(DE3) by induction with 1 mM IPTG for 3–4 h at 37 °C. After cell lysis and centrifugation for 20 min at 40,000 × *g*, the pellet was resuspended in lysis buffer (50 mM Tris, pH 7.0, 250 mM NaCl, 20 mM BME) + 1% TritonX-100. The mixture was then subjected to centrifugation at 40,000 g. Subsequently, resuspension and centrifugation were repeated two more times with lysis buffer without TritonX-100. The pellet was then solubilized in IMAC buffer (6 M GdmCl, 50 mM Tris, pH 8.0, 100 mM NaCl) overnight. After another round of centrifugation, the supernatant was applied to a Ni-NTA gravity follow column (Cytiva). The column was washed with IMAC buffer, and the protein then eluted using IMAC buffer + 500 mM imidazole. The protein was then dialyzed against 20 mM Tris, pH 7.5, 50 mM NaCl, 1 mM EDTA, and 10 mM BME. The precipitated protein was solubilized in 6 M GdmCl, 100 mM NaPi, pH 7.0, 100 mM NaCl, 1 mM EDTA, and 5 mM DTT, and the concentration adjusted to 5 mg/mL. VDAC1 wild-type and variants were refolded at 4 °C by dropwise addition of the protein into a 10x volumetric excess of refolding buffer (25 mM NaPi, pH 7.0, 100 mM NaCl, 1 mM EDTA, 5 mM DTT, and 1% LDAO (Avanti Polar Lipids)). The mixture was then stirred for 6–8 h and then dialyzed overnight against 10 mM NaPi, pH 6.5, 1 mM EDTA, 100 mM NaCl, and 3 mM DTT. Subsequently, size-exclusion chromatography was conducted by applying the protein to a HiLoad™ 16/600 Superdex™ 200 pg column (Cytiva). The column was equilibrated in 10 mM NaPi, pH 6.5, 100 mM NaCl, 1 mM EDTA, 5 mM DTT, and 0.1% LDAO (for the proteins used in BS³

and PM40 modification experiments, 20 mM HEPES-NaOH, pH 7.5, 50 mM NaCl, 0.5 mM EDTA, and 0.1% LDAO was used instead). Monomeric protein was pooled and concentrated to >300 μM.

To produce $^2$H,$^{15}$N-ε- CH$_3$ -$^{13}$C-methionine-VDAC1-V3M, protein expression was conducted in M9 medium containing 99% D$_2$O, 1 g/L $^{15}$N-NH$_4$Cl and 2 g/L $^2$H, $^{12}$C-glucose. 70 mg/L L-methionine (ε-methyl-$^{13}$C) (Sigma-Aldrich) was added to the M9 expression medium 1 h before induction of protein expression with 1 mM IPTG. After 4 h of shaking at 37 °C, cells were harvested by centrifugation. $^2$H,$^{15}$N-CH$_3$ε-$^{13}$C-methionine-VDAC1-V3M was purified as described for wild-type VDAC1[7,22,86]. For the production of $^2$H,$^{15}$N-labeled VDAC1, the addition of methionine (ε-methyl-$^{13}$C) was omitted.

Plasmids encoding for cBid and the soluble domains of Bak (BakΔTM, residues 1–186) and BclxL (BclxLΔTM, residues 1–212, BclxLΔLT, residues 1–43, 85–212, and BclxLΔLT fused to VDAC1-N: BclxLΔLT-LEGG-VDAC1-N(1–26)-GG-His$_6$) were expressed in *E. coli* as previously described[64,68,69,85]. In brief, *E. coli* BL21 (DE3) cells were transformed with the respective plasmids and grown at 37 °C. For BakΔTM, protein expression was induced with 0.1 mM IPTG and grown for 16–20 h at 20 °C, whereas for cBid and BclxL, protein expression was induced with 1 mM IPTG and grown for 4 h at 37 °C. In the resulting cBid protein, a thrombin cleavage site is replacing the original caspase-8 site[68]. Purification of cBid, BakΔTM, and BclxL was performed as previously described[68]. Briefly, cell pellets were resuspended in lysis buffer (50 mM Tris, pH 8, 200 mM NaCl, 1 mM EDTA) supplemented with 2 mM PMSF and 1 mg/mL lysozyme. The suspension was then subjected to sonication for 30 min (1-s pulse, 3-s pause, 30% amplitude), incubated together with DNase I and 5 mM MgCl$_2$, and the cell debris was removed via centrifugation. The supernatant was applied to a Ni-NTA gravity flow column equilibrated with buffer A (20 mM Tris, pH 8.0, 100 mM NaCl, 5 mM BME), washed with 20 CV buffer A, 20 CV buffer A supplemented with 10 mM imidazole, followed by elution with 20 CV buffer A supplemented with 400 mM imidazole. The elution fraction was dialyzed into buffer A overnight, and for cBid, 20 U thrombin/L cell pellet was added. 5% glycerol was added to protein solutions before concentrating for size-exclusion chromatography (SEC) on an ÄKTA Pure system equipped with a HiLoad 16/600 Superdex 75 pg column equilibrated with 20 mM NaPi, pH 7.0, 50 mM NaCl, 0.1 mM EDTA, and 5 mM BME.

Production of isotope-labeled BclxLΔTM or BclxLΔLT was done in *E. coli* BL21 (DE3). Cells were grown in M9 medium containing 99% D$_2$O, 1 g/L $^{15}$N-NH$_4$Cl and 2 g/L $^2$H, $^{12}$C-glucose (for $^2$H,$^{15}$N-labeling) and additionally 300 mg/L stereospecific LV precursor ethyl-2-hydroxy 2-$^{13}$C-methyl 3-oxobutanoate[87,88] and 80 mg/L of the Ile precursor $^{13}$C-methyl-α-ketobutyrate[89] (Merck) together with 2.5 g/L $d_4$-succinate (Eurisotope, Merck) and 0.8 g/L 3-[$^{13}$CH$_3$]-2-D-Ala (Eurisotope, Merck), added to the bacterial culture approximately 1 h before induction with 1 mM IPTG. The choice of these precursors and reagents resulted in $^{13}$CH$_3$ labeling of the *pro-S* methyl group in Val/Leu, Ile δ$_1$, and Ala β-methyl groups in an otherwise per-deuterated $^{12}$C background.

Constructs encoding GB1-VDAC peptides were cloned into an in-house modified pET vector encoding GB1 with an N-terminal His$_6$ tag and TEV site, followed by a thrombin site (His$_6$-TEV-GB1-thrombin-peptide). We produced the following constructs: GB1-VDAC1-N (residues 2–25) and GB1-VDAC2-N (residues 2–36). Expression and purification of the GB1-VDAC peptides were described previously[48]. Following SEC, the GB1-VDAC peptides were digested with TEV (1:40 molar ratio) in the presence of 5 mM BME overnight at 4 °C. Reverse Ni-NTA was performed to isolate the cleaved peptides, which eluted in the flow-through and wash steps. Cleavage of the GB1-VDAC peptides was confirmed by SDS-PAGE analysis.

Synthetic stapled VDAC1 peptides were purchased from Bio-Synthesis and Biomatik. The isotope-labeled ($^{13}$C and $^{15}$N) VDAC1 peptide (residues 1–26) was produced recombinantly in *E. coli* as a fusion construct with GB1 (His$_6$-GB1-TEV-VDAC1-N (1–26))[48]. Cells were grown in minimal media supplemented with 1 g/L $^{15}$NH$_4$Cl and 2 g/L $^{13}$C$_6$-glucose until an OD$_{600}$ of 0.7 at 37 °C. Protein production was induced by the addition of 1 mM IPTG, and cells were shaken for further 4 h at 37 °C, followed by harvesting by centrifugation. Protein purification was done by Ni-NTA chromatography, followed by digestion with TEV protease and reverse Ni-NTA chromatography to collect the VDAC1-N peptide in the flow-through and wash fractions, which were dialyzed against H$_2$O, lyophilized, and dissolved in NMR buffer (20 mM NaPi, pH 7.0, 50 mM NaCl, 0.5 mM EDTA, 5 mM DTT) for further analysis.

## Lipid nanodisc assembly
Assembly of VDAC1-containing lipid nanodiscs was done according to established protocols[40,49,50,90]. VDAC1 and VDAC1-V3M nanodiscs with MSP1D1ΔH5, MSP1D1, cMSP1D1ΔH5 and cMSP1D1 were assembled with a VDAC1:MSP:lipid ratio of 1:6:40, 1:6:60, 1:6:55 and 1:6:80, respectively. The lipid mixture consisted of 75% DMPC and 25% DMPG. Respective amounts of MSP buffer (20 mM Tris pH 7.5, 100 mM NaCl, 0.5 mM EDTA), lipids in 100 mM sodium cholate, MSP, and VDAC1 or VDAC1-V3M in LDAO were mixed and incubated for 1 h at room temperature (RT). Subsequently, the mixture was incubated for 1.5 h at RT with 0.5 g washed Bio-beads SM-2 (Bio-Rad) per mL of reaction, while rocking. The beads were then removed by filtration and washed with buffer. The assembled nanodiscs were then subjected to Ni-NTA and a HiLoad 16/600 Superdex 200 pg column (Cytiva) on an Äkta Purifier system controlled by Unicorn 7 software. For the preparation of nanodiscs for cryo-EM, the size-exclusion chromatography buffer was 20 mM HEPES, pH 7.0, 100 mM NaCl, 0.5 mM EDTA, and 2 mM DTT.

## Chemical crosslinking and modification of VDAC1
Protein crosslinking in detergent micelles was performed in 20 mM HEPES-NaOH, pH 7.5, 50 mM NaCl, with a final VDAC1 concentration of 20 μM in the presence of 20 mM of different detergents by diluting VDAC1 stock at least 15×. After a 30 min preincubation at room temperature, 40 times molar excess of the amino-selective crosslinker BS$^3$ (ThermoFisher Scientific, Cat. No. 21580) was applied for 30 min, as reported previously[84]. The reaction was stopped by the addition of 50 mM (final) Tris/HCl pH 7.5 and further incubation for 15 min. For polyethyleneglycol (40 kDa) maleimide (PM40) modification, 20 μM PM40 (Sigma-Aldrich) was applied for 15 min, and the reaction was stopped by the addition of a large excess of β-mercaptoethanol. Finally, the samples were analyzed by SDS-PAGE and Coomassie staining.

Nanodisc samples were prepared as described above. 20 μM PM40 was applied directly after detergent removal to 20 μM VDAC in nanodiscs as indicated above. For the preparation of liposomes, 10-15 μM VDAC1 mixed with 5 mg/ml lipids (100% POPC or 50% POPC + 50% POPG or 40% POPC + 30% POPE + 10% POPG + 10% DOPS + 10% Cardiolipin to mimic mitochondrial OM) solubilized with 0.5% LDAO and 0.1% TX100. After 30 min preincubation, detergents were removed using 2 rounds of Bio-Beads SM-2 (200 mg/ml at each round). 40× BS$^3$ or 20 μM PM40 was applied on liposome samples, and the samples were 2× flash frozen and thawed on a thermoblock at room temperature. Following 30 min of reaction time, the samples were quenched as above and finally analyzed by SDS-PAGE and Coomassie or silver staining.

## Liposome translocation assay
10 μM VDAC1 was mixed with 10 mg/mL Egg PC lipids solubilized with 0.5% LDAO and 0.1% Triton X-100 in 20 mM HEPES-NaOH, pH 7.5, 100 mM NaCl. After a 30 min preincubation, detergents were removed using 3 rounds of Bio-Beads SM-2 (2 × 100 mg/mL and 1 × 200 mg/mL). Molecules (2 mM ATP, 1.5 mg/mL cytochrome C or lysozyme, 1 mg/mL BSA) to be transported were added into liposomes, and three rounds of freeze-thaw cycles were applied. Then, the liposomes were passed through 0.2 μm filter 15 times in a mini extruder (Avanti Polar Lipids)

and were injected into a Superose 6 Increase 10/300 GL (Cytiva). Finally, the peak fractions were analyzed using SDS-PAGE and silver staining. ATP was detected by chemiluminescence with an ATP determination kit (Promega, cat# A22066) following the instructions of the manufacturer.

## Liposome permeabilization assay

The liposome permeabilization assays were performed at 30 °C on a SpectraMax iD5 multimode plate reader (Molecular Devices) controlled by the SoftMax Pro 7 software as previously described[66,91]. Briefly, 40–50 nM cBid, 50 nM BclxLΔTM, and 1–100 μM VDAC-N peptides were used for measurements at low BakΔTM concentrations (50–200 nM), whereas 1 μM BakΔTM and 500 nM BclxLΔTM were used for measurements under auto-active BakΔTM conditions. VDAC1-N peptide was titrated into the Bak-BclxL complex from 10 to 100 μM under cBid-activated conditions and from 25 to 100 μM under auto-active BakΔTM conditions. For comparison of VDAC1-N WT peptide to VDAC1-N L10A (refer to Supplementary Table 4 for peptide sequences), 100 μM of either VDAC1-N peptide was added into the Bak-BclxL complex under cBid-activated conditions. As a positive control, a peptide spanning the BH3 domain (residues 140–165) of the neutralizing BH3-only protein Bad (Bad-BH3) was used, which is known to bind BclxL with low nanomolar affinity[92]. For this experiment, Bad-BH3 peptide was titrated into the Bak-BclxL complex from 1 to 100 nM under cBid-activated conditions. The liposomes were prepared with the following lipid composition (mass percentage) to mimic the outer mitochondrial membrane (OMM), as previously described[66]. Briefly, the OMM-like lipids were prepared by combining 38% L-α-phosphatidylcholine (PC), 25% L-α-phosphatidylethanolamine (PE), 10% L-α-phosphatidylinositol (PI), 10% 1,2-dioleyl-sn-glycero-3-phospho-L-serine (DOPS), 7% 1,1′,2,2′-tetra-(9Z-octadecenoyl)cardiolipin, and 10% 18:1 DGS-NTA (Ni$^{2+}$). For the liposome permeabilization assay at low BakΔTM concentrations (50–200 nM), 2–5% 18:1 DGS-NTA (Ni$^{2+}$) was used, and the mass of PC was adjusted to compensate. The lipids (5 mg) were mixed in chloroform and dried under nitrogen gas flow, resuspended in 0.25 mL of assay buffer (10 mM HEPES, pH 7.0, 200 mM KCl, 5 mM MgCl$_2$), sonicated in a sonication bath until homogenous, subjected to five freeze-thaw cycles, then extruded using a 100 nm polycarbonate membrane. The liposomes were prepared with the addition of the polyanionic dye 8-aminoaphthalene-1,3,6-trisulfonic acid (ANTS) and the cationic quencher *p*-xylene-bis-pyridinium bromide (DPX) as described[66,91]. For experiments at low BakΔTM concentrations (50–200 nM), 0.05–0.15 mg/mL OMM-like lipids doped with 2–5% (w/w) 18:1 DGS-NTA (Ni$^{2+}$) were used. For experiments under BakΔTM auto-active conditions, 0.3 mg/mL OMM-like lipids doped with 10% (w/w) 18:1 DGS-NTA (Ni$^{2+}$) were used. All resulting data were averaged from three technical replicates unless a clear outlier due to experimental artifact was removed. Data analysis was performed by first normalizing to the minimum value in each dataset, then calculating the mean for the technical replicates. The data represented in each of the figures was scaled such that 0 represents the BakΔTM-BclxLΔTM complex and 1 represents either auto-activated BakΔTM or cBid-activated BakΔTM. The data are represented as relative ANTS fluorescence (%).

## Peptide synthesis

Alanine-scanning peptides (Supplementary Table 4) were prepared according to the standard Fmoc-Solid Phase Peptide Synthesis (Fmoc-SPPS) using a tritylchloride polystyrene (TCP) resin[93]. The completion of the coupling reactions was controlled by analytical ESI-MS. If necessary, the coupling reactions were repeated, especially when Fmoc-Asp(O$^t$Bu)-OH was coupled to the H-Leu-rest. In the following, acid-labile groups were used for protection of side chains of amino acids: Pbf for Arginine; *t*Bu for Threonine, Serine, Aspartic acid, and Tyrosine; Boc for Lysine. They were removed using a mixture of

trifluoroacetic acid (TFA)/DCM/triisopropylsilane (TIPS)/water (80:10:5:5) for 1.5 h at RT. The resulting peptides were purified by semi-preparative HPLC, and their purity was confirmed by analytical HPLC-ESI-MS.

## HPLC and mass spectrometry

Analytical HPLC-ESI-MS was performed on a Hewlett-Packard Series HP 1100 equipped with a Finnigan LCQ mass spectrometer using a YMC-Hydrosphere C18 column (12 nm pore size, 3 μm particle size, 125 × 2.1 mm) or YMC-Octyl C8 column (20 nm pore size, 5 μm particle size, 250 × 2.1 mm) and H$_2$O (0.1% *v/v* formic acid)/MeCN (0.1% *v/v* formic acid) as eluents. Semi-preparative HPLC was performed using a Beckmann instrument (System Gold, solvent delivery module 126, UV detector 166), a YMC ODS-A column (20 × 250 mm, 5 μm), flow rate: 8 mL/min, linear gradients of H$_2$O (0.1% *v/v* TFA) and MeCN (0.1% *v/v* TFA).

## Circular dichroism (CD) spectroscopy

CD spectra and thermal transitions were measured with a Jasco J-1500 spectropolarimeter controlled by the Spectra Manager software. Secondary structure estimation based on CD spectra was done with the BestSel server[94]. Thermal melting experiments were conducted at a heating rate of 1.5 °C/min and at a wavelength of 220 nm (monitoring the α-helical content of the protein) or 215 and 217 nm (for β-sheet content) and analyzed with a Boltzmann equation[95].

## Isothermal titration calorimetry (ITC)

ITC experiments to characterize the affinity between BclxLΔLT and VDAC1 peptides were conducted with a MicroCal PEAQ-ITC (Malvern Panalytical) at 20 °C and analyzed by the MicroCal PEAQ-ITC Analysis Software v1.41. For the linear VDAC1-N peptide, 250 μM of BclxLΔLT in 10 mM NaPi, pH 7.0, 50 mM NaCl, 0.5 mM EDTA, and 1 mM DTT in the cell was titrated with 2.5 VDAC1 peptide in the syringe using 20 2 μL injections. For the stabled peptide, 50 μM BclxLΔLT and 500 μM peptide were used. Data were analyzed by the MicroCal PEAQ-ITC Analysis Software v1.41.

## Crystallization, diffraction data collection, and processing

The crystallization experiments with the single-chain BclxLΔLT-VDAC1-N construct were performed at the X-ray Crystallography Platform at Helmholtz Munich. The initial crystallization screening was done at 292 K using 10 mg/mL of protein with a nanodrop dispenser in sitting-drop 96-well plates and commercial screens. After selecting the best hits from the screening, manual optimization was performed. The best X-ray diffraction data set was collected for a crystal grown in 1.6 M ammonium sulfate, 0.1 M bicine, pH 9.0. For the X-ray diffraction experiments, the crystals were mounted in a nylon fiber loop and flash-cooled to 100 K in liquid nitrogen. The cryoprotection was performed for 2 s in a reservoir solution complemented with 20% (v/v) ethylene glycol. Diffraction data were collected at 0.99999 Å wavelength on the X06SA beamline (SLS, Villigen, Switzerland) at 100 K. Data set was indexed and integrated using XDS[96] (v. 01.10.14) and scaled using SCALA[97,98] (v. 01.10.14). Intensities were converted to structure-factor amplitudes using the program TRUNCATE[99] (v. 8.0.019). Supplementary Table 3 summarizes data collection and processing statistics.

## X-ray structure determination and refinement

The structure of BclxLΔLT-VDAC1-N complex was solved by molecular replacement (PHASER[100], v. 2.8.3) using the structure of BclxL published by Muchmore et al. (PDB: 1MAZ[101]). Model rebuilding was performed in COOT[102] (v. 0.9.8.93). The refinement was done in REFMAC5[103] (v. 5.8.0158) using the maximum likelihood- target function. The stereochemical analysis of the final model was done in PROCHECK[104] (v. 3.5.4) and MolProbity[105] (v. 4.02b-467). The analysis of the Ramachandran plot indicated that 96% of the amino acid

residues are in the most favored region and 4% in the additionally allowed region. The final model is characterized by R$_{work}$ and R$_{free}$ factors of 16.98% and 20.06%, respectively (Supplementary Table 3). Atomic coordinates and structure factors have been deposited in the Protein Data Bank under accession code 9HPS.

## NMR spectroscopy

NMR experiments were recorded on Bruker spectrometers operating at 500 to 950 MHz proton frequency equipped with cryogenic probes and controlled by Topspin 4.0 (Bruker Biospin). Spectral analysis was done with NMRFAM-SPARKY[106].

NMR titrations of *U*-[$^2$H, $^{15}$N]- ε-$^{13}$CH$_3$-methionine-labeled VDAC1-V3M in nanodiscs with BclxLΔTM (residues 1–213) were conducted on an 800 MHz $^1$H frequency spectrometer. Experiments were conducted in 20 mM NaPi, pH 6.8, 50 mM NaCl, 0.5 mM EDTA, and 2 mM DTT with 7% D$_2$O at 313 K. 2D-[$^{13}$C, $^1$H]-HMQC experiments with VDAC1-V3M in MSP1D1ΔH5 (50 μM) and VDAC1-V3M in MSP1D1 (75 μM) were recorded with 64 transients per increment and 48 complex data points in the indirect $^{13}$C dimension. BclxLΔTM was then added stepwise at concentrations of 100 μM, 500 μM, and 1 mM.

NMR titrations of $^2$H, $^{15}$N-labeled BclxLΔTM with VDAC1 in MSP1D1ΔH5 nanodiscs (8 nm diameter) were conducted at 950 MHz proton frequency. A 2D-[$^{15}$N,$^1$H]-TROSY spectrum of 100 μM $^2$H, $^{15}$N-labeled BclxLΔTM in 20 mM NaPi pH 6.5, 50 mM NaCl, 0.5 mM EDTA and 2 mM DTT with 7% D$_2$O, supplemented with protease inhibitor (cOmplete™, Roche) was recorded at 303 K. Subsequently, VDAC1 nanodiscs were added to at molar ratios of 1:1, 1:2, 1:4, and 1:10. The 1:2 titration point (100 μM $^2$H, $^{15}$N-labeled BclxLΔTM + 200 μM VDAC1 in MSP1D1ΔH5 nanodiscs) was used for analysis. To exclude effects of the interaction of BclxLΔTM with the lipid bilayer surface of the nanodisc particle, a 2D-[$^{15}$N,$^1$H]-TROSY spectrum of 100 μM $^2$H,$^{15}$N-labeled BclxLΔTM in presence of 200 μM empty MSPΔH5 nanodiscs was recorded and used as a reference. An identical workflow was done for probing the interaction between $^2$H, $^{15}$N-labeled BclxLΔTM and VDAC1 in MSP1D1 lipid nanodiscs (10 nm diameter). Furthermore, 100 μM $^2$H, $^{15}$N-labeled BclxLΔTM in complex with 100 μM PUMA BH3 peptide was titrated with 200 μM VDAC1 in MSP1D1ΔH5 nanodiscs and compared to the 2D-[$^{15}$N,$^1$H]-TROSY spectrum of $^2$H, $^{15}$N-labeled BclxLΔTM in complex with the PUMA BH3 peptide.

The interaction between $^2$H,$^{15}$N-labeled BclxLΔLT (residues 1–44 and 85–213) and the linear and stapled VDAC1-N peptides (residues 1–24 and 7–21 stapled between residues 11 and 15) was probed by NMR 2D-[$^{15}$N,$^1$H]-TROSY experiments at 303 K and at 500 or 800 MHz $^1$H frequency, respectively. The buffer was 20 mM NaPi, pH 7.0, 50 mM NaCl, 0.5 mM EDTA, 5 mM DTT. We used 100 μM of $^2$H,$^{15}$N-labeled BclxLΔLT and added the respective peptide until no marked chemical shift perturbation changes could be observed (10-fold excess for the linear peptide and 2.5-fold excess for the stapled peptide). The obtained binding curves were analyzed by a one-site binding model. For the stapled peptide, binding was also monitored with 100 μM *U*-$^2$H,$^{15}$N, Ile-δ$_1$,Leu-δ$_2$, Val-γ$_2$, Ala-β-[$^{13}$CH$_3$]-labeled BclxLΔLT using 2D-[$^{13}$C,$^1$H]-HMQC experiments measured at 303 K and at 800 MHz $^1$H frequency.

NMR was used to obtain binding affinities of VDAC1-N peptides in an alanine scanning experiment. VDAC1-N peptides (Supplementary Table 4) were titrated into 100 μM $^2$H,$^{15}$N-labeled BclxLΔLT in 20 mM NaPi, pH 7.0, 50 mM NaCl, 0.5 mM EDTA, 5 mM DTT. The resulting chemical shift perturbation was plotted against the peptide concentration to extract $K_D$ values for each peptide using a one-site binding model using OriginPro (OriginLab) for data fitting.

NMR spectra of 100 μM $^1$H,$^{15}$N-labeled VDAC1-N peptide were obtained at 303 K at 500 MHz $^1$H frequency in 20 mM NaPi, pH 7.0, 50 mM NaCl, 0.5 mM EDTA, 5 mM DTT. For mapping the binding site within the peptide for BclxL, a 5-fold excess of BclxLΔTM was added,

and chemical shift perturbations were mapped on the structure of VDAC1-N extracted from full-length VDAC1[7].

## Cryo-EM sample preparation

For cryo-EM sample preparation, 4 μL protein solution prepared as mentioned above was applied to a glow-discharged UltrAuFoil R 1.2/1.3, 300 mesh (Quantifoil Micro Tools). The grids were blotted for 4.5 s and plunge-frozen in liquid ethane after 1 s of drain time using a blot force of −1, at 100% relative humidity at 13 °C (Vitrobot Mark IV, Thermo Fischer Scientific). The plunged EM grids were clipped and used for automated dataset collection using the EPU software with a Krios G4 Cryo-TEM (Thermo Fischer Scientific) at 300 kV, equipped with an E-CFEG, a Selectris X Energy Filter at a slit width of 10 eV, and Falcon 4i Direct Electron Detector.

## Cryo-EM data processing

Movies for all datasets were monitored for quality and preprocessed on CryoSPARC Live[107,108]. The gain-corrected, motion-corrected, dose-weighted, and CTF-estimated micrographs were exported for further processing to CryoSPARC (v3.1.0-v4.6.2). For all subsequent jobs described, the following modifications were made to the default settings:

*2D Classification*: class2D_max_res and class2D_max_res_align were usually increased to 9 and 12 respectively, class2D_force_max was set to false, class2D_num_full_iter was set to 2–4, class2D_num_full_i-ter_batch was set to 40–100, and class2D_num_full_iter_batchsize_-per_class was set to 400–1000.

*Ab-Initio Reconstruction*: abinit_max_res was decreased to 6–9, abinit_init_res was reduced to 15-25, abinit_minisize_init was set to 300, and abinit_minisize was set to 1000.

*Non-Uniform Refinement*: refine_res_align_max was usually set to 5-12, to guard against refining on noise, especially in earlier jobs. To avoid masking artifacts that were obvious, especially in the cMSP1ΔH5 dataset, refine_dynamic_mask_far_ang was set to 18-24, and refine_dy-namic_mask_start_res to 6-8.

A compilation of jobs created, and the processing tree can be seen in Supplementary Fig. 4. The data acquisition statistics can be seen in Supplementary Table 1.

To corroborate that the missing density for the N-terminal helix of VDAC1 in cMSP1ΔH5 nanodiscs originates from the helix being detached from the β-barrel rather than being caused by misalignment of particles, we estimated the expected density features of a rotationally misaligned cMSP1ΔH5 particle stack, in the case of stable internal helix positioning. Starting from a map of the monomeric VDAC1 in cMSP1D1, which was already aligned with its flat side along the YZ plane, the center of the barrel's coordinates was manually identified in ChimeraX using a marker. Those were used to recenter the map and its associated particles, using CryoSPARC's "*Volume Alignment Tools*". 1000 random particles were selected and symmetry expanded to a C100 point group symmetry, using CryoSPARC's "*Symmetry Expansion*" job type. The resulting map was created by reconstructing the particles without any alignment, in CryoSPARC's "*Reconstruct Only*" Job.

## Molecular dynamics (MD) simulations

The complex between VDAC1 and the soluble domain of BclxL was assembled using the herein determined crystal structure of BclxL in complex with VDAC1-N. The C-terminal end of VDAC1-N in the complex was linked to the β-barrel of VDAC1 using Chimera[109]. This assembled complex between full-length VDAC1 and BclxL was inserted into a hexagonal box of 1,2-di-myristoyl-sn-glycero-3-phosphocholine (DMPC) and 1,2-di-myristoyl-sn-glycero-3-phosphoglycerol (DMPG) (3:1 ratio) lipid bilayer using the CHARMM-GUI web server (http://www.charmm-gui.org)[110,111] in water supplemented with 0.15 M KCl. Equilibration of the system was done at 310 K in 6 cycles, as outlined in

Supplementary Table 2. An MD simulation of 200 ns duration was carried out with the isothermal-isobaric ensembles at 310 K with the program NAMD v.12[112] on an in-house CPU cluster. A smoothing function was applied to truncate short-range electrostatic interactions. Periodic boundary conditions with a simulation box of approximately 86 Å by 43; 75 Å by 120 Å were used. Visualization of the MD trajectory was done with VMD[113].

VDAC1, with its N-terminal helix located in its canonical internal position as well as in the helix-outside position, was subjected to extended MD simulations in a POPC:POPG (3:1) lipid bilayer environment. The simulation was set up with the CHARMM-GUI web server (http://www.charmm-gui.org)[110,111] in water supplemented with 0.15 M KCl and run in triplicates for 5 μs at 310 K using the CUDA-enhanced version of GROMACS 2025[114] on an in-house GPU workstation. Periodic boundary conditions with a simulation box of approx. 144 Å by 144 Å by 96 Å (helix inside) or 120 Å by 120 Å by 108 Å (helix outside) were used. Long-range Coulomb interactions were accounted for by the particle mesh Ewald method[115]. The NPT ensembles were sampled at an ambient pressure of 1.0 bar, employing the c-rescale semiisotropic barostat[116], and temperature control was done by v-rescale method[117]. Analysis of the r.m.s.d. values for the entire protein, the β-barrel, and the N-terminal helix, respectively, of three independent simulations with different seed values was done with VMD v. 1.9.4[113]. All details of the MD setup are summarized in Supplementary Table 2.

### Reporting summary
Further information on research design is available in the Nature Portfolio Reporting Summary linked to this article.

## Data availability
The atomic coordinates have been deposited at the Protein Data Bank (PDB) under accession code 9HPS (X-ray structure of BclxLΔLT bound to the VDAC1-N-terminal α-helix). This structure was solved by molecular replacement using PDB code 1MAZ[101] (BclxL apo structure). The herein determined cryo-EM maps have been deposited to the Electron Microscopy Data Bank (EMDB) and the Electron Microscopy Public Image Archive (EMPIAR) under the accession codes EMD-55061 and EMPIAR-13001 (VDAC1 monomer in cMSP1D1 nanodiscs), EMD-55062 and EMPIAR-13010 (VDAC1 dimer in cMSP1D1 nanodiscs), and EMD-55094 and EMPIAR-13011 (VDAC1 monomer in cMSP1D1ΔH5 nanodiscs), respectively. Source data generated in this study are provided in the Supplementary Information/Source Data. MD input files and calculated trajectories for VDAC1 with different positions of the N-terminal helix (5 μs each), as well as the energy-minimized structural model of the VDAC1-Bclxl complex generated in this study, are available in the FigShare repository [https://figshare.com/s/dce14b2c4b97eff221f9]. Source data are provided with this paper.

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

## Acknowledgements

This study was supported by the Helmholtz Society (to F.H. VG-NG-1035) and the German Research Foundation (DFG) within the CRC1035 (project B13, project number 201302640 to F.H.). The authors acknowledge spectrometer time at the Bavarian NMR Center (www.bnmrz.org), Drs. Elisabeth Häusler and Shakeel Shahid for initial experiments, and Drs. Gerd Gemmecker and Sam Asami for general NMR support, Marie Tran for help with protein production, and Dr. Oleg V. Maltsev for peptide synthesis. The authors acknowledge the use of the X-ray Crystallography Platform at Helmholtz Munich. The cryo-EM data were collected at "Cryo-EM SoN," the cryo-EM infrastructure of the University of Münster, funded by the DFG—Project number 496113311. C.G. acknowledges funding through CRC1430, Project A04 (DFG, 424228829) and CRC1348, Project A15 (DFG, 386797833).

## Author contributions

M.D, U.G., G.B., K.L., R.J., K.F., and F.H. conducted research and analyzed data. M.D., U.G., G.B., C.G., R.J., and F.H. wrote the manuscript. All authors commented on and approved the manuscript. F.H. designed the study. D.N., C.G., and F.H. acquired funding.

## Funding

## Competing interests

The authors declare no competing interests.
