## [Transparent Peer Review file · Nature Communications]

Structural basis of apoptosis induction by the mitochondrial voltage dependent anion channel

Corresponding Author: Professor Franz Hagn

Version 0:

Reviewer comments:

Reviewer #1

(Remarks to the Author)

This manuscript presents a compelling and potentially paradigm-shifting model in which oligomerization of VDAC1, that occurs under mitochondrial stress, leads to the extrusion of its N-terminal α -helix from the β -barrel, enabling it to bind directly to the BH3-binding groove of the anti-apoptotic protein BclxL. The interaction with BclxL leads to the release of the proapoptotic proteins Bak and Bax from BclxL which then interferes with BclxL antiapoptotic function..

In short, the summary of the mechanism is:

VDAC1 \rightarrow Stress \rightarrow Oligomerization \rightarrow Helix exposure \rightarrow BclxL binding \rightarrow Bak/Bax release \rightarrow MOMP \rightarrow Apoptosis

As noted, this work presents a truly novel model, and the authors have done an excellent job in assembling and presenting their data. However, given that several of the findings challenge prevailing views and prior data in the field, aspects of the manuscript are likely to be viewed as controversial. To strengthen the manuscript and preempt criticism from the broader community, I will offer constructive pushback on key points that would benefit from clarification or additional support. Addressing alternative interpretations and reinforcing the mechanistic claims with additional context or validation will significantly enhance the impact and credibility of the study.

While the background effectively contextualizes the role of VDAC1 in mitochondrial function and apoptosis, some aspects require more balanced framing to align with the current literature.

1. The introduction tends to overstate the consensus around VDAC1 as a direct regulator of apoptosis, whereas many of its proposed interactions—particularly with Bax and Bak—remain under debate. Clarifying that these interactions are still the subject of active investigation would improve the manuscript's scientific positioning.
2. The authors do not sufficiently distinguish VDAC1 from its isoforms, notably VDAC2, which has a well-documented role in sequestering Bak. Introducing this distinction early would enhance clarity and help avoid conflating mechanistic pathways.
3. The background also lacks acknowledgment of prior structural studies that consistently place the N-terminal helix inside the VDAC1 β -barrel. Especially there is a large body of crosslinking data that shows the helix is affixed during gating, so there should be a clear distinction between gating and interactions with other proteins.
 - a. Given that the proposed model hinges on helix extrusion, it would strengthen the manuscript to explicitly recognize this point and frame the helix release as a novel and potentially controversial hypothesis. Overall, softening definitive language and providing a more nuanced view of existing controversies would better prepare the reader for the bold claims that follow.

VDAC1 oligomerization leads to exposure of its N-terminal α -helix

Results:

1. The authors begin by addressing whether VDAC1's N-terminal α -helix, previously observed stably tucked within the β -barrel, can become exposed under conditions that promote apoptosis. They hypothesize that such exposure is facilitated by oligomerization, which has been linked to apoptosis in prior studies. To investigate this, they perform chemical crosslinking experiments using BS3 under varying detergent and lipid environments. Their screen reveals that negatively charged environments—including the bile acid cholate and POPG-containing liposomes—promote significant VDAC1 oligomerization, whereas zwitterionic detergents like LDAO and CHAPS do not. These findings suggest that electrostatic interactions may drive a conformational change that enables oligomer formation and potentially helix extrusion.

Critique part 1

While the crosslinking experiments indicate that negatively charged environments promote VDAC1 oligomerization, the interpretation that this reflects functional or physiologically relevant structural changes—such as helix release—should be viewed with caution.

- In particular, SDS-PAGE is a limited and sometimes misleading tool for assessing oligomeric states of membrane proteins, which are highly sensitive to detergent composition and denaturation artifacts. Based on my own extensive experience with VDAC1 purification, I can attest that the protein exhibits substantial sensitivity to a wide range of detergents, often forming broad, aggregated peaks on SEC that reflect misfolding or partial denaturation rather than specific, stable oligomers. The detergents used here—particularly bile acids like cholate—may destabilize the β -barrel and promote structural rearrangements that are non-physiological.
- Furthermore, BS3 crosslinks amines and there many lysine residues located in the soluble loops and termini of β -sheets. How do the authors confirm that these oligomers are formed through physiological lateral interactions and not via random interactions like head-to-head etc. or within partially denatured proteins. Would it be possible to crosslink more specifically, i.e. engineered cysteines?
- The authors also suggest different causes for oligomerization and expulsion of the helix. In micelles and liposomes it is supposed to be the negative charge which leads to oligomerization and increased pressure. However, both micelles and liposomes are rather flexible and it is not clear how pressure would increase in these systems.
- Similarly, non-circularized nanodiscs also exhibit significant flexibility and it is hard to imagine that a difference of 2 nm in diameter would exert significant pressure on VDAC with an outer diameter of 5.5 nm.
- Can the authors comment on the large number of additional bands visible in Fig. S1d+e? Many don't seem to correspond to multiples of 31 kDa. Figs. 1b and S1c+d also show two bands around 30-35 kDa that are both labeled as monomer. What is the explanation for that? Finally, Fig. S1e seems to show some artifact with the absence of stain in the area from 63-135 kDa.
- The band of pegylated VDAC in cholate is also shifted towards lower molecular weight (Fig. 1c+e, S1a+b). Can the authors comment on that?
- While the authors suggest that oligomerization facilitates N-terminal helix extrusion, an alternative and equally plausible interpretation is that these harsh conditions lead to partial unfolding, artificially freeing the helix from the barrel interior—the claim that these conditions promote a native, regulated exposure of the N-terminus remains speculative.

2. To assess whether oligomerization leads to exposure of the N-terminal α -helix, the authors introduced a single cysteine at position T6C within a cysteine-free VDAC1 construct. They used the bulky cysteine-reactive reagent maleimide-PEG40 (PM40; ~40 kDa, 6.2 nm radius) to selectively label solvent-accessible cysteines. Indeed, they observed a moderate PEGylation in cholate conditions—matching those that promote oligomerization—whereas zwitterionic detergents showed less labeling. The authors interpret this as evidence that oligomerization facilitates helix exposure. To test the converse—whether helix stabilization suppresses oligomerization—they employed the E73V mutant, previously shown to stabilize the β -barrel and the helix–barrel interaction. E73V showed reduced oligomerization and lower PEGylation. Conversely, introducing an L10A mutation into the E73V background restored both helix exposure and oligomerization, supporting a link between structural destabilization, oligomerization, and helix disengagement.

Critique part 2:

While the PEGylation assay is a creative approach to assess helix exposure, there are several caveats that weaken the conclusions.

- PEGylation efficiency is not only a function of solvent accessibility but also depends on local mobility, orientation, and reaction environment—all of which may be dramatically altered in harsh detergents. The assumption that the 40 kDa PEG polymer is strictly excluded from the VDAC pore based on static diameter measurements does not account for the dynamic, possibly partially unfolded states of VDAC1 under harsh detergent conditions. As shown previously, detergents like cholate can denature β -barrels, and aggregation-prone states could transiently expose buried residues. Thus, PEGylation may be detecting non-native exposure, not a regulated conformational switch.
- The use of the E73V mutant is informative but not definitive. While it reduces oligomerization and labeling, this could reflect enhanced overall protein stability, not necessarily direct prevention of helix release. The restoration of labeling in the E73V/L10A double mutant is intriguing, but again, it's unclear whether this reflects specific structural remodeling or global destabilization, especially given that L10 is deeply embedded in the helix-barrel interface. Additionally, it has been shown that pH promotes only dimerization of VDAC via E73. While dimers are oligomers, the authors' data suggests that dimerization does not lead to the extrusion of the helix.
- The physiological relevance of these findings remains tenuous. As with the crosslinking experiments, the detergents and lipids used here do not faithfully mimic the mitochondrial outer membrane, raising concerns that the observed helix disengagement may not occur in vivo or may be an artifact of the experimental system.

Recommendation for results leading to Fig.1

The data presented in support of VDAC1 oligomerization and N-terminal helix exposure will likely receive significant scrutiny, as they are derived under harsh, non-physiological conditions. Both the crosslinking and PEGylation assays are performed in artificial environments—such as bile acid detergents and high concentrations of negatively charged lipids—that may not accurately reflect the native mitochondrial outer membrane. Moreover, the SDS-PAGE analysis shows only a modest fraction of oligomerized species, and the PEGylation signal, while intriguing, is indirect and potentially confounded by detergent-induced misfolding or non-native conformational states.

Cryo-EM structures of VDAC1 in lipid nanodiscs reveal conformational switching

Results:

To directly assess N-terminal helix positioning in different oligomeric or membrane states, the authors performed single-particle cryo-EM on VDAC1 reconstituted into circularized lipid nanodiscs of two different sizes (11 nm and 9 nm in diameter). In larger cMSP1D1 nanodiscs (~11 nm), both monomeric and dimeric VDAC1 species were resolved at ~7 Å resolution. In both cases, the N-terminal α -helix was clearly visible within the interior of the β -barrel, consistent with the canonical VDAC1 structure. In contrast, reconstitution into smaller cMSP1DH5 nanodiscs (~9 nm) yielded a more

heterogeneous population. Approximately 65% of particles adopted “tight” nanodisc conformations with direct VDAC1–scaffold interactions and no discernible lipid density. In these particles, the N-terminal helix was no longer visible in the cryo-EM map, suggesting increased flexibility or displacement from the barrel interior. The authors interpret the loss of helix density as structural evidence that the N-terminal helix becomes exposed to the cytosol under these constrained conditions, consistent with earlier PEGylation data and MD simulations showing increased barrel flexibility in the absence of the helix.

Critique

- The cryo-EM data and protocols seem of good technical quality with some caveats. The loss of particles after 2D classification, in particular for the sample in cMSP1 H5 is excessive, indicating that there is significant heterogeneity or bad particles in the sample.
- The FSC curve for this sample is also concerning. The large differences in reported resolution for the different masks and especially the sharp drop and rebound in resolution for the “corrected” curve indicate a problem with refinement or masking.
- The strong sidedness of the local resolution and loss of feature in the map, especially for the MSP is also peculiar. Do the authors have any explanation for that or any indication whether it could coincide with the location of the helix?
- I am also a bit surprised that the cryo-EM processing indicates very homogeneous samples for the two peaks from the sample in cMSP1D1, given how close those peaks are together in the SEC.
- The structural interpretation that the N-terminal helix is extruded based solely on the absence of density must be treated with caution. At the ~7 Å resolution reported, flexibility, orientation disorder, or partial occupancy can all lead to density loss, especially for short helices. The lack of N-terminal helix density in the smaller nanodiscs may reflect mobility or reconstruction limitations rather than bona fide extrusion.
- These “tight” nanodiscs represent a highly artificial state, characterized by a loss of visible lipid density and intimate protein–scaffold contact. It is unclear whether this environment mimics physiological mitochondrial membrane conditions or imposes mechanical constraints that displace the helix non-specifically.
- Their interpretation is further complicated by particle heterogeneity: a fraction of particles (~15%) showed expanded pore diameters which the authors determined were an artifact and these were excluded from refinement. An additional 20% apparently had the helix inserted, but no classes captured a clear outward-facing helix conformation. Thus, the evidence remains indirect and negative—absence of helix density is not equivalent to demonstration of helix ejection or exposure.
- Additionally, the authors do not provide a quantitative comparison of pore diameters between the helix-inserted and helix-displaced states. If the proposed mechanism involves physical compression by the ‘tight’ nanodisc leading to helix extrusion, one would expect a measurable reduction in pore diameter. Such an analysis would help substantiate the structural basis for displacement and should be included to support this model. Furthermore, the distances between the VDAC pore wall and the MSPs seem to be almost identical in all three samples, casting some doubt on the claim that the cMSP1 H5 nanodiscs are really significantly tighter.
- The fact that the local resolution for the helix in the cMSP1D1 sample with monomeric VDAC is significantly lower, also indicates higher mobility suggesting that possibly dimerization stabilizes the position of the helix.
- On the other hand, the density for the helix in the two protomers of the dimer also suggests that they might not be equal.
- Do the authors know whether the dimer in the nanodisc is in a parallel or antiparallel orientation?
- Finally, the authors should clarify whether focused classification or 3D variability analysis was attempted to resolve subpopulations with differing N-terminal helix states. These methods are well suited to identifying flexible or heterogeneous regions and could reveal additional insight into helix mobility that is currently inferred only by absence of density.
- It is also somewhat contradictory that the authors claim that the tight nanodisc constricts VDAC, yet at the same time it is supposed to be more flexible with the helix pushed out of the pore.

Recommendation for results leading to Fig.2

The data presented in support of VDAC1 N-terminal helix exposure will likely receive significant scrutiny, as they are derived from cryo-EM analysis of protein embedded in tightly constrained, nanodiscs with no lipids that may not faithfully mimic the mitochondrial outer membrane. While the absence of density for the N-terminal helix in the smaller (~9 nm) nanodiscs is interpreted as evidence of helix displacement, this conclusion remains speculative without direct visualization of the helix in an exposed position. Ideally, the authors should capture a reproducible class or subpopulation showing defined helical density external to the β -barrel, which would more convincingly support their model of helix extrusion.

Critique

The NMR and ITC data is very convincing. It does appear to show the ‘tight’ nanodisc support direct interaction between the VDAC1 N-terminal helix and the BH3-binding groove of BclxL. Additionally, while peptide binding data support the notion that the N-terminal helix can engage with BclxL, the affinities are modest (low μ M), and the increased binding of the hydrocarbon-stapled version raises concerns about artificial enhancement of the interaction through structural stabilization—the expelled N-terminus would not have this luxury. The stapled peptide is more helical by design, which may not reflect the natural conformation of the helix upon release.

Without direct structural visualization of the full-length complex—e.g., through cryo-EM or crosslinking mass spectrometry—the conclusion that VDAC1-N functionally mimics a BH3-only domain remains speculative. Sequence similarity to canonical BH3 motifs is minimal, and functional assays demonstrating competitive displacement of Bak or Bax from BclxL by full-length VDAC1 would be necessary to establish physiological relevance.

Furthermore, if the helix becomes exposed under these conditions and interacts with BclxL, the authors should attempt structural capture of a VDAC1–BclxL complex. Incubating the ‘tight’ nanodiscs with BclxL followed by purification and imaging could reveal a stable complex, which would substantiate both binding specificity and the functional relevance of helix exposure. At the very least, 2D class averages showing additional density attributable to BclxL would lend support to their model.

BclxL binds to the exposed N-terminal α -helix of VDAC1

Results:

Following evidence for N-terminal helix exposure, the authors next examined whether this region mediates direct interaction with anti-apoptotic BclxL. Using 2D- $[^{15}\text{N}, ^1\text{H}]$ -TROSY NMR with isotope-labeled BclxL, they observed significant chemical shift perturbations (CSPs) and line broadening upon addition of VDAC1 reconstituted in 8 nm nanodiscs—where the helix is proposed to be exposed—but not with 10 nm nanodiscs, where the helix remains inserted. These effects were abolished when BclxL was pre-saturated with a high-affinity BH3 peptide, indicating competition for the canonical BH3-binding groove. Mapping the CSPs confirmed that VDAC1-N binds at this groove. To further validate the interaction, the authors used a 24-residue VDAC1 N-terminal peptide and a stabilized, helical hydrocarbon-stapled version (st-VDAC1-N). Isothermal titration calorimetry (ITC) showed that the stapled peptide bound BclxL with a significantly higher affinity ($\sim 2 \mu\text{M}$) than the linear version ($\sim 29 \mu\text{M}$). NMR titrations using either backbone or methyl-labeled BclxL confirmed the binding specificity, and CSPs again clustered at the BH3-binding groove. A final titration using ^{15}N -labeled VDAC1-N peptide demonstrated direct binding to BclxL, and sequence alignment revealed partial homology with canonical BH3 domains.

Critique:

The authors present a compelling set of NMR and ITC data supporting direct interaction between the VDAC1 N-terminal helix and the BH3-binding groove of BclxL.

- Here I can only refer back to physiological relevance of the ‘tight’ nanodisc and that the helix exposure—and by extension, the interaction—is conditional on a non-physiological setup
- Additionally, while peptide binding data support the notion that the N-terminal helix can engage BclxL, the affinities are modest (low μM), and the increased binding of the hydrocarbon-stapled version raises concerns about artificial enhancement of interaction through structural stabilization. The stapled peptide is more helical by design, which may not reflect the natural conformation of the helix upon release.

Recommendation for results leading to Fig.3

The NMR data is quite strong. The data shows that ‘tight’ nanodisc with an exposed N-terminus interacts with BclxL, thus this full-length complex could be isolated and visualized providing direct structural evidence of the capture. Sequence similarity to canonical BH3 motifs is minimal, and functional assays demonstrating competitive displacement of Bak or Bax from BclxL by full-length VDAC1 would be necessary to establish physiological relevance.

High-Resolution Structure of the BclxL–VDAC1-N Complex

Result:

To obtain a high-resolution view of the interaction between VDAC1's N-terminal α -helix and BclxL, the authors generated a single-chain fusion construct in which the 26-residue VDAC1-N segment was covalently linked to the C-terminus of a truncated, crystallizable form of BclxL (BclxL Δ LT). This approach yielded crystals suitable for X-ray diffraction and enabled structure determination at 1.95 Å resolution. The resulting structure shows the VDAC1-N helix bound to BclxL's BH3-binding groove in an orientation generally consistent with canonical BH3 interactions, albeit shifted toward the periphery of the groove. The helix forms an amphipathic structure with key hydrophobic residues—particularly Leu10—contributing to the interface. Alanine scanning confirmed the functional relevance of these residues, with Leu10 showing the greatest impact on binding affinity. Interestingly, this residue is also important for anchoring the helix inside the VDAC1 barrel, and its mutation increases helix exposure. Finally, a full-length VDAC1–BclxL complex model was constructed via *in silico* docking and refined by molecular dynamics, positioning the extruded VDAC1-N helix adjacent to BclxL in a membrane-compatible orientation.

Critique:

The 1.95 Å crystal structure of the BclxL–VDAC1-N complex provides valuable atomic-level insight into the formation of the complex. The artificial tethering of the VDAC1-N peptide to the C-terminus of BclxL Δ LT may constrain conformational flexibility and promote an interaction that does not naturally occur—as seen with purification tags being resolved in binding pockets. Thus, the observed shift of the VDAC1-N binding interface and the extended conformation of the helix's C-terminus—may suggest that this represent a crystallographically stabilized or sterically accommodated contact driven by construct design. Based on Fig. 4b and Fig. 4d the binding grooves for VDAC-N and BakBH3 seem to be quite different. Or is the binding site of the unwound C-terminus of the VDAC-N peptide still part of the original BH3 binding groove? The difference in binding sites also casts some doubts on how effectively they would compete under physiological conditions. I am also doubtful whether the C-terminal half of the VDAC-N would undergo such an extensive unwinding upon release from the pore under physiological conditions. The authors also fail to explain why they used the native VDAC-N sequence for the crystallization experiment, even though they demonstrated that the stapled peptide had much higher affinity to BclxL. Additionally, there is very little description of the *in-silico* modeling of the full-length complex. I believe this should be expanded as the complex is truly what is desired. It is also not clear whether the orientation and position of the VDAC-N helix on BclxL at the end of the simulation is still the same as in the crystal structure. Based on Fig. 4g it seems to be quite different and much closer to VDAC and different from what is shown in Fig. 4c. While the alanine scanning experiments support some specificity, the overall binding affinity remains moderate, and does not aid in orienting the complex.

Recommendation for results leading to Fig.4

Altogether, the crystal structure offers a valuable atomic-level view of a potential interaction between the VDAC1 N-terminal helix and BclxL, establishing a plausible scaffold for complex formation. While the alanine scanning experiments lend

support to specific residue-level contacts, the overall binding affinity remains moderate and does not provide sufficient information to determine the native orientation of the full-length complex. The *in silico* model of the VDAC1–BclxL complex should be expanded with additional structural or biochemical constraints. In particular, the authors should clarify how BclxL engages the membrane surface and whether this orientation is compatible with the spatial positioning of VDAC1-N upon extrusion. Furthermore, given the number of proposed models in the literature regarding how Bcl-2 family proteins engage VDAC1, it would be helpful to contextualize this structure more explicitly: What does this new model confirm, and what competing hypotheses can now be ruled out? A clearer articulation of the structural and mechanistic implications of the complex is needed to understand its broader relevance in apoptotic regulation. Panels Fig. 4f and Fig. S10b are redundant, but Fig. 4f does not indicate that it is actually a reduction in affinity that is shown. Additionally, the order of the supplementary tables should be changed so that they match the order of the data in the text.

The VDAC1 N-terminus acts as a sensitizer BH3 protein to inhibit anti-apoptotic BclxL

Results:

To test whether the exposed VDAC1 N-terminal helix can functionally neutralize anti-apoptotic Bcl-2 proteins, the authors performed a liposome-based membrane permeabilization assay using Bak Δ TM (a soluble form of Bak lacking its transmembrane domain) and BclxL Δ TM. In this system, BclxL inhibits Bak-mediated pore formation, consistent with its anti-apoptotic role. However, titration with increasing concentrations of the VDAC1-N peptide (25–100 μ M) restored Bak activity in a dose-dependent manner, as evidenced by fluorescence dequenching due to dye release from the liposomes. This effect act by sequestering anti-apoptotic proteins and thus promoting apoptosis. Importantly, VDAC1-N alone could not activate Bak, nor did it substitute for a direct activator like cBid. These results support a model in which VDAC1-N does not directly activate Bak but instead displaces BclxL from Bak, thereby relieving inhibition and allowing Bak to execute membrane permeabilization (MOMP).

Critique:

I found this section particularly interesting, and the authors are to be commended for taking a functional approach to test their proposed mechanism. The liposome-based assay demonstrates that the N-terminal peptide of VDAC1 can restore Bak-mediated pore formation in the presence of BclxL, supporting the idea that VDAC1-N may sequester anti-apoptotic BclxL and thereby promote apoptosis. This is a plausible and biologically meaningful mechanism. However, several aspects of the experimental design and interpretation warrant further consideration.

First, the concentrations of N-terminal peptide used in the assay (25–100 μ M) far exceed the reported binding affinity (~2 μ M), raising concerns that the observed displacement may result from non-specific effects or mass-action kinetics rather than selective competition at the BH3-binding groove. Furthermore, the authors do not test whether unrelated amphipathic or α -helical peptides could elicit similar effects, leaving open the possibility that the result reflects general membrane or protein perturbation at high concentrations rather than specific functional mimicry.

Additionally, although the authors propose that N-terminal peptide acts analogously to BH3-only sensitizer proteins, they do not include a positive control using a canonical BH3-only sensitizer (e.g., BAD or NOXA) within the same experimental setup. Including such a benchmark would be critical to evaluating the relative potency and specificity of the N-terminal peptide. Finally, the findings would be more convincing if the functional assay were conducted with the ‘tight’ nanodisc sample containing full-length VDAC1 and a native, rather than engineered, N-terminal helix—especially since the stapled peptide used in prior assays likely enhances helicity and binding artificially.

Recommendation for results leading to Fig.5

This functional assay adds an important mechanistic layer to the study, linking N-terminal peptide exposure to the modulation of apoptotic signaling through BclxL neutralization. The experimental design is conceptually strong and aligns with the sensitizer BH3-only model proposed in earlier sections. However, to solidify the physiological relevance of this mechanism, several enhancements are recommended. First, the authors should address the supraphysiological concentrations of the peptide used in the assay and consider dose–response experiments that better reflect the reported binding affinity. Second, inclusion of a canonical BH3-only sensitizer protein (e.g., BAD or NOXA) as a positive control would allow for more rigorous benchmarking of VDAC1-N’s pro-apoptotic potency and specificity. Third, testing unrelated amphipathic peptides as negative controls would help rule out non-specific effects. Finally, conducting the assay with full-length VDAC1—in its ‘tight’ nanodisc conformation—would provide a more physiologically grounded system and reduce reliance on synthetic peptide models. These additional controls would significantly enhance the impact and interpretability of the functional data. Also, the authors should designate and used consistent terminology for when they are using an intact VDAC with the N-terminus exposed and when they are using the peptide.

Critique and Recommendation for the Discussion Section

The Discussion provides a comprehensive synthesis of the study’s findings and integrates them into a clear mechanistic model in which VDAC1 oligomerization drives N-terminal helix exposure, enabling interaction with BclxL and functioning as a BH3-like sensitizer to promote apoptosis. While this conceptual framework is compelling and well-articulated, several issues deserve further refinement to improve clarity and address overinterpretation.

First, while the authors emphasize the functional relevance of the ‘tight’ nanodisc system and its mimicry of an oligomeric VDAC1 state, this remains speculative. The notion that nanodisc-induced conformational changes replicate those triggered by physiological apoptosis stimuli (e.g., calcium, pH, or ROS) should be tempered and framed as a hypothesis rather than a conclusion. Moreover, there has been no evidence of oligomerization providing such strong forces to destabilize the helical interaction with the pore. Is there any evidence of such? This really should be discussed as it is the underlying premise. All critical inferences made here—such as helix extrusion, BclxL neutralization, and VDAC1’s classification as a BH3

sensitizer—are made without direct in vivo evidence. While prior literature supports aspects of this model, further validation in cellular systems will be needed to solidify VDAC1's proposed role.

In summary, the Discussion is well structured and highlights the novelty of the findings, but it should be revised to more carefully distinguish between established results and plausible, yet unproven, interpretations. It also is a bit long and should be reduced. The integration of cellular validation or more restrained language around speculative claims would strengthen the impact of this otherwise exciting mechanistic proposal.

Overall Recommendation:

This manuscript presents a novel and ambitious model in which the N-terminal α -helix of VDAC1, upon exposure through oligomerization, binds to the anti-apoptotic protein BclxL and thereby facilitates Bak-mediated mitochondrial outer membrane permeabilization (MOMP) and apoptosis. The study employs a broad and impressive array of structural and biophysical techniques, including cryo-EM, NMR, ITC, and crystallography, and attempts to bridge the gap between membrane protein dynamics and apoptotic signaling—a mechanistic connection that has long been hypothesized but not rigorously resolved.

The findings are bold and represent a significant conceptual departure from prior work. The idea that VDAC1, a β -barrel channel, could serve a BH3-like regulatory role in apoptosis is paradigm-shifting and will generate substantial interest in the mitochondrial biology and apoptosis communities.

Although I strongly believe this data should be published and shared for additional scrutiny, much of the model hinges on experimental systems that are highly engineered or removed from physiological like conditions. The reliance on denaturing detergents, negatively charged lipids, tightly packed nanodiscs, and synthetic peptides raises concerns about potential artifacts or overinterpretation. Several conclusions—particularly regarding helix extrusion, complex orientation, and BclxL neutralization—lack direct evidence in a cellular context. Additionally, structural interpretations from cryo-EM are based largely on absence of density, which at $\sim 6\text{--}7$ Å resolution must be interpreted with caution. Importantly, the authors do not show direct visualization of VDAC1-N outside the pore or capture a full-length VDAC1–BclxL complex in its native membrane environment.

I believe the study makes a provocative and potentially field-shifting contribution, but several claims must be tempered and additional controls or validation would be needed to support publication.

Reviewer #2

(Remarks to the Author)

This article describes new structures of VDAC1 and complexes of VDAC1's N-terminus with an anti-apoptotic protein, bclxl. The article is well written and clearly describes the experiments, results and conclusions. It could be accepted as is for publication and would already rank as high quality within Nat. Commun. publications. Given the chance to revise, the authors may wish to consider the following minor points:

- 1) Figure 1, the data is for detergents, that may enter the VDAC pore, but the conclusions are for lipids (scheme, panel f). This may warrant a comment.
- 2) The statement about 'our data' that VDAC oligomerization is the trigger. Line 180. Which data exactly indicates this causal relationship? Oligomerization as a marker of apoptosis, sure, but it would seem that something else must initiate the oligomerization, and then it would seem difficult to untangle cause and effect. E.g., could it be that N-terminal helix release triggers oligomerization? Is the wording 'correlated' line 449 more appropriate?
- 3) Line 428. There is now additional relevant literature, with additional structures available including in lipids and in complex with ER machinery.
- 4) Regarding the 9 nm nanodiscs, is the helix sufficient to align the particles? if not, then the helix density would be 'smeared out' in the 9 nm case, but not in the 11, due to the asymmetry. Are there structural features in the reconstruction that allow the orientation of VDAC to be fit to the remaining density even without the helix?
- 5) cMSP1DH5 is listed in the figure heading of Figure 2 as cMSPDH5. The size is discussed as 9 nm in the text wrt. Fig 2k, but labeled as 7.5 in Figure 2.

Reviewer #3

(Remarks to the Author)

In their paper, Daniilidis et al set out to address open questions about the role of VDAC1 in apoptosis induction. The authors examine the role of oligomerisation and the N-terminal helix of VDAC1 for its ability to act as a binding partner for pro-survival protein Bcl-xl, a well-established apoptotic stimulus, with a variety of methods including biochemical studies in charged detergents and lipids and cryoEM in different sized lipidic nanodiscs, as well as directly examining interactions with Bcl-xl using NMR and x-ray crystallography. The authors then used liposome-based pore-forming assays to confirm that VDAC1-N peptide can dissociate the anti-apoptotic Bcl-xl Bak complex and prevent liposome permeabilisation, lending further support to their hypothesis.

Overall, this is a well-conducted and exhaustive study that presents an intriguing possibility regarding the N-terminal helix of VDAC1. However, I am not convinced about the mechanism causing the release of the N-terminal helix. The observed loss of VDAC1 stability could explain most of the data regarding what triggers N-terminal helix release. The subsequent events—particularly the interaction with Bcl-xl and the release of Bak inhibition—are quite convincing, though additional controls, including the mutated peptides, would have strengthened the data (see below).

Specific comments:

- 1) The authors spend a significant portion of their manuscript showing that specific stimuli such as negatively charged

detergents and lipids cause VDAC1 oligomerisation and N-terminal release. Contrary to that, they claim that similar release could be observed in tight nanodiscs that constrain VDAC1 to a single monomeric species. An alternative explanation to this data could be the loss of protein stability and partial unfolding of VDAC1 in both conditions leading to the N-terminal release. This partial unfolding is not necessarily specific, or part of a well-defined biochemical pathway, such as proposed in Fig.5d, but rather a consequence of (partial) protein denaturation in the experimental conditions. Specific examples include:

1. The effect of cholate on VDAC1 stability could be due primarily to its detergent properties (higher CMC compared to other detergents tested, up to 14 mM at pH 7.5 according to Anatrace – very close to 20 mM used in the assay) rather than its charge. This proximity to the CMC might cause protein aggregation, resulting in apparent crosslinking. I remain unconvinced that charge plays a significant role in either stability or oligomerisation. This can be tested by using another negatively charged detergent with a lower CMC.
2. The conclusion about very large oligomers in POPG is unclear, as there appears to be a band even in the PC T6C condition that is very close to the larger band observed in the PC/PG condition.
3. VDAC1 is clearly less stable in 8nm nanodiscs (Fig.1e), but is this a consequence of N-terminal helix exposure or simply reduced stability due to lateral constraints?

II) I am also unconvinced about the release of the N-terminal helix in the 65% population in cMSP1dH5 sample as judged from the cryo-EM data. It is true that there is no density for the N-terminal helix in the map, but there could be an alternative explanation to that than the unfolding of the VDAC1 N-terminus: Unlike the MSP1D1 population, these particles are symmetrical (lacking either a dimer or asymmetrical lipid distribution within the nanodisc as opposed to the cMSP1D1 populations). As such, the absence of density for the N-terminal helix could be due to averaging effects, essentially from a lack of fiducial markers for particle alignments.

III) I cannot properly comment on the NMR experiments as this is outside my expertise. However, could the authors compare the chemical shifts induced by VDAC1 in 8 nm nanodiscs with those from the st-VDAC1-N peptide on Bcl-xL? If the authors' hypothesis that the N-terminus is released in 8nm nanodiscs is correct, then these samples should have similar perturbations on Bcl-xL as the N-terminal peptide alone.

IV) The authors have identified L10 as crucial for interaction with BclXI, suggesting that L10A VDAC1 or its VDAC1-N L10A would be ideal controls to test the specificity of VDAC1 N-terminus in inducing Bak pore formation (Fig. 5a). This is particularly important given the very high peptide concentrations used in the assay, to ensure the effect is specific.

V) While the exposure of the N-terminal helix of VDAC1 is linked to both oligomerisation and Bcl-xL dissociation from apoptotic effectors, there is a lack of commitment to a mechanism in the text. The authors should set out explicitly what they propose as a mechanism for apoptosis induction – i.e. what explicitly causes oligomerisation – merely upregulation/crowding? What about oligomerisation causes N-terminal helix exposure? Or is it the other way around? This can be written hypothetically; however, this additional evidence should allow for a clearer hypothesis.

VI) The major stimulus in this study is using negatively charged lipids which is suggested to cause both oligomerisation and N-terminal helix exposure or disorder. There is a lack of examination of the mitochondrial context of VDAC1 or controls to prove that both are caused simply by disorder or aggregation. Does the OMM outer leaflet contain lipids that would support N-helical exposure or prevent it? Are there specific aspects of apoptotic stimuli that modulate this? i.e. does a certain stimulus lead to an increase in negatively charged head groups in the OMM? Is this effect common to other VDACs? E.g. VDAC2 is established as being involved in Bak sequestration and mitophagy initiation caused by reactive oxygen species, are there residues or chemical environment differences that explain the studied function of VDAC1 that wouldn't occur with other VDACs?

Minor comments:

- 1) Line 117 may be referring to Figure S1d.
- 2) There might be a potential artifact in Figure S1e in the L10C and L10C+E73V lanes (PC-only conditions), as the overall exposure for these lanes looks significantly lower, even for non-specific bands.
- 3) There is a lack of clarity in the figures about the orientation of VDAC1 with regard to the mitochondrial membrane. The figures would be improved significantly by ensuring the OMM, cytosol and IMS are clearly indicated and all figures, including figure 5.
- 4) The methods section also contains some ambiguity, with citations for parts of methods instead of brief methodologies.

Reviewer #4

(Remarks to the Author)

Version 1:

Reviewer comments:

Reviewer #1

(Remarks to the Author)
Reviewer #1 – Second-Round Comments

I would like to thank the authors for their substantial and careful revisions to the manuscript and for their detailed point-by-point responses. Many of my original concerns—particularly regarding the framing of VDAC1's role, distinguishing between isoforms, and clarifying the structural evidence—have been addressed, and the manuscript has been strengthened as a result. Several of my more technical concerns (e.g., potential detergent artifacts, limitations of PEGylation and crosslinking approaches, and interpretation of cryo-EM maps at modest resolution) remain only partially resolved. However, I appreciate that the authors have acknowledged these caveats in the revised text and have tempered some of their claims accordingly. This is an appropriate way forward given the scope of the study and the technical challenges inherent in capturing dynamic conformational states of VDAC1.

The study still presents bold mechanistic conclusions that will likely generate discussion in the field, but I believe the revisions have made the manuscript clearer, more balanced, and more transparent about the limitations of the current data.

Recommendation: I support publication pending only minor editorial refinements, primarily to ensure that speculative aspects remain framed as hypotheses rather than established conclusions.

Reviewer #2

(Remarks to the Author)

The authors have fully addressed my concerns, and in particular provide a convincing explanation regarding the particle alignment and potential for averaging of the density for the smallest nanodiscs, something that was asked by all three reviewers.

Reviewer #3

(Remarks to the Author)

I thank the authors for providing additional experiments to strengthen their hypothesis. In particular, the CD data is very important to show that the stability of VDAC1 is not reduced in cholate. In my opinion, it would be good to add Rebuttal Figure 10 to the supplemental figures of the manuscript.

The differences between PC and PC/PG lipids in terms of VDAC oligomerisation and PM40 binding are very subtle. However, there appears to be a difference.

I find the explanation for the absence of a helix in the 8nm anodisc dataset very elegant – showing the calculated result of uncertainty as a ring in the pore. I am satisfied by this explanation. I wonder if some of this discussion should be included in the methods section of the manuscript?

Could the authors add IMS and cytosol labels to panels in Figure 1f to help orient the readers?

In summary, this is an exhaustive study providing an interesting insight into the potential mechanism of VDAC1 involvement in the induction of apoptosis. With the incorporated changes, I think this is a good fit for the journal and I congratulate the authors on this large amount of work.

Reviewer #4

(Remarks to the Author)

We would like to sincerely thank all reviewers for their constructive and detailed comments that
helped us to improve the manuscript by including new experimental data, more precise
explanations of the existing data, and by re-writing parts of the manuscript.

We would like to emphasize that we did our best to corroborate the mechanistic model presented
in this paper by various methods, including biochemical assays, biophysical experiments and
structural methods, accumulating a large set of consistent data. For instance, our explanation for
the missing cryo-EM density for the N-terminal VDAC1 helix in small nanodiscs is substantiated
by the biochemical maleimide-PEG40 (PM40) experiments, CD-detected stability measurements,
NMR spectra and titration experiments with full-length VDAC and the X-ray structure of the
VDAC1-N-BclxL complex. Thus, even though the modest resolution obtained by cryo-EM for
such a relatively small and thus very challenging system would exclude a definitive conclusion,
the multitude of consistent data presented here provide a plausible and clear picture.

Reviewer #1 (Remarks to the Author):

This manuscript presents a compelling and potentially paradigm-shifting model in which
oligomerization of VDAC1, that occurs under mitochondrial stress, leads to the extrusion of its N-
terminal α -helix from the β -barrel, enabling it to bind directly to the BH3-binding groove of the anti-
apoptotic protein BclxL. The interaction with BclxL leads to the release of the proapoptotic proteins
Bak and Bax from BclxL which then interferes with BclxL antiapoptotic function.
In short, the summary of the mechanism is:
VDAC1 \rightarrow Stress \rightarrow Oligomerization \rightarrow Helix exposure \rightarrow BclxL binding \rightarrow Bak/Bax release \rightarrow
MOMP \rightarrow Apoptosis

As noted, this work presents a truly novel model, and the authors have done an excellent job in
assembling and presenting their data. However, given that several of the findings challenge prevailing
views and prior data in the field, aspects of the manuscript are likely to be viewed as controversial. To
strengthen the manuscript and preempt criticism from the broader community, I will offer constructive
pushback on key points that would benefit from clarification or additional support. Addressing
alternative interpretations and reinforcing the mechanistic claims with additional context or validation
will significantly enhance the impact and credibility of the study.
While the background effectively contextualizes the role of VDAC1 in mitochondrial function and
apoptosis, some aspects require more balanced framing to align with the current literature.

1. The introduction tends to overstate the consensus around VDAC1 as a direct regulator of apoptosis,
whereas many of its proposed interactions—particularly with Bax and Bak—remain under debate.
Clarifying that these interactions are still the subject of active investigation would improve the
manuscript's scientific positioning.

**We acknowledge that the precise role of VDAC1 in apoptosis, particularly its direct interactions**
**with partner proteins such as Bax and Bak, is an area of ongoing research and debate. The**
**manuscript states that "the exact mechanistic role of VDAC in the intrinsic induction pathways**
**of apoptosis remains highly controversial". Furthermore, in the discussion, we mention that**
**"previous studies have suggested the direct involvement of VDAC1 in forming a large pore in the**
**outer mitochondrial membrane", but also state that "Our data does not support a direct role of**
**VDAC1". We now added a statement in the discussion section stating that a more global**
**investigation on the interaction of VDAC1 with partner proteins is required to estimate the**
**broader role of VDAC1 in apoptosis induction. The goal in this paper was rather to show that**
**VDAC1 can expose its N-terminal helix and interact with anti-apoptotic BclxL. Thus, we feel that**
**our study will be a valuable trigger for more detailed studies along these lines.**

2. The authors do not sufficiently distinguish VDAC1 from its isoforms, notably VDAC2, which has a
well-documented role in sequestering Bak. Introducing this distinction early would enhance clarity and
help avoid conflating mechanistic pathways.

This manuscript primarily focuses on the most abundant isoform, VDAC1, as it is the isoform
often implicated in apoptosis linked to oligomerization and interaction with anti-apoptotic Bcl2
family proteins. While we feel that a detailed comparison with all VDAC isoforms is beyond the
scope of this specific study, we do provide a multiple sequence alignment of the N-terminal
regions of human VDAC1, VDAC2, and VDAC3 in the supplementary information (Fig. S12e).
This figure shows near-identical sequences for VDAC1 and VDAC3 N-termini and a 10-amino
acid N-terminal extension in VDAC2. Moreover, now we show that both VDAC1-N and
VDAC2-N cannot induce BAK pore formation (Suppl. Fig. 12f). With this workflow, we think
we do not overly generalize the results presented in this manuscript. Further studies are
required to address this question as stated in the end of discussion.

3. The background also lacks acknowledgment of prior structural studies that consistently place the N-
terminal helix inside the VDAC1 β -barrel. Especially there is a large body of crosslinking data that
shows the helix is affixed during gating, so there should be a clear distinction between gating and
interactions with other proteins.

We explicitly acknowledge the established structural view: "In the previously published
structures of VDAC1 the N-terminal α -helix is stably attached to the pore interior". Our work
builds upon this by investigating conditions that could lead to a conformational change. The
manuscript also references studies indicating that the helix-inserted state is strongly favored
under normal conditions.

The manuscript distinguishes between the canonical helix-inserted state, relevant for metabolite
transport and typical voltage gating, and the proposed helix-exposed state that occurs under
specific conditions like stress-induced oligomerization. Our data from thermal melting
experiments indeed show that the helix-exposed state is less stable, suggesting the absence of the
N-terminal helix at the barrel wall leads to structural instability, which has been previously linked
to a loss in functionality and voltage gating activity. We do not argue against these previous
findings.

To further address the comment of this referee, we now included additional references on VDAC1
crosslinking and gating.

a. Given that the proposed model hinges on helix extrusion, it would strengthen the manuscript to
explicitly recognize this point and frame the helix release as a novel and potentially controversial
hypothesis. Overall, softening definitive language and providing a more nuanced view of existing
controversies would better prepare the reader for the bold claims that follow.

We agree that the proposed helix extrusion is the key novel finding in this manuscript, which
has been neither considered much nor experimentally shown so far. This aspect and its
controversy are now mentioned in the discussion together with the existing data on the role of
the N-terminal helix for partner protein binding. We agree that there is controversy in the point
that the helix can even dissociate from the pore wall, which we tried very hard to show here.
However, such a helix-extrusion model can explain a large body of literature and thus appears
very convincing in the light of reported protein-protein interactions. For VDAC1 gating under
normal conditions, we agree that no helix extrusion is required.

VDAC1 oligomerization leads to exposure of its N-terminal α -helix

Results:

1. The authors begin by addressing whether VDAC1's N-terminal α -helix, previously observed stably
tucked within the β -barrel, can become exposed under conditions that promote apoptosis. They
hypothesize that such exposure is facilitated by oligomerization, which has been linked to apoptosis in
prior studies. To investigate this, they perform chemical crosslinking experiments using BS3 under
varying detergent and lipid environments. Their screen reveals that negatively charged
environments—including the bile acid cholate and POPG-containing liposomes—promote significant
VDAC1 oligomerization, whereas zwitterionic detergents like LDAO and CHAPS do not. These
findings suggest that electrostatic interactions may drive a conformational change that enables

oligomer formation and potentially helix extrusion.

Critique part 1

While the crosslinking experiments indicate that negatively charged environments promote VDAC1
oligomerization, the interpretation that this reflects functional or physiologically relevant structural
changes—such as helix release—should be viewed with caution.

• In particular, SDS-PAGE is a limited and sometimes misleading tool for assessing oligomeric states
of membrane proteins, which are highly sensitive to detergent composition and denaturation artifacts.
Based on my own extensive experience with VDAC1 purification, I can attest that the protein exhibits
substantial sensitivity to a wide range of detergents, often forming broad, aggregated peaks on SEC
that reflect misfolding or partial denaturation rather than specific, stable oligomers. The detergents
used here—particularly bile acids like cholate—may destabilize the β -barrel and promote structural
rearrangements that are non-physiological.

**We understand detergents must be used with caution since they might destabilize or even**
**denature VDAC1. However, we only use SDS-PAGE (under denaturing SDS conditions) to detect**
**oligomeric states after crosslinking at the indicated detergent conditions, i.e., for monitoring**
**higher molecular weight, covalently linked oligomers.**

**To assess the folding state of VDAC in the detergents used in this study, we measured its**
**secondary structure and thermal stability in each case. We now show in Fig.S2a and Fig.S2b that**
**VDAC1 exhibits a beta sheet secondary structure as expected and a cooperative unfolding**
**transition with a slightly shifted thermal melting point for cholate. Since we perform the**
**crosslinking and PM40 modification at ambient temperature, well below the melting temperature**
**in each condition, VDAC can be safely assumed to be correctly folded. We should also note that**
**we prepare the VDAC1 samples in LDAO as described in the literature, which has also been used**
**for structure determination, followed by a transfer of the folded VDAC1 from LDAO into the**
**other detergent, minimizing the danger of obtaining a misfolded protein preparation.**

**We now included the harsh detergent SDS as a negative control and indeed found that the CD-**
**spectral signature is markedly altered, rather resembling an alpha-helical structure.**
**Furthermore, the CD-detected thermal melting behavior of VDAC1 in SDS does not show a**
**cooperative, sigmoidal shape (Suppl. Fig. 2b), in stark contrast to the other detergent conditions**
**(LDAO, Cholate, CHAPS) that have been used for this study. TX100 could not be tested in these**
**assays since this detergent interferes with the CD-measurement due its aromatic ring structure.**

• Furthermore, BS3 crosslinks amines and there many lysine residues located in the soluble loops and
termini of β -sheets. How do the authors confirm that these oligomers are formed through
physiological lateral interactions and not via random interactions like head-to-head etc. or within
partially denatured proteins. Would it be possible to crosslink more specifically, i.e. engineered
cysteines?

**Chemical crosslinking via BS3 works in a random manner through primary amines in this**
**purified system. However, the efficiency of crosslinking is much more significant when the**
**proteins stay together. Especially for membrane proteins with only relatively small solvent**
**exposed loops, crosslinking probability due to random collisions reduces dramatically. The**
**control experiments shown below (Rebuttal Figure 1a) clearly demonstrate this phenomenon: in**
**the presence of SDS, where VDAC is not correctly folded and present as a monomer, BS3**
**crosslinking efficiency is highly reduced. Only in the presence of cholate, a significant increase in**
**protein crosslinking is observed and subsequently increasing cholate concentration does not affect**
**the oligomerization pattern. Similarly, when cMSP1D1, the membrane scaffold protein that tends**
**to dimerize, was used as a control, we do not observe unspecific protein crosslinks of higher**
**oligomers but just the expected dimeric species (faint band at 48 kDa) (Rebuttal Figure 1b, BS³**
**crosslinking lanes).**

 **Rebuttal Figure 1: (a) Effect of increasing detergent concentrations on VDAC1 crosslinking. (b)**
 **Control experiment for BS3 crosslinking and PM40 modification using a single cysteine**
 **containing protein, cMSP1D1 in LDAO and cholate.**

 **The design of engineered cysteines for crosslinking is only suitable if the mode of**
 **oligomerization is exactly known and even then, requires the screening of many positions. Thus,**
 **to obtain general information on the degree of oligomerization we used amino-specific**
 **crosslinking. Together with the controls and the additional experiments on VDAC1 stability and**
 **structure (CD), we can rule out that crosslinking of unfolded species or in irrelevant**
 **orientations is dominating our readout.**

 • The authors also suggest different causes for oligomerization and expulsion of the helix. In micelles
 and liposomes, it is supposed to be the negative charge which leads to oligomerization and increased
 pressure. However, both micelles and liposomes are rather flexible and it is not clear how pressure
 would increase in these systems.

 **The effect of lateral pressure on the oligomerization is indeed difficult to clarify, whereas we**
 **could clearly show that the negative charge induced stronger oligomerization and helix**
 **exposure in detergents and lipids. In lipids, in addition to a PG headgroup we now also included**
 **cholesterol-hemisuccinate (CHS) that is also negatively charged and could observe an increase**
 **in helix exposure, further highlighting the role of negatively charged lipids (Suppl. Fig. 1g).**
 **Additionally, we revised “pressure” terms to prevent confusion.**

 • Similarly, non-circularized nanodiscs also exhibit significant flexibility and it is hard to imagine that
 a difference of 2 nm in diameter would exert significant pressure on VDAC with an outer diameter of
 5.5 nm.

 **We agree with this reviewer that the word “pressure” is a bit misleading in this context.**
 **However, we have experimental evidence based on CD, NMR spectroscopy and cryo-EM that**
 **even the small non-circularized MSP nanodiscs show the same effect on VDAC1 helix exposure**
 **as observed with the circularized nanodiscs. The relevant size in the nanodiscs used here is the**
 **inner diameter, which is 8 nm and 6 nm respectively (also stated in the manuscript now),**
 **leading to changes in the thermal stability of VDAC1 (CD experiments), to the exposure of the**
 **N-terminus (PM40 assay), and to an interaction of full-length VDAC with BclxL, as detected by**
 **2D NMR. Overall, the linear versus circular nanodiscs do not show a marked difference in lipid**
 **phase transition behavior, as shown by us in a previous study (Daniilidis et al, JMB, 2022).**
 **Thus, we also do not expect a big difference between linear and circularized nanodiscs here.**

 • Can the authors comment on the large number of additional bands visible in Fig. S1d+e? Many don't
 seem to correspond to multiples of 31 kDa. Figs. 1b and S1c+d also show two bands around 30-35
 206 kDa that are both labeled as monomer. What is the explanation for that? Finally, Fig. S1e seems to
 207 show some artifact with the absence of stain in the area from 63-135 kDa.

**Suppl. Fig. 1d+e show the silver-stained gels of the crosslinked VDAC in liposomes. Silver**
**staining is very sensitive to any trace amount of protein contamination or degradation.**
**The two bands that are labeled as “monomer” actually are VDAC with or without internal**
**crosslinks that exhibit different running behavior in SDS PAGE.**
**The artifacts between 63 and 135 kDa in Fig S1e is caused by the presence of excess, uncoupled**
**PM40, which cannot be stained neither with Coomassie nor with silver staining. We now include**
**these explanations in the figure legend to Suppl. Fig. 1.**

• The band of pegylated VDAC in cholate is also shifted towards lower molecular weight (Fig. 1c+e,
S1a+b). Can the authors comment on that?

**The different running behavior most likely is caused by the amount of PM40-modified VDAC1.**
**As explained in the previous comment, the amount of uncoupled PM40 in the gel distorts**
**surrounding bands. This leads to a higher distortion in LDAO lanes (please see the shapes of**
**crosslinked bands in LDAO lanes and MSP lanes). In cholate, the PM40 modification is more**
**efficient, thus reducing the amount of unmodified PM40 (no or less distortion).**

• While the authors suggest that oligomerization facilitates N-terminal helix extrusion, an alternative
and equally plausible interpretation is that these harsh conditions lead to partial unfolding, artificially
freeing the helix from the barrel interior—the claim that these conditions promote a native, regulated
exposure of the N-terminus remains speculative.

**The zwitterionic detergent LDAO is also considered a harsh detergent with a relatively short**
**hydrophobic tail. Cholate with its steroid ring structure and the negative charge is certainly not**
**more denaturing than LDAO. Strikingly, in LDAO, helix exposure is not observed. In addition,**
**we could clearly correlate the occurrence of an oligomeric state, which is dissolved in LDAO (as**
**also observed for SDS in rebuttal figure 1a) but not in Cholate, with helix exposure. We believe**
**that this is a coupled process, i.e. helix exposure leads to barrel destabilization and a**
**destabilized barrel is subsequently stabilized by oligomerization – a concept that has been**
**discussed previously for VDAC (e.g. Mannella, *Int. J. Mol. Sci.* 2021, 22, 1685.**
**<https://doi.org/10.3390/ijms22041685>). This in turn means that oligomerization can drive helix**
**exposure. We agree with this reviewer that the VDAC1 stability is a key factor that governs**
**helix exposure. We have included additional data on the structural features of VDAC as well as**
**its thermal stability in the detergents used in this study (now in Suppl. Fig. 2a and 2b).**
**Furthermore, we used SDS as a non-native control which led to unfolding and consequently to**
**completely different behavior in all experiments.**

2. To assess whether oligomerization leads to exposure of the N-terminal α -helix, the authors
introduced a single cysteine at position T6C within a cysteine-free VDAC1 construct. They used the
bulky cysteine-reactive reagent maleimide-PEG40 (PM40; ~40 kDa, 6.2 nm radius) to selectively
label solvent-accessible cysteines. Indeed, they observed a moderate PEGylation in cholate
conditions—matching those that promote oligomerization—whereas zwitterionic detergents showed
less labeling. The authors interpret this as evidence that oligomerization facilitates helix exposure. To
test the converse—whether helix stabilization suppresses oligomerization—they employed the E73V
mutant, previously shown to stabilize the β -barrel and the helix–barrel interaction. E73V showed
reduced oligomerization and lower PEGylation. Conversely, introducing an L10A mutation into the
E73V background restored both helix exposure and oligomerization, supporting a link between
structural destabilization, oligomerization, and helix disengagement.

Critique part 2:

While the PEGylation assay is a creative approach to assess helix exposure, there are several caveats
that weaken the conclusions.

• PEGylation efficiency is not only a function of solvent accessibility but also depends on local
mobility, orientation, and reaction environment—all of which may be dramatically altered in harsh
detergents. The assumption that the 40 kDa PEG polymer is strictly excluded from the VDAC pore

based on static diameter measurements does not account for the dynamic, possibly partially unfolded
states of VDAC1 under harsh detergent conditions. As shown previously, detergents like cholate can
denature β -barrels, and aggregation-prone states could transiently expose buried residues. Thus,
PEGylation may be detecting non-native exposure, not a regulated conformational switch.

**As mentioned above, VDAC1 is compactly folded in cholate derived from CD spectroscopic**
**evidence. As a control for the effect of the environment on PM40 modification, we used a**
**protein (cMSP1D1) with an exposed single cysteine as a positive control. As can be clearly seen**
**in the rebuttal figure 1b for this exposed scenario, PM40 modification always takes place**
**irrespective of the used detergent and its concentration (PM40 lanes).**

**Additionally, we used other cysteine variants of VDAC1 as controls (Rebuttal Figure 2). While**
**oligomerization pattern of the VDAC1 variants are not affected (Rebuttal figure 2a), PM40**
**modification ratio changes expectedly depending on the location of the cysteines (Rebuttal**
**figure 2b). Rebuttal figure 2c shows the location of the residues on VDAC structure. A170 is**
**located at the inner surface of the barrel, close to the binding site of VDAC1-N. As expected,**
**due to lower accessibility, the PM40 modification probability of this residue was significantly**
**lower for A170C than that of the more exposed T6C. However, in the presence of cholate, this**
**region inside the barrel can still be slightly modified by PM40, which is not the case with**
**LDAO. These results confirm that in cholate, but not in LDAO, the binding site of the N-**
**terminal helix could be slightly modified by PM40 due to absence of the helix at its place. As**
**another control, we used the S57C/N76C double VDAC1 variant. Both residues are located at**
**the inner periphery of VDAC1 barrel, closer to the cytoplasmic exit of the pore, quite distant**
**from the binding site of the N-helix. In this case, PM40 modification worked in similar**
**efficiencies in both detergents, suggesting that the barrel is not altered in these detergents.**
**Together with the CD data (Rebuttal figure 2d), these data suggest that VDAC is correctly**
**folded in both detergents and the use of the PM40 modification assay as a tool to probe N-helix**
**exposure provides relevant information on the oligomerization-dependent conformational**
**switching of VDAC1.**

Rebuttal Figure 2: (a) BS³ crosslinking and (b) PM40 modification of VDAC mutants in LDAO and cholate. (c) The location of cysteines on VDAC structure: T6C is found on the N-terminal helix; A170C is found at the helix-binding site of the barrel; and S57C N76C mutant is found at the periphery close to the cytoplasmic exit of the barrel. (d) CD spectra of the VDAC1 mutants.

• The use of the E73V mutant is informative but not definitive. While it reduces oligomerization and labeling, this could reflect enhanced overall protein stability, not necessarily direct prevention of helix release. The restoration of labeling in the E73V/L10A double mutant is intriguing, but again, it's unclear whether this reflects specific structural remodeling or global destabilization, especially given that L10 is deeply embedded in the helix-barrel interface. Additionally, it has been shown that pH promotes only dimerization of VDAC via E73. While dimers are oligomers, the authors' data suggests that dimerization does not lead to the extrusion of the helix.

In our understanding, dimers are dimers, oligomers are larger assemblies. Depending on the available methods and technology, we here provide cumulating evidence to corroborate the proposed mechanism. With the large set of VDAC1 variants and conditions, we think that there is a fair amount of evidence that argues against an unspecific denaturation event of VDAC. Furthermore, we have not seen a strong dimer in our VDAC1 preparations after crosslinking but rather very large oligo/multimers. Thus, we do not discuss the dimeric state in the present manuscript.

• The physiological relevance of these findings remains tenuous. As with the crosslinking experiments, the detergents and lipids used here do not faithfully mimic the mitochondrial outer membrane, raising concerns that the observed helix disengagement may not occur *in vivo* or may be an artifact of the experimental system.

From a technical standpoint, the presented structural experiments must be performed *in vitro* with suitable membrane mimetics. VDAC1 structures have been determined in LDAO detergent micelles showing the correct structure. We introduced negatively charged lipids to

323 mimic the charge properties of the mitochondrial membrane. Similar systems have been
described to favor VDAC oligomerization previously using biophysical methods (REF: Betaneli
et al., 2012; Lafargue et al., 2025). It has also been reported that the negative charge density of
the MOM is increased during apoptosis induction (Matsko et al., 2001, Ostrander et al., 2001,
10.1074/jbc.M107067200). Thus, we firmly believe that chosen membrane mimetics, ranging
from various detergent micelles to pure lipid systems, provides a realistic picture.
Nevertheless, as close as we can get, we also performed the liposome experiments in a
mitochondria-like setup (40%POPC+30%POPE+10%POPG+10%DOPS+10%Cardiolipin). In
this setup, we observed a similar behavior in BS3 crosslinking and PM40 modification
experiments (now in Supp. Fig. 1f and g). Differently from PC and PC/PG liposomes, the
addition of 5 mM CHS did not result in a significant increase in PM40 modification efficacy
possibly due to the presence of cardiolipin. Similar results were observed also in a previous
publication (Betaneli et al., 2012).

Recommendation for results leading to Fig.1

The data presented in support of VDAC1 oligomerization and N-terminal helix exposure will likely
receive significant scrutiny, as they are derived under harsh, non-physiological conditions. Both the
crosslinking and PEGylation assays are performed in artificial environments—such as bile acid
detergents and high concentrations of negatively charged lipids—that may not accurately reflect the
native mitochondrial outer membrane. Moreover, the SDS-PAGE analysis shows only a modest
fraction of oligomerized species, and the PEGylation signal, while intriguing, is indirect and
potentially confounded by detergent-induced misfolding or non-native conformational states.

**We employed detergent and lipid conditions that are commonly used for *in vitro* and structural**
**studies of VDAC1. Thus, while we acknowledge the fact that this is not done in a cellular and**
**native environment, we think that the available *in vitro* tools are essential to gain structural and**
**mechanistic insights. Furthermore, the fact that VDAC1 oligomers are presented in apoptotic**
**cells aligns very well with the herein presented data.**

Cryo-EM structures of VDAC1 in lipid nanodiscs reveal conformational switching

Results:

To directly assess N-terminal helix positioning in different oligomeric or membrane states, the authors
performed single-particle cryo-EM on VDAC1 reconstituted into circularized lipid nanodiscs of two
different sizes (11 nm and 9 nm in diameter). In larger cMSP1D1 nanodiscs (~11 nm), both
monomeric and dimeric VDAC1 species were resolved at ~7 Å resolution. In both cases, the N-
terminal α -helix was clearly visible within the interior of the β -barrel, consistent with the canonical
VDAC1 structure. In contrast, reconstitution into smaller cMSP1DH5 nanodiscs (~9 nm) yielded a
more heterogeneous population. Approximately 65% of particles adopted “tight” nanodisc
conformations with direct VDAC1–scaffold interactions and no discernible lipid density. In these
particles, the N-terminal helix was no longer visible in the cryo-EM map, suggesting increased
flexibility or displacement from the barrel interior. The authors interpret the loss of helix density as
structural evidence that the N-terminal helix becomes exposed to the cytosol under these constrained
conditions, consistent with earlier PEGylation data and MD simulations showing increased barrel
flexibility in the absence of the helix.

Critique

• The cryo-EM data and protocols seem of good technical quality with some caveats. The loss of
particles after 2D classification, in particular for the sample in cMSP1DH5 is excessive, indicating
that there is significant heterogeneity or bad particles in the sample.

**We would like to express our gratitude to the reviewer for acknowledging our work. The loss of**
**particles during processing is influenced by multiple factors, primarily sample heterogeneity and**
**data processing strategies. In our hands, particularly for membrane proteins, it is common for**
**only a small fraction of the initial dataset to be included in the final data to achieve the most**

optimal results with regard to resolution, often as low as 1-5%. Therefore, this outcome is not
unexpected.

Specifically, regarding the tight nanodisc, we suspect that the additional particle loss compared
to other nanodisc samples may be attributed not only to large heterogeneity but also to the
absence of the central helix. Consequently, the small VDAC barrel lacks this distinct asymmetric
feature in its lumen, which is present in other nanodisc samples, making alignment more
challenging. To achieve optimal results, we employed more lenient picking and stricter selection
criteria in both 2D and 3D processing, to obtain high-quality 2D classes and volumes comparable
to those of the other samples. However, this approach inevitably led to increased particle loss.
Please also see our comments below (lines 1023-1046).

• The FSC curve for this sample is also concerning. The large differences in reported resolution for
the different masks and especially the sharp drop and rebound in resolution for the “corrected” curve
indicate a problem with refinement or masking.

We thank the reviewer for pointing this out. However, when masking a small, poorly resolved
region, as required for the refinement of the small nanodisc, jumps in the FSC are not completely
unexpected and difficult to avoid. To improve masking, we un-binned the particles, thus
increasing the gridding, and repeated the processing workflow. The new resulting maps are
consistent with the ones previously reported; the resulting FSC does not show any rapid decreases
in cross correlation and reports equivalent resolution. We updated Suppl. Fig. 4 accordingly.

**Rebuttal Figure 3: FSCs of VDAC1 in cMSP1D1AH5. (a) Previous FSC (b) Current FSC in**
**revised Figure S4**

• The strong sidedness of the local resolution and loss of feature in the map, especially for the MSP is
also peculiar. Do the authors have any explanation for that or any indication whether it could coincide
with the location of the helix?

Similarly, for the smaller nanodisc sample, the same issues were encountered when calculating
local resolution with the previous mask. Upon re-running the calculations with un-binned
particles, the observed sidedness of the local resolution of the map is significantly reduced. The
remaining sidedness can likely be attributed to the closer proximity of VDAC to the MSP on that
side of the reconstruction, but we do not observe any correlation with the location of the helix.

**Rebuttal Figure 4: Local Resolution. (a) Original (b) Replaced in Figure S4**

• I am also a bit surprised that the cryo-EM processing indicates very homogeneous samples for the
two peaks from the sample in cMSP1D1, given how close those peaks are together in the SEC.

**Indeed, monomer classes were observed in the late peak, as illustrated in Supplementary Figure**
**4. Additionally, few 2D class averages of the early peak do reveal dimers. Our intention with**
**Supplementary Figure 4 was to highlight the absence of size heterogeneity in the other samples,**
**except for the cMSP1D1ΔH5 reconstruction.**

**We regret that the original Supplementary Figure 3 and its description were misleading, and we**
**have revised them to clarify the confusion. The "100%" label has been removed from the more**
**homogeneous samples, and the legend has been revised to better convey our observations.**

*Revised legend Supp Fig. S3.*

*Figure S3. Homogeneity of VDAC1 nanodisc samples and cryo-EM analysis. (a) Comparative size*
*exclusion chromatogram of both samples. Fractions collected are highlighted. (b) SDS-PAGE of*
*cMSP1D1 VDAC1 (c) SDS-PAGE of cMSP1D1ΔH5 VDAC1 (d-f) Representative 2D class averages*
*of species found in each sample. (d) Large nanodiscs containing a single VDAC1 protomer from the*
*early peak of cMSP1E3D1 VDAC1 (e) Large nanodiscs containing a single VDAC1 dimer from the*
*early peak of cMSP1D1 VDAC1. Notice that monomeric contaminants were found and eliminated*
*in this dataset. (f) Variety of species found in the cMSP1D1ΔH5 VDAC1 dataset and their relative*
*abundance.*

• The structural interpretation that the N-terminal helix is extruded based solely on the absence of
density must be treated with caution. At the ~7 Å resolution reported, flexibility, orientation disorder,
or partial occupancy can all lead to density loss, especially for short helices. The lack of N-terminal
helix density in the smaller nanodiscs may reflect mobility or reconstruction limitations rather than
bona fide extrusion.

**We agree with the reviewer. Given the final resolution achieved, it is very difficult to ascribe**
**whether missing density corresponds indeed to extrusion or to other factors. Another factor can**
**be attributed to such difference might be the sample quality.**

**Here, we would like to emphasize the following points: The reviewer can be assured that our cryo-**
**EM sample preparation, including ice thickness and imaging conditions, has been thoroughly**
**optimized, ensuring that all datasets are of comparable quality. Notably, in the cMSP1D1 sample**
**containing a monomeric VDAC, the inserted helix is readily visible even during 2D classification,**
**without requiring extensive processing. In contrast, the tight nanodisc dataset, which was**
**processed identically, does not exhibit any additional density in the barrel, despite additional**
**significant efforts focusing on detecting the inserted helix.**

**However, other explanations of the missing structure, such as flexibility and/or partial occupancy**
**of the tail, as suggested by the reviewer, are all consistent with our observations and do support**
**our hypothesis. A more flexible tail would indeed be one that interacts more readily with BclxL.**

Overall, while the cryoEM data alone could be attributed to various factors, we urge the reviewer to consider the collective evidence presented in the manuscript. Our biochemical studies and NMR data provide strong support for helix extrusion, and when viewed in conjunction with the cryoEM results, it leads us to conclude that helix extrusion is the most plausible mechanism. We kindly request that this reviewer considers the entirety of our data when interpreting the cryoEM results, rather than evaluating them in isolation.

- These “tight” nanodiscs represent a highly artificial state, characterized by a loss of visible lipid density and intimate protein–scaffold contact. It is unclear whether this environment mimics physiological mitochondrial membrane conditions or imposes mechanical constraints that displace the helix non-specifically.

We think that this state is a mimic of the oligomeric form where VDAC1 is engaging in protein-protein interactions, leading to less lipids surrounding the protein. Although a slightly different barrel conformation is a possibility in nanodiscs and oligomers, we see an exposed helix at both conditions. The nanodisc technology is a suitable tool for us to be able to investigate defined structural states of VDAC by structural methods, which would not have been possible otherwise.

- Their interpretation is further complicated by particle heterogeneity: a fraction of particles (~15%) showed expanded pore diameters which the authors determined were an artifact and these were excluded from refinement. An additional 20% apparently had the helix inserted, but no classes captured a clear outward-facing helix conformation. Thus, the evidence remains indirect and negative—absence of helix density is not equivalent to demonstration of helix ejection or exposure.

The reviewer is correct in their pointing out of indirect data. However, as we mentioned earlier, we urge the reviewer to integrate all experimental results in their critique. While cryoEM lends itself to several plausible explanations, other lines of evidence presented herein (NMR, biochemical assays etc.) point, in our opinion, to the extrusion being a valid and demonstrable mechanism for VDAC. Furthermore, we point the attention to the reviewer to our answer above (lines 372-385).

- Additionally, the authors do not provide a quantitative comparison of pore diameters between the helix-inserted and helix-displaced states. If the proposed mechanism involves physical compression by the ‘tight’ nanodisc leading to helix extrusion, one would expect a measurable reduction in pore diameter. Such an analysis would help substantiate the structural basis for displacement and should be included to support this model. Furthermore, the distances between the VDAC pore wall and the MSPs seem to be almost identical in all three samples, casting some doubt on the claim that the cMSP1DH5 nanodiscs are significantly tighter.

The pore diameter appears to be relatively consistent across all reconstructions, measuring approximately 35-40 Å (Rebuttal Figure 5). Although we conducted a direct comparison of both reconstructions at the same sigma threshold, the limited resolution prevents us nevertheless from ruling out minor differences in pore diameter, which could potentially influence helix extrusion. Furthermore, we cannot determine at which stage of the reconstitution process helix extrusion occurs.

**Rebuttal Figure 5. Direct comparison of MSP1D1 VDAC1 dimer (gray) and MSP1D1ΔH5**
**VDAC1 tight (blue) shown in top view orientation. Scale Bar 40 Å.**

**In both MSP1D1 and MSP1D1ΔH5, the distance between VDAC1 and the MSP is indeed**
**identical, but only on the side of VDAC adjacent to the MSP. The other side of VDAC either**
**interacts with lipids (monomer in MSP1D1) and in the case of the dimer, additionally with an**
**adjacent VDAC monomer (see Rebuttal Figure 6; lipids are indicated in white). In contrast, the**
**tight nanodisc is surrounded by the MSP on all sides. The narrow MSP excludes any**
**interactions of the proteins with lipids; thus, it is significantly tighter.**

**Rebuttal Figure 6. Direct comparison of our 3 reconstructions, shown in top view orientation.**
**VDAC is shown in orange, MSP in gray and noise density that can be attributed to lipids in**
**white.**

• The fact that the local resolution for the helix in the cMSP1D1 sample with monomeric VDAC is
significantly lower, also indicates higher mobility suggesting that possibly dimerization stabilizes the
position of the helix.

**We kindly disagree with the reviewer that such a difference in local resolution in distinct**
**samples is significant enough to draw conclusions in this resolution scheme. While there might**
**be a possibility that the reviewer is correct, and the N-helix in the dimer could be more stable**
**than in the monomer, we prefer to avoid any possible overstatements at this stage.**

• On the other hand, the density for the helix in the two protomers of the dimer also suggests that they
might not be equal.

**We agree with the reviewer on that point. There seems to be an increased density for the helix of**
**one of the protomers compared to the other. However, given the resolution achieved, we cannot**
**unequivocally assign this difference to a distinct conformation or differential occupancy, as**
**opposed to simply a stochastically favorable alignment on one of the two protomers. Given,**
**however, that the best currently resolved dimeric crystal structures of VDAC1 (5XDN, 5XDO)**
**show two identical protomers, we assume this to be the case here as well, lacking more concrete**
**evidence.**

• Do the authors know whether the dimer in the nanodisc is in a parallel or antiparallel orientation?

The structure of VDAC1 is quite symmetric in the y axis at this resolution, because the helix is
almost at the center of the barrel. However, we set out to calculate if the orientation can be
rigorously tested. We constructed pdb files for parallel and anti-parallel orientations based on
structure 5xdo. Those pdb files were rigid-body-fitted into the density and we calculated the CC
of the fitting (reporting the map-to-model Cross Correlation in the region of the model) as
calculated using phenix.get_cc_mtz_pdb (resolution = 7, fixed models true).
The up-up (parallel) orientation, like structure 5XDO, exhibits the highest correlation, and we
utilized this configuration to generate the figures. Nevertheless, the difference in cross-
correlation (CC) between this orientation and the down-up conformation is negligible, making it
impossible to conclusively rule out the latter possibility.

Orientation	CC
Up-Up Dimer	0.527
Up-Up	0.595
Up-Down	0.578
Down-Up	0.589
Down-Down	0.544
Down-Down Dimer	0.511

**“Up-Up Dimer” in this table refers to the pdb fitted as the resolved dimer, as opposed to two**
**split protomers fitted individually in the “Up-Up” row.**

• Finally, the authors should clarify whether focused classification or 3D variability analysis was
attempted to resolve subpopulations with differing N-terminal helix states. These methods are well
suited to identifying flexible or heterogeneous regions and could reveal additional insight into helix
mobility that is currently inferred only by absence of density.

**Despite our best efforts to resolve any additional heterogeneity, including additional internal N-**
**helix positions (the reviewer can be assured that we tried all possible procedures), the suggested**
**approaches did not yield any further insights.**

• It is also somewhat contradictory that the authors claim that the tight nanodisc constricts VDAC, yet
at the same time it is supposed to be more flexible with the helix pushed out of the pore.

**Constriction causes extrusion of the helix generating local disturbances at VDAC1’s canonical**
**barrel conformation, which in turn causes flexibility in the barrel and the N-terminal helix. Our**
**thermal melting experiments (Suppl. Fig. 2b) and MD simulations (Suppl. Fig. 5) clearly show**
**this phenomenon.**

Recommendation for results leading to Fig.2

The data presented in support of VDAC1 N-terminal helix exposure will likely receive significant
scrutiny, as they are derived from cryo-EM analysis of protein embedded in tightly constrained,
nanodiscs with no lipids that may not faithfully mimic the mitochondrial outer membrane. While the
absence of density for the N-terminal helix in the smaller (~9 nm) nanodiscs is interpreted as evidence
of helix displacement, this conclusion remains speculative without direct visualization of the helix in
an exposed position. Ideally, the authors should capture a reproducible class or subpopulation
showing defined helical density external to the β -barrel, which would more convincingly support their
model of helix extrusion.

**Despite our best efforts, we have not been able to resolve the N-helix. We would like to**
**emphasize that the mechanistic model presented in this paper is not just based on the cryo-EM**
**results but shown in concert by various methods, including biochemical assays, biophysical**

experiments and structural methods, accumulating a large set of consistent data. For instance,
our explanation for the missing cryo-EM density for the N-terminal VDAC1 helix in small
nanodiscs is substantiated by the biochemical PM40 experiments, CD-detected stability
measurements, NMR spectra and titration experiments with full-length VDAC and the X-ray
structure of the VDAC1-N-BclxL complex. Thus, even though the modest resolution obtained
by cryo-EM for a relatively small and thus very challenging system would exclude a definitive
conclusion, the multitude of consistent data presented here provides a plausible and clear
picture.

Critique

The NMR and ITC data is very convincing. It does appear to show the ‘tight’ nanodisc support direct
interaction between the VDAC1 N-terminal helix and the BH3-binding groove of BclxL. Additionally,
while peptide binding data support the notion that the N-terminal helix can engage with BclxL, the
affinities are modest (low μM), and the increased binding of the hydrocarbon-stapled version raises
concerns about artificial enhancement of the interaction through structural stabilization—the expelled
N-terminus would not have this luxury. The stapled peptide is more helical by design, which may not
reflect the natural conformation of the helix upon release.

**It is well-known that intrinsically disordered proteins can fold into specific structures in**
**different environments or in contact with their binding partners. VDAC1-N was shown**
**previously to be unstructured in solution, but it folds into a helical structure in the vicinity of a**
**membrane surface (Reif et al., JMB, 2018). The use of a stapled peptide is an approximation for**
**this state in our experiments. Such chemical strategies are commonly used to enhance**
**interactions of helical peptides in *in vitro* studies (Refs: 10.1016/j.molcel.2006.08.020 and**
**10.1038/s41467-021-27851-y).**

**Although the described affinities are modest, the local concentrations of VDAC1, being the most**
**abundant isoform in the MOM, can easily compete for binding to BclxL, especially upon**
**oligomerization. In addition, in this study, we show that the linear peptide also binds and we**
**obtained our crystal structure with a linear peptide that was C-terminally fused to BclxL. Thus,**
**the stapled peptide should be considered a common tool for *in vitro* studies to enhance the**
**affinity by populating the bound conformation. This is particularly important for setups where**
**the partner proteins are not attached to the membrane (as seen in mitochondria for BclxL and**
**VDAC) which would strongly enhance the local concentrations to enable the formation of**
**complexes with μM affinities.**

Without direct structural visualization of the full-length complex—e.g., through cryo-EM or
crosslinking mass spectrometry—the conclusion that VDAC1-N functionally mimics a BH3-only
domain remains speculative. Sequence similarity to canonical BH3 motifs is minimal, and functional
assays demonstrating competitive displacement of Bak or Bax from BclxL by full-length VDAC1
would be necessary to establish physiological relevance.

**NMR experiments using isotope-labeled BclxL in Fig. 3 are performed in the full-length VDAC**
**context using differently sized nanodiscs. As can clearly be seen, our CSP data on BclxL cluster**
**around the BH3 binding site of BclxL. This result aligns very well with the CSPs induced by the**
**stapled or the unmodified peptide (Suppl. Fig. 8). Moreover, our crystal structure shows**
**VDAC1-N located at the very same binding site in the BclxL-VDAC1-N fusion protein. In**
**addition, we now included assay data with the well-known sensitizer BH3 peptide BAD, which**
**induces a similar response than seen with VDAC1-N.**

Furthermore, if the helix becomes exposed under these conditions and interacts with BclxL, the
authors should attempt structural capture of a VDAC1–BclxL complex. Incubating the ‘tight’
nanodiscs with BclxL followed by purification and imaging could reveal a stable complex, which
would substantiate both binding specificity and the functional relevance of helix exposure. At the very
least, 2D class averages showing additional density attributable to BclxL would lend support to their
model.

We set out to collect a VDAC1-cMSP1D1ΔH5-BclxL dataset. In a population of 2D class averages (approximately 30% of the particles in side-view orientation), we observe additional noise concentrated on one side of the barrel, in a subpopulation of particles, which is not observed in the other cryo-EM VDAC samples in the absence BclxL. Despite extensive classification efforts, we were unable to resolve the features in 3D. While it's tempting to speculate that this region of increased signal might correspond to BclxL, we cannot rule out other possibilities, e.g., noise from adjacent particles. Thus, we prefer to avoid any overstatements at this stage, and we do not include these data in the revised manuscript. However, these first results underline that BclxL interactions with full-length VDAC1 should be highly variable, making it a very challenging target for cryo-EM. This is not surprising, given that the interaction is mediated by a flexible, exposed helix with an affinity in the μM range.

Rebuttal Figure 7. Comparison of 2DCA with and without BclxL. (a) Representative 2D class averages (2DCA) of noiseless VDAC in MSP1D1ΔH5 (b) Representative 2DCA of VDAC1 in MSP1 ΔH5 with some additional noise on one side of the particle. All images were post-processed with a gamma value of $\gamma=1.75$ in CorelDraw, to increase visibility of noisy regions.

BclxL binds to the exposed N-terminal α -helix of VDAC1

Results:

Following evidence for N-terminal helix exposure, the authors next examined whether this region mediates direct interaction with anti-apoptotic BclxL. Using 2D- $[^{15}\text{N},^1\text{H}]$ -TROSY NMR with isotope-labeled BclxL, they observed significant chemical shift perturbations (CSPs) and line broadening upon addition of VDAC1 reconstituted in 8 nm nanodiscs—where the helix is proposed to be exposed—but not with 10 nm nanodiscs, where the helix remains inserted. These effects were abolished when BclxL was pre-saturated with a high-affinity BH3 peptide, indicating competition for the canonical BH3-binding groove. Mapping the CSPs confirmed that VDAC1-N binds at this groove. To further validate the interaction, the authors used a 24-residue VDAC1 N-terminal peptide and a stabilized, helical hydrocarbon-stapled version (st-VDAC1-N). Isothermal titration calorimetry (ITC) showed that the stapled peptide bound BclxL with a significantly higher affinity ($\sim 2 \mu\text{M}$) than the linear version ($\sim 29 \mu\text{M}$). NMR titrations using either backbone or methyl-labeled BclxL confirmed the binding specificity, and CSPs again clustered at the BH3-binding groove. A final titration using ^{15}N -labeled VDAC1-N peptide demonstrated direct binding to BclxL, and sequence alignment revealed partial homology with canonical BH3 domains.

Critique:

The authors present a compelling set of NMR and ITC data supporting direct interaction between the VDAC1 N-terminal helix and the BH3-binding groove of BclxL.

- Here I can only refer back to physiological relevance of the ‘tight’ nanodisc and that the helix exposure—and by extension, the interaction—is conditional on a non-physiological setup

- Additionally, while peptide binding data support the notion that the N-terminal helix can engage BclxL, the affinities are modest (low μM), and the increased binding of the hydrocarbon-stapled version raises concerns about artificial enhancement of interaction through structural stabilization.

The stapled peptide is more helical by design, which may not reflect the natural conformation of the helix upon release.

**Please see our previous comments regarding the concerns about non-physiological experimental**
**conditions and lines 602-617 for the use of the stapled peptide.**
**The relevance of our approach is corroborated by the high level of consistency of the NMR**
**experiments in the full-length context and the use of N-terminal peptide.**
**As mentioned before, local concentrations of VDAC1-N on mitochondria upon oligomerization**
**are most likely high enough to compete for BclxL.**

Recommendation for results leading to Fig.3

The NMR data is quite strong. The data shows that ‘tight’ nanodisc with an exposed N-terminus
interacts with BclxL, thus this full-length complex could be isolated and visualized providing direct
structural evidence of the capture. Sequence similarity to canonical BH3 motifs is minimal, and
functional assays demonstrating competitive displacement of Bak or Bax from BclxL by full-length
VDAC1 would be necessary to establish physiological relevance.

**As mentioned above, our data from biochemical and biophysical assays, NMR, cryoEM, X-ray**
**crystallography and pore forming assays need to be considered together and not as separate**
**datasets. This is the case for most studies, where one method cannot give a 100% safe and**
**definite answer. Since the affinity between BclxL and VDAC is not high enough, we cannot**
**purify the full-length complex for e.g. cryo-EM experiments. As shown above, we performed**
**cryo-EM studies with the complex but could not resolve a clear density for BclxL. While we**
**agree that a complex structure would be helpful to confirm the existing NMR and X-ray**
**structural data, we also think that an optimization of the sample preparation conditions for**
**cryo-EM using e.g. chemical crosslinking methods, would require substantial and time-**
**consuming efforts that we think are beyond the scope of this manuscript, especially since we**
**already provide a large set of structural data on the complex. We used this data to establish a**
**reliable full-length structural model. This was possible because our X-ray structure of the**
**BclxL-VDAC1-N complex shows a defined density for VDAC1-N up to the position where the**
**N-terminus is connected to the VDAC1 beta barrel.**

**Regarding the second comment of this reviewer, we already showed by NMR (Fig. 3) that full-**
**length VDAC1 in nanodiscs does not bind to BclxL if it is saturated with a BAK BH3 peptide. In**
**the pore forming assay, it is technically not feasible to work with full-length VDAC1, which**
**needs to be incorporated into liposomes at a sufficiently high concentration. Furthermore, since**
**VDAC1 is also permeable for small molecules, the sheer presence of VDAC1 in liposomes would**
**lead to a fluorescence signal independent of the formation of Bak pores.**

High-Resolution Structure of the BclxL–VDAC1-N Complex

Result:

To obtain a high-resolution view of the interaction between VDAC1's N-terminal α -helix and BclxL,
the authors generated a single-chain fusion construct in which the 26-residue VDAC1-N segment was
covalently linked to the C-terminus of a truncated, crystallizable form of BclxL (BclxL Δ LT). This
approach yielded crystals suitable for X-ray diffraction and enabled structure determination at 1.95 Å
resolution. The resulting structure shows the VDAC1-N helix bound to BclxL's BH3-binding groove
in an orientation generally consistent with canonical BH3 interactions, albeit shifted toward the
periphery of the groove. The helix forms an amphipathic structure with key hydrophobic residues—
particularly Leu10—contributing to the interface. Alanine scanning confirmed the functional
relevance of these residues, with Leu10 showing the greatest impact on binding affinity. Interestingly,
this residue is also important for anchoring the helix inside the VDAC1 barrel, and its mutation
increases helix exposure. Finally, a full-length VDAC1–BclxL complex model was constructed via
*in silico* docking and refined by molecular dynamics, positioning the extruded VDAC1-N helix adjacent
to BclxL in a membrane-compatible orientation.

Critique:

The 1.95 Å crystal structure of the BclxL–VDAC1-N complex provides valuable atomic-level insight
into the formation of the complex. The artificial tethering of the VDAC1-N peptide to the C-terminus
of BclxL Δ LT may constrain conformational flexibility and promote an interaction that does not

naturally occur— as seen with purification tags being resolved in binding pockets. Thus, the observed
shift of the VDAC1-N binding interface and the extended conformation of the helix's C-terminus—
may suggests that this represent a crystallographically stabilized or sterically accommodated contact
driven by construct design. Based on Fig. 4b and Fig. 4d the binding grooves for VDAC-N and
BakBH3 seem to be quite different. Or is the binding site of the unwound C-terminus of the VDAC-N
peptide still part of the original BH3 binding groove? The difference in binding sites also casts some
doubts on how effectively they would compete under physiological conditions. I am also doubtful
whether the C-terminal half of the VDAC-N would undergo such an extensive unwinding upon
release from the pore under physiological conditions. The authors also fail to explain why they used
the native VDAC-N sequence for the crystallization experiment, even though they demonstrated that
the stapled peptide had much higher affinity to BclxL.

**We do not have any evidence that the C-terminal end of VDAC1-N is in a non-native**
**conformation when bound to BclxL. As outlined in the results section and sketched in Figure 4,**
**the orientation of VDAC1-N and BclxL in that structure is highly consistent with the parallel**
**orientation of both components at the MOM. We do not see electron density of the linker**
**between BclxL and VDAC1-N. Thus, we cannot tell whether the complex is formed within one**
**fusion protein or between two protomers. However, in the light of crystal packing patterns it**
**seems highly plausible that the complex is formed between two protomers, which would**
**minimize any possibly conformational strain.**

**The binding sites of VDAC1-N and Bak-BH3 are not identical but are overlapping since they**
**are located in the same binding groove of BclxL. Thus, competition is possible, as seen also in**
**our NMR data and assays. The first few residues of the extended part of VDAC1-N are still in**
**the binding groove while the remaining five amino acids are pointing sideways. However, since**
**the binding sites in the BclxL groove are heavily overlapping, a stable interaction between**
**BclxL and VDAC1-N is not possible if BclxL is saturated with e.g. Bak-BH3 (see Fig. 3c).**

**Furthermore, the VDAC1-N region, once exposed to the pore exterior, becomes unfolded and**
**folds again upon binding to BclxL. In the VDAC1 structure, the internal helix is composed of**
**residues 11-20, which aligns very well with the helical content in the BclxL complex. Thus, even**
**if the helix would not unfold upon extrusion, no extensive unwinding would be required.**

**Finally, we used the stapled peptide for NMR experiments and obtained excellent data that**
**indicates that the binding mode is identical to the crystal structure that has been obtained with**
**the single-chain construct. Of course, we tried to crystallize also the BclxL-stapled peptide**
**complex but finally only the fusion protein approach worked. Due to the high local**
**concentration of the fused VDAC1 peptide binding was highly favored. Such an approach is**
**widely used to crystallize low affinity complexes in the μM range. In our view, this result is even**
**more relevant because we obtained structural data with the wild-type VDAC1-N sequence.**

**Additionally, there is very little description of the in-silico modeling of the full-length complex. I**
**believe this should be expanded as the complex is truly what is desired. It is also not clear whether the**
**orientation and position of the VDAC-N helix on BclxL at the end of the simulation is still the same**
**as in the crystal structure. Based on Fig. 4g it seems to be quite different and much closer to VDAC**
**and different from what is shown in Fig. 4c. While the alanine scanning experiments support some**
**specificity, the overall binding affinity remains moderate, and does not aide in orienting the complex.**

**With the crystal structure of the BclxL-VDAC1-N complex and the existing structure of full-**
**length VDAC1, we were able to assemble the full-length complex. The 200 ns simulation was**
**only used to relax rotamers in the assembled structure. At the end of the simulation, the**
**complex between VDAC1-N and BclxL was still intact and showed an identical interaction**
**pattern as in the X-ray structure (See rebuttal figure 8 below). The surface versus cartoon**
**representations might have given the wrong impression. However, it is also not unlikely that**

**BclxL will eventually dissociate if the simulation time is heavily extended, as expected for a**
**complex with μM affinity.**
**The orientation of the complex is highly defined by the determined experimental X-ray**
**structure, which is also highly consistent with the known orientations of both proteins in the**
**OMM. We have now included more details on the MD simulation setup in Suppl. Table 2 and in**
**the Suppl. Methods.**

**Rebuttal figure 8: Structural overlay between the full-length VDAC1-BclxL model after a 200**
**ns MD simulation (sand) and the herein determined X-ray structure of the BclxL-VDAC1-N**
**complex (blue). The binding mode between BclxL and VDAC1-N is not altered by the MD**
**simulation, but rather structural clashes and rotamers are resolved.**

Recommendation for results leading to Fig.4
Altogether, the crystal structure offers a valuable atomic-level view of a potential interaction between
the VDAC1 N-terminal helix and BclxL, establishing a plausible scaffold for complex formation.
While the alanine scanning experiments lend support to specific residue-level contacts, the overall
binding affinity remains moderate and does not provide sufficient information to determine the native
orientation of the full-length complex. The in silico model of the VDAC1-BclxL complex should be
expanded with additional structural or biochemical constraints. In particular, the authors should
clarify how BclxL engages the membrane surface and whether this orientation is compatible with the
spatial positioning of VDAC1-N upon extrusion. Furthermore, given the number of proposed models
in the literature regarding how Bcl-2 family proteins engage VDAC1, it would be helpful to
contextualize this structure more explicitly: What does this new model confirm, and what competing
hypotheses can now be ruled out? A clearer articulation of the structural and mechanistic implications
of the complex is needed to understand its broader relevance in apoptotic regulation.
Panels Fig. 4f and Fig. S10b are redundant, but Fig. 4f does not indicate that it is actually a reduction
in affinity that is shown.

**With the multitude of structural and biophysical data, we believe that the structural model is**
**well-founded, and we do not see any constructive further experimental data that can be**
**included in addition to the Ala scan of the VDAC1-N peptide, the X-ray structure of the**
**complex and the extensive NMR dataset. Membrane localization and orientation of BclxL is**
**well explored and mediated by its C-terminal region (truncated in our construct). We show**
**clearly that an interaction between both proteins is possible, based on the known membrane**
**binding and orientation features of both proteins (see Fig. 4c and 4g). It is correct that Figures**
**4f and Suppl. Fig. 10b are redundant. However, we would like to keep Suppl. Fig. 10b for the**
**sake of clarity for the reader to be able to fully evaluate Suppl. Fig. 10 as a stand-alone figure.**

Concerning Fig. 4f, an increase in K_d corresponds to a decrease in affinity. Fig. 4f shows the fold
increase in K_d relative to wt. We now added an extra indication that the affinity is reduced when
the K_d value is reduced. The relevance of the complex is finally discussed in the discussion
section. However, due to space limitations and for the sake of clarity we cannot discuss every
(well or less well founded) model of a Bcl2 protein-VDAC complex in the literature. We rather
focused on plausible and confirmed models.

Additionally, the order of the supplementary tables should be changed so that they match the order of
the data in the text.

**Thanks. This is done now.**

The VDAC1 N-terminus acts as a sensitizer BH3 protein to inhibit anti-apoptotic BclxL

Results:

To test whether the exposed VDAC1 N-terminal helix can functionally neutralize anti-apoptotic Bcl-2
proteins, the authors performed a liposome-based membrane permeabilization assay using Bak Δ TM
(a soluble form of Bak lacking its transmembrane domain) and BclxL Δ TM. In this system, BclxL
inhibits Bak-mediated pore formation, consistent with its anti-apoptotic role. However, titration with
increasing concentrations of the VDAC1-N peptide (25–100 μ M) restored Bak activity in a dose-
dependent manner, as evidenced by fluorescence dequenching due to dye release from the liposomes.
This effect act by sequestering anti-apoptotic proteins and thus promoting apoptosis. Importantly,
VDAC1-N alone could not activate Bak, nor did it substitute for a direct activator like cBid. These
results support a model in which VDAC1-N does not directly activate Bak but instead displaces
BclxL from Bak, thereby relieving inhibition and allowing Bak to execute membrane
permeabilization (MOMP).

Critique:

I found this section particularly interesting, and the authors are to be commended for taking a
functional approach to test their proposed mechanism. The liposome-based assay demonstrates that
the N-terminal peptide of VDAC1 can restore Bak-mediated pore formation in the presence of BclxL,
supporting the idea that VDAC1-N may sequester anti-apoptotic BclxL and thereby promote
apoptosis. This is a plausible and biologically meaningful mechanism. However, several aspects of
the experimental design and interpretation warrant further consideration.
First, the concentrations of N-terminal peptide used in the assay (25–100 μ M) far exceed the reported
binding affinity (\sim 2 μ M), raising concerns that the observed displacement may result from non-
specific effects or mass-action kinetics rather than selective competition at the BH3-binding groove.
Furthermore, the authors do not test whether unrelated amphipathic or α -helical peptides could elicit
similar effects, leaving open the possibility that the result reflects general membrane or protein
perturbation at high concentrations rather than specific functional mimicry.

**We agree that the requirement of up to 100 μ M VDAC1-N peptide asks for a better explanation.**
**Since we are working with an established *in vitro* pore forming assay where the VDAC1 peptide**
**is added in solution (and not attached to the membrane as with full-length VDAC1), the potency**
**of VDAC1-N is reduced. Furthermore, it should be kept in mind that in contrast to a direct**
**binding assay, the pore forming assay is performed in presence of other binding partners (Bak**
**and cBid) that shave a nM affinity for BclxL. In this competition assay, the nM binding affinity**
**needs to be competed out by the μ M potency of VDAC1-N. We now also included a mutated**
**VDAC1 peptide (L10A) that shows strongly reduced binding affinity (Suppl. Fig. 12g). This**
**peptide is by far less potent than the wild-type peptide, further corroborating the well-founded**
**conclusion that the interaction is specific. A higher affinity (nM) BH3-sensitizer peptide derived**
**from BAD showed a similar effect as seen for VDAC1-N but with a steeper concentration**
**dependency, as expected for that high-affinity binder (see Suppl. Fig. 12d). We also exclude**
**membrane perturbations by performing control experiments (Suppl. Fig. 13) with the peptide**
**only (without Bak or BclxL) where we did not see any leakage in the liposomes induced by the**
**peptide.**

Additionally, although the authors propose that N-terminal peptide acts analogously to BH3-only
sensitizer proteins, they do not include a positive control using a canonical BH3-only sensitizer (e.g.,
BAD or NOXA) within the same experimental setup. Including such a benchmark would be critical to
evaluating the relative potency and specificity of the N-terminal peptide. Finally, the findings would
be more convincing if the functional assay were conducted with the ‘tight’ nanodisc sample
containing full-length VDAC1 and a native, rather than engineered, N-terminal helix—especially
since the stapled peptide used in prior assays likely enhances helicity and binding artificially.

**Thanks for this excellent comment. As stated above, we included BAD BH3 in the assay and**
**observed a similar activity profile as with VDAC1-N, where no direct activation of Bak was**
**observed but a neutralization of the inhibitory activity of BclxL (Fig. 5 and Suppl. Fig. 12d).**

Recommendation for results leading to Fig.5

This functional assay adds an important mechanistic layer to the study, linking N-terminal peptide
exposure to the modulation of apoptotic signaling through BclxL neutralization. The experimental
design is conceptually strong and aligns with the sensitizer BH3-only model proposed in earlier
sections. However, to solidify the physiological relevance of this mechanism, several enhancements
are recommended. First, the authors should address the supraphysiological concentrations of the
peptide used in the assay and consider dose–response experiments that better reflect the reported
binding affinity. Second, inclusion of a canonical BH3-only sensitizer protein (e.g., BAD or NOXA)
as a positive control would allow for more rigorous benchmarking of VDAC1-N’s pro-apoptotic
potency and specificity. Third, testing unrelated amphipathic peptides as negative controls would help
rule out non-specific effects. Finally, conducting the assay with full-length VDAC1—in its ‘tight’
nanodisc conformation—would provide a more physiologically grounded system and reduce reliance
on synthetic peptide models. These additional controls would significantly enhance the impact and
interpretability of the functional data. Also, the authors should designate and used consistent
terminology for when they are using an intact VDAC with the N-terminus exposed and when they are
using the peptide.

**Please see our comments above.**

Critique and Recommendation for the Discussion Section

The Discussion provides a comprehensive synthesis of the study’s findings and integrates them into a
clear mechanistic model in which VDAC1 oligomerization drives N-terminal helix exposure, enabling
interaction with BclxL and functioning as a BH3-like sensitizer to promote apoptosis. While this
conceptual framework is compelling and well-articulated, several issues deserve further refinement to
improve clarity and address overinterpretation.

First, while the authors emphasize the functional relevance of the ‘tight’ nanodisc system and its
mimicry of an oligomeric VDAC1 state, this remains speculative. The notion that nanodisc-induced
conformational changes replicate those triggered by physiological apoptosis stimuli (e.g., calcium,
pH, or ROS) should be tempered and framed as a hypothesis rather than a conclusion. Moreover, there
has been no evidence of oligomerization providing such strong forces to destabilize the helical
interaction with the pore. Is there any evidence of such? This really should be discussed as it is the
underlying premise.

All critical inferences made here—such as helix extrusion, BclxL neutralization, and VDAC1’s
classification as a BH3 sensitizer—are made without direct in vivo evidence. While prior literature
supports aspects of this model, further validation in cellular systems will be needed to solidify
VDAC1’s proposed role.

In summary, the Discussion is well structured and highlights the novelty of the findings, but it should
be revised to more carefully distinguish between established results and plausible, yet unproven,
interpretations. It also is a bit long and should be reduced. The integration of cellular validation or
more restrained language around speculative claims would strengthen the impact of this otherwise
exciting mechanistic proposal.

**We agree that the conformational changes in the “tight” nanodisc system might not be**
**completely identical to the situation in VDAC1 oligomers. However, the induced release of the**
**N-terminal helix is a common feature that makes us confident that the nanodisc system is a**
**valuable tool to be able to study such exciting conformational changes by structural methods.**
**We like to emphasize that the focus of this manuscript is to explore the structural basis of**
**VDAC1’s role in apoptosis, taking the well-documented VDAC1 oligomer as a marker for**
**apoptosis as a starting point. We have some clues from the literature on what factors induce this**
**oligomerization process, but the picture is far from clear at this point. There could be other**
**mechanisms involving VDAC1 that could contribute to its pro-apoptotic activity, which we also**
**discussed in this paper.**
**We now added a statement in the final paragraph emphasizing the need for a future *in vivo***
**investigation.**

Overall Recommendation:
This manuscript presents a novel and ambitious model in which the N-terminal α -helix of VDAC1,
upon exposure through oligomerization, binds to the anti-apoptotic protein BclxL and thereby
facilitates Bak-mediated mitochondrial outer membrane permeabilization (MOMP) and apoptosis.
The study employs a broad and impressive array of structural and biophysical techniques, including
cryo-EM, NMR, ITC, and crystallography, and attempts to bridge the gap between membrane protein
dynamics and apoptotic signaling—a mechanistic connection that has long been hypothesized but not
rigorously resolved.
The findings are bold and represent a significant conceptual departure from prior work. The idea that
VDAC1, a β -barrel channel, could serve a BH3-like regulatory role in apoptosis is paradigm-shifting
and will generate substantial interest in the mitochondrial biology and apoptosis communities.
Although I strongly believe this data should be published and shared for additional scrutiny, much of
the model hinges on experimental systems that are highly engineered or removed from physiological
like conditions. The reliance on denaturing detergents, negatively charged lipids, tightly packed
nanodiscs, and synthetic peptides raises concerns about potential artifacts or overinterpretation.
Several conclusions—particularly regarding helix extrusion, complex orientation, and BclxL
neutralization—lack direct evidence in a cellular context. Additionally, structural interpretations from
cryo-EM are based largely on absence of density, which at $\sim 6\text{--}7$ Å resolution must be interpreted with
caution. Importantly, the authors do not show direct visualization of VDAC1-N outside the pore or
capture a full-length VDAC1–BclxL complex in its native membrane environment.
I believe the study makes a provocative and potentially field-shifting contribution, but several claims
must be tempered and additional controls or validation would be needed to support publication.

**Thanks for this overall very positive assessment. Further studies will be required to address the**
**many remaining open questions, which are, however, beyond the scope of a single paper.**

Reviewer #2 (Remarks to the Author):
This article describes new structures of VDAC1 and complexes of VDAC1's N-terminus with an anti-
apoptotic protein, bclxl. The article is well written and clearly describes the experiments, results and
conclusions. It could be accepted as is for publication and would already rank as high quality within
Nat. Commun. publications. Given the chance to revise, the authors may wish to consider the
following minor points:
1) Figure 1, the data is for detergents, that may enter the VDAC pore, but the conclusions are for
lipids (scheme, panel f). This may warrant a comment.

**We thank reviewer #2 for this comment. Our conclusion/model in Fig. 1f is derived from data**
**obtained from biochemical data in detergents, nanodiscs and lipids.**

2) The statement about ‘our data’ that VDAC oligomerization is the trigger. Line 180. Which data
exactly indicates this causal relationship? Oligomerization as a marker of apoptosis, sure, but it would
seem that something else must initiate the oligomerization, and then it would seem difficult to

untangle cause and effect. E.g., could it be that N-terminal helix release triggers oligomerization? Is
the wording 'correlated' line 449 more appropriate?

**We agree with this reviewer that these two events are reciprocal and tightly correlated. As a**
**trigger for oligomerization/helix exposure, VDAC expression level, lipid composition, ROS**
**damage and many other factors might be relevant (reported in the literature and cited in the**
**introduction and discussion section). VDAC1 oligomerization could also be prevented by**
**binding to a partner protein in the inner mitochondrial membrane. While this question is highly**
**relevant and interesting, our study here did not aim at clarifying the cause of VDAC1**
**oligomerization but rather the structural and functional consequences of this known marker of**
**apoptosis. To obtain a detailed molecular understanding of this crucial step, a more extensive**
**study needs to be performed probing partner protein interactions under apoptotic versus**
**normal conditions.**

3) Line 428. There is now additional relevant literature, with additional structures available including
in lipids and in complex with ER machinery.

**We added additional literature to our citations.**

4) Regarding the 9 nm nanodiscs, is the helix sufficient to align the particles? if not, then the helix
density would be 'smeared out' in the 9 nm case, but not in the 11, due to the asymmetry. Are there
structural features in the reconstruction that allow the orientation of VDAC to be fit to the remaining
density even without the helix?

**We thank this reviewer for their insightful comments. Firstly, we kindly ask the reviewer to refer**
**to our answer to the reviewer #1 (lines 405-412), referring to how we address the processing and**
**the integration of the cryoEM data into our overall model. However, the reviewer is correct that**
**asymmetry plays a role in the resolutions achieved herein. The existence of a structured helix at**
**a fixed position, as is the case for the reconstructions in MSP1D1, should result in an additional**
**density in the center of the structure in case of rotational misalignment, even in absolute**
**rotational alignment uncertainty, as the reviewer fears; this is something we do not currently**
**observe. To demonstrate that above point, we forced a maximum uncertainty in the rotational**
**poses of our single particle images in our monomer MSP1D1 dataset, by symmetrically expanding**
**the particles around the axis of VDAC on the side of the nanodisc, and then back-projecting. The**
**results (Rebuttal Fig. 9) show as expected a ring at the center of the pore (because of rotationally**
**averaging the central helix). In contrast, albeit very extensive efforts, we do not see any additional**
**density or noise in the 9 nm nanodisc reconstruction. The clear absence of the helix might reflect**
**increased helix dynamics and not necessarily extrusion. However, we ask this reviewer to consider**
**the biochemical studies and NMR data providing strong support for the extrusion mechanism.**
**When viewed in conjunction with the cryoEM results, this leads us to conclude that helix**
**extrusion is the most plausible mechanism.**

**Rebuttal Figure 9. Rotationally randomized reconstruction of monomeric VDAC1 in MSP1D1.**

**(a) Top view. Notice the red ring. (b) Side view with the VDAC barrel transparent. Even in total**

**rotational error, a ring is still observable, even without further alignment. [Notes on methods:**
**The structure was recentered around the VDAC pore, and 2000 randomly selected particles**
**were symmetry expanded to C36. The back-projection was computed without new alignment of**
**the images.]**

5) cMSP1DH5 is listed in the figure heading of Figure 2 as cMSPDH5. The size is discussed as 9 nm
in the text wrt. Fig 2k, but labeled as 7.5 in Figure 2.

**Thanks. This issue has been fixed. 75 A scale bar in this figure is based on the resolved cryoEM**
**structure in which the side chains of the MSPs are not resolved.**

Reviewer #3 (Remarks to the Author):

In their paper, Daniildis et al set out to address open questions about the role of VDAC1 in apoptosis
induction. The authors examine the role of oligomerisation and the N-terminal helix of VDAC1 for its
ability to act as a binding partner for pro-survival protein Bcl-xl, a well-established apoptotic
stimulus, with a variety of methods including biochemical studies in charged detergents and lipids and
cryoEM in different sized lipidic nanodiscs, as well as directly examining interactions with Bcl-xl
using NMR and x-ray crystallography. The authors then used liposome-based pore-forming assays to
confirm that VDAC1-N peptide can dissociate the anti-apoptotic Bcl-xl Bak complex and prevent
liposome permeabilisation, lending further support to their hypothesis.

Overall, this is a well-conducted and exhaustive study that presents an intriguing possibility regarding
the N-terminal helix of VDAC1. However, I am not convinced about the mechanism causing the
release of the N-terminal helix. The observed loss of VDAC1 stability could explain most of the data
regarding what triggers N-terminal helix release. The subsequent events—particularly the interaction
with Bcl-xl and the release of Bak inhibition—are quite convincing, though additional controls,
including the mutated peptides, would have strengthened the data (see below).

**Thanks to this reviewer for their very positive evaluation of our work.**

Specific comments:

I) The authors spend a significant portion of their manuscript showing that specific stimuli such as
negatively charged detergents and lipids cause VDAC1 oligomerisation and N-terminal release.
Contrary to that, they claim that similar release could be observed in tight nanodiscs that constrain
VDAC1 to a single monomeric species. An alternative explanation to this data could be the loss of
protein stability and partial unfolding of VDAC1 in both conditions leading to the N-terminal release.
This partial unfolding is not necessarily specific, or part of a well-defined biochemical pathway, such
as proposed in Fig.5d, but rather a consequence of (partial) protein denaturation in the experimental
conditions. Specific examples include:

**Even though we tried to validate known factors that might be relevant for VDAC1**
**oligomerization and apoptosis induction, the present manuscript is not intended to**
**systematically evaluate such factors. VDAC oligomerization has been described on a cellular**
**level as a marker for apoptosis previously. Thus, we take this fact as a starting point for our**
**current study and connect it to its interaction with BclxL, as suggested in previous studies.**

**We agree with this reviewer that it should be made very clear that the N-terminal helix**
**exposure is not caused by VDAC misfolding. We are convinced that VDAC1 stability is indeed**
**an important factor for oligomerization and helix release. We now included CD-detected**
**thermal melting data and CD spectra to estimate the secondary structure content and stability**
**in each detergent sample in Suppl. Fig. 2a and Suppl. Fig. 2b (see also comments to reviewer #1-**
**lines 125-144 and our control experiments in rebuttal figures 1 and 2).**

1. The effect of cholate on VDAC1 stability could be due primarily to its detergent properties (higher
 CMC compared to other detergents tested, up to 14 mM at pH 7.5 according to Anatrace – very close
 to 20 mM used in the assay) rather than its charge. This proximity to the CMC might cause protein
 aggregation, resulting in apparent crosslinking. I remain unconvinced that charge plays a significant
 role in either stability or oligomerisation. This can be tested by using another negatively charged
 detergent with a lower CMC.

**We now used higher detergent concentrations (up to 100 mM) for the crosslinking and PM40**
 **assays (Rebuttal figures 10a and 10b). We observed almost identical crosslinking patterns and**
 **PM40 modification at higher detergent conditions. Moreover, we observe the expected beta-**
 **sheet secondary structure (as is the case for LDAO) and a cooperative unfolding transition,**
 **indicating a stable fold even at higher detergent conditions at room temperature (Rebuttal**
 **figures 10c and 10d). Zwitterionic CHAPS keeps the beta barrel structure compactly folded at**
 **room temperature but does not expose its N-terminal helix (see manuscript Fig. 1). In contrast,**
 **if we use a negatively charged but very harsh detergent (SDS), the secondary structure of**
 **VDAC1 changes to alpha-helix without a cooperative melting behavior (now in Suppl. Fig. 2a**
 **and 2b). This negative control corroborates our conclusion that VDAC1 is properly folded in**
 **the detergent conditions used for the in vitro experiments. Similar behavior can be seen in**
 **nanodiscs where a highly cooperative unfolding transition is observed as well. Together with the**
 **data from liposomes with different types of lipids, the effect of charge for oligomerization and**
 **helix exposure becomes more obvious. Additionally, we repeated our liposome experiments with**
 **the addition of CHS (a negatively charged cholesterol analogue) (Now in Fig. S1g). The addition**
 **of CHS also caused a significant increase in VDAC1 helix exposure in addition to POPG lipids.**

**Rebuttal Figure 10. Effect of increasing detergent concentrations on VDAC1 crosslinking (same**
 **figure as in rebuttal figure 1a) (a) and PM40 modification (b). CD spectra (c) and thermal**
 **melting curves (d) of VDAC1-T6C mutant in different detergents at different concentrations.**
 **TX100 is not shown due to its interference with the measurement process. Similarly, higher**
 **CHAPS concentration interferes at lower wavelengths. Therefore, only the melting curve is**
 **shown.**

 2. The conclusion about very large oligomers in POPG is unclear, as there appears to be a band even
 in the PC T6C condition that is very close to the larger band observed in the PC/PG condition.

In liposomes, the oligomerization of VDAC1 is not easy to control. Our BS3 crosslinking data in Suppl. Fig. 1d shows only slight changes in oligomerization. When the lanes T6C PC and PC/PG are compared, a slightly stronger “very large oligomer” band for PC/PG can be observed. Now, we added a new set of experiments (Suppl. Fig. 1f and 1g) in liposomes. In Suppl. Fig. 1g, it becomes more obvious that the oligomeric species becomes stronger with the addition of CHS and this can be correlated with the degree of the PM40 modification.

3. VDAC1 is clearly less stable in 8nm nanodiscs (Fig.1e), but is this a consequence of N-terminal helix exposure or simply reduced stability due to lateral constraints?

We think that both effects contribute to the reduction in stability. However, our observations show that the N-terminal helix is an important factor for VDAC1 stability, which becomes apparent during the refolding procedure. VDAC1 without its N-terminal helix cannot be refolded into LDAO detergent micelles, whereas full-length VDAC1 gives good refolding yields.

II) I am also unconvinced about the release of the N-terminal helix in the 65% population in cMSP1dH5 sample as judged from the cryo-EM data. It is true that there is no density for the N-terminal helix in the map, but there could be an alternative explanation to that than the unfolding of the VDAC1 N-terminus: Unlike the MSP1D1 population, these particles are symmetrical (lacking either a dimer or asymmetrical lipid distribution within the nanodisc as opposed to the cMSP1D1 populations). As such, the absence of density for the N-terminal helix could be due to averaging effects, essentially from a lack of fiducial markers for particle alignments.

We kindly ask the reviewer to refer to our answer to the reviewer #1 (lines 372-385), where we provide further details of our particle alignment strategy regarding the tight nanodisc. We also would like to turn the attention of this reviewer to our answer to reviewer #2 (lines 1023-1046).

III) I cannot properly comment on the NMR experiments as this is outside my expertise. However, could the authors compare the chemical shifts induced by VDAC1 in 8 nm nanodiscs with those from the st-VDAC1-N peptide on Bcl-xL? If the authors' hypothesis that the N-terminus is released in 8nm nanodiscs is correct, then these samples should have similar perturbations on Bcl-xL as the N-terminal peptide alone.

Please see Figs. 3d and h where we color-coded the CSP magnitude induced by the VDAC1-N peptide and the full-length VDAC1 in 8 nm nanodiscs. The obtained patterns in each case strongly indicate that the perturbation patterns are very similar. This shows that the N-terminal helix is the only interaction site in VDAC1 with BclxL.

IV) The authors have identified L10 as crucial for interaction with BclxL, suggesting that L10A VDAC1 or its VDAC1-N L10A would be ideal controls to test the specificity of VDAC1 N-terminus in inducing Bak pore formation (Fig. 5a). This is particularly important given the very high peptide concentrations used in the assay, to ensure the effect is specific.

Thank you for this constructive comment. We now included a VDAC1-N L10A peptide in the pore forming assay and could observe the anticipated loss in activity compared to the wt peptide (Fig. 5c).

V) While the exposure of the N-terminal helix of VDAC1 is linked to both oligomerisation and Bcl-xL dissociation from apoptotic effectors, there is a lack of commitment to a mechanism in the text. The authors should set out explicitly what they propose as a mechanism for apoptosis induction – i.e. what explicitly causes oligomerisation – merely upregulation/crowding? What about oligomerisation causes N-terminal helix exposure? Or is it the other way around? This can be written hypothetically; however, this additional evidence should allow for a clearer hypothesis.

**Thank you for your comment. We did not explicitly explore the causes for VDAC1**
**oligomerization but rather considered this well-known fact as a starting point for our structural**
**investigations. We validated existing literature mostly centered around lipid composition.**
**However, a more systematic study to really explore the cause of VDAC1 oligomerization under**
**apoptotic conditions, which could involve many factors, ranging from expression levels,**
**metabolites and ROS to partner proteins is beyond the scope of the current manuscript and will**
**be investigated in future studies. Thus, there is no commitment to a mechanism on this topic.**
**Furthermore, we believe that helix exposure is happening downstream of the oligomerization**
**since oligomerization has a stabilizing effect on the barrel in its helix-exposed state. Our NMR**
**experiments with VDAC1 in 8 and 10 nm nanodiscs (Suppl. Figs. 6 and 9) show that the helix**
**inserted and helix exposed states exist in an equilibrium and, depending on the size of the**
**nanodiscs, we observe both states (10 nm nanodiscs) or just the helix exposed state (8 nm). If we**
**consider the tight nanodisc as a proxy for the oligomeric state, this data suggests that the helix**
**exposed state is favored upon oligomerization. We do discuss this topic in the manuscript, but**
**further studies are required to come up with a definitive mechanistic picture.**

VI) The major stimulus in this study is using negatively charged lipids which is suggested to cause
both oligomerisation and N-terminal helix exposure or disorder. There is a lack of examination of the
mitochondrial context of VDAC1 or controls to prove that both are caused simply by disorder or
aggregation. Does the OMM outer leaflet contain lipids that would support N-helical exposure or
prevent it? Are there specific aspects of apoptotic stimuli that modulate this? i.e. does a certain
stimulus lead to an increase in negatively charged head groups in the OMM? Is this effect common to
other VDACs? E.g. VDAC2 is established as being involved in Bak sequestration and mitophagy
initiation caused by reactive oxygen species, are there residues or chemical environment differences
that explain the studied function of VDAC1 that wouldn't occur with other VDACs?

**We appreciate this comment. We have now improved the clarity of the state of knowledge in the**
**literature on what stimuli can trigger VDAC1 oligomerization. There are multiple factors**
**known, such as negative charge density (Matsko et al., 2001, Ostrander et al., 2001,**
**10.1074/jbc.M107067200) or upregulation of VDAC protein levels. A systematic cellular study**
**on the molecular causes of VDAC1 oligomerization-induced apoptosis would be highly**
**desirable. However, our study here rather aimed at providing a structural and functional**
**rationale of VDAC1 oligomerization on apoptosis induction, i.e. our starting point here is the**
**well-known fact that VDAC1 oligomerization is found in apoptotic cells. We tried our best to**
**include all relevant literature on possible triggers of VDAC1 oligomerization but feel that a**
**systematic cellular study is well beyond the scope of the present manuscript.**

**Furthermore, we are aware that there are also VDAC2 and VDAC3 isoforms. However,**
**VDAC1 is the most abundant isoform and thus studied here. We used a VDAC2 N-terminal**
**peptide (which is ~10 amino acids longer than VDAC1) to assay whether VDAC2 might be able**
**to induce pore formation of Bak, which is not the case. VDAC3-N is almost identical to VDAC1-**
**N. Thus, we did not investigate VDAC3 here. We now added more literature references to**
**emphasize the importance of VDAC2 and its possible complex formation with pro-apoptotic**
**Bak.**

Minor comments:

1) Line 117 may be referring to Figure S1d.

**Thanks. This issue has been fixed.**

2) There might be a potential artifact in Figure S1e in the L10C and L10C+E73V lanes (PC-only
conditions), as the overall exposure for these lanes looks significantly lower, even for non-specific
bands.

**Thank you for your comment. All lanes suffer from the residual PM40 that leads to the blank**
**region at around the two lanes at 60-120 kDa size range. Technically, it is impossible to have**
**different degrees of exposure in the applied silver staining procedure. Despite that general issue,**
**we do not see PM40 modification in PC but only in PC/PG mixtures since the L10C position is a**
**bit more internal than T6C and thus PM40 modification is more selective. Additionally, for the**
**differences of non-specific bands, one should consider the sensitivity of the silver staining**
**procedure for potential contaminations and degradations in different protein preps and even in**
**lipid stocks.**

3) There is a lack of clarity in the figures about the orientation of VDAC1 with regard to the
mitochondrial membrane. The figures would be improved significantly by ensuring the OMM, cytosol
and IMS are clearly indicated and all figures, including figure 5.

**We agree with this reviewer that this feature should be added to the figures to enhance**
**mechanistic clarity, which is now implemented in the updated main figures.**

4) The methods section also contains some ambiguity, with citations for parts of methods instead of
brief methodologies.

**We now added more details on methodologies in the Suppl. methods to enhance clarity.**

Reviewer #4 (Remarks to the Author):

I co-reviewed this manuscript with one of the reviewers who provided the listed reports. This is part
of the Nature Communications initiative to facilitate training in peer review and to provide
appropriate recognition for Early Career Researchers who co-review manuscripts.

**Thanks to this Early Career Researcher for their input. We sincerely support this initiative and**
**would like to emphasize the importance of obtaining experience in qualified peer review early**
**on.**

**We would like to thank all four reviewers for their final and positive comments, which helped**
**us to improve the paper.**

Reviewer #1 – Second-Round Comments

I would like to thank the authors for their substantial and careful revisions to the manuscript and for
their detailed point-by-point responses. Many of my original concerns—particularly regarding the
framing of VDAC1’s role, distinguishing between isoforms, and clarifying the structural evidence—
have been addressed, and the manuscript has been strengthened as a result. Several of my more
technical concerns (e.g., potential detergent artifacts, limitations of PEGylation and crosslinking
approaches, and interpretation of cryo-EM maps at modest resolution) remain only partially resolved.
However, I appreciate that the authors have acknowledged these caveats in the revised text and have
tempered some of their claims accordingly. This is an appropriate way forward given the scope of the
study and the technical challenges inherent in capturing dynamic conformational states of VDAC1.

**Thanks to this reviewer for the very positive final assessment of our work.**

The study still presents bold mechanistic conclusions that will likely generate discussion in the field,
but I believe the revisions have made the manuscript clearer, more balanced, and more transparent
about the limitations of the current data.

Recommendation: I support publication pending only minor editorial refinements, primarily to ensure
that speculative aspects remain framed as hypotheses rather than established conclusions.

**In the current manuscript we are not aware of any speculations beyond the framework of the**
**current literature. We here provide a functional model based on a large body of previous**
**literature as well as the data presented here in a consistent manner. In the introduction and the**
**discussion sections we describe the established pathways of apoptosis induction and try our best**
**to include the role of VDAC into a larger context. Thus, we are unable to tune down our**
**conclusion further without severely impacting the overall message and significance of the paper.**

Reviewer #2 (Remarks to the Author):

The authors have fully addressed my concerns, and in particular provide a convincing explanation
regarding the particle alignment and potential for averaging of the density for the smallest nanodiscs,
something that was asked by all three reviewers.

**Thanks to this reviewer for their positive conclusion on our manuscript. We appreciate their**
**constructive and positive comments in the first round of review.**

Reviewer #3 (Remarks to the Author):

I thank the authors for providing additional experiments to strengthen their hypothesis. In particular,
the CD data is very important to show that the stability of VDAC1 is not reduced in cholate. In my
opinion, it would be good to add Rebuttal Figure 10 to the supplemental figures of the manuscript.
The differences between PC and PC/PG lipids in terms of VDAC oligomerisation and PM40 binding
are very subtle. However, there appears to be a difference.

I find the explanation for the absence of a helix in the 8nm anodisc dataset very elegant – showing the
calculated result of uncertainty as a ring in the pore. I am satisfied by this explanation. I wonder if
some of this discussion should be included in the methods section of the manuscript?

Could the authors add IMS and cytosol labels to panels in Figure 1f to help orient the readers?

In summary, this is an exhaustive study providing an interesting insight into the potential mechanism
of VDAC1 involvement in the induction of apoptosis. With the incorporated changes, I think this is a

good fit for the journal and I congratulate the authors on this large amount of work.

**Thanks for these positive final comments. We appreciate the comment on VDAC1 helix**
**averaging and now included further details in the method section of the manuscript, and added**
**a short statement into the main text as well as a figure panel to Supplementary Fig 5. We also**
**fixed the mentioned labeling issue in Figure 1f.**

Reviewer #4 (Remarks to the Author):

I co-reviewed this manuscript with one of the reviewers who provided the listed reports. This is part
of the Nature Communications initiative to facilitate training in peer review and to provide
appropriate recognition for Early Career Researchers who co-review manuscripts.

**Thanks to this reviewer for their contribution to the review process.**